# The DeepMIP contribution to PMIP4: methodologies for selection, compilation and analysis of latest Paleocene and early Eocene climate proxy data, incorporating version 0.1 of the DeepMIP database

Christopher J. Hollis[1], Tom Dunkley Jones[2], Eleni Anagnostou[3,4], Peter K. Bijl[5], Margot J. Cramwinckel[5], Ying Cui[6], Gerald R. Dickens[7], Kirsty M. Edgar[2], Yvette Eley[2], David Evans[8], Gavin L. Foster[3], Joost Frieling[5], Gordon N. Inglis[9], Elizabeth M. Kennedy[1], Reinhard Kozdon[10], Vittoria Lauretano[9], Caroline H. Lear[11], Kate Littler[12], Lucas Lourens[5], A. Nele Meckler[13], B. David A. Naafs[9], Heiko Pälike[14], Richard D. Pancost[9], Paul N. Pearson[11], Ursula Röhl[14], Dana L. Royer[15], Ulrich Salzmann[16], Brian A. Schubert[17], Hannu Seebeck[1], Appy Sluijs[5], Robert P. Speijer[18], Peter Stassen[18], Jessica Tierney[19], Aradhna Tripati[20], Bridget Wade[21], Thomas Westerhold[14], Caitlyn Witkowski[22], James C. Zachos[23], Yi Ge Zhang[24], Matthew Huber[25] and Daniel J. Lunt[26].

[1] GNS Science, Lower Hutt, New Zealand

[2] School of Geography, Earth and Environmental Sciences, University of Birmingham, UK

[3] Ocean and Earth Science, National Oceanography Centre Southampton, University of Southampton, UK

[4] GEOMAR Helmholtz Centre for Ocean Research, Kiel, Germany

[5] Institute for Marine and Atmospheric Research, Utrecht University, Netherlands

6 Department of Earth and Environmental Studies, Montclair State University, Montclair, USA

7 Department of Earth, Environmental and Planetary Sciences, Rice University, Texas, USA

[8] Institute of Geosciences, Goethe University Frankfurt, Frankfurt am Main, Germany

[9] School of Chemistry & School of Earth Sciences, University of Bristol, UK

[10] Lamont–Doherty Earth Observatory of Columbia University, USA

[11] School of Earth and Ocean Sciences, Cardiff University, UK

[12] Camborne School of Mines & Environment and Sustainability Institute, University of Exeter, UK

[13] Bjerknes Centre for Climate Research and Department of Earth Science, University of Bergen, Norway

[14] MARUM – Center for Marine and Environmental Sciences, University of Bremen, Germany

[15] Department of Earth & Environmental Sciences, Wesleyan University, USA

[16] Department of Geography, Northumbria University, UK

[17] School of Geosciences, University of Louisiana at Lafayette, Louisiana, USA

[18] Department of Earth and Environmental Sciences, KU Leuven, Belgium

[19] Department of Geosciences, University of Arizona, USA

[20] Department of Earth and Planetary Sciences, Institute of the Environment and Sustainability, Department of Atmospheric and Oceanic Sciences, Center for Diverse Leadership in Science,, University of California, Los Angeles, USA

[21] Department of Earth Sciences, University College London, UK

[22] Department of Marine Microbiology and Biogeochemistry (MMB), NIOZ Royal Netherlands Institute for Sea Research and Utrecht University, Den Burg, The Netherlands

[23] Earth and Planetary Sciences Department, University of California, Santa Cruz, USA

[24] Department of Oceanography, Texas A&M University, USA

[25] Department of Earth, Atmospheric, and Planetary Sciences, Purdue University, USA

[26] School of Geographical Sciences, University of Bristol, UK

*Correspondence to*: Christopher J. Hollis (c.hollis@gns.cri.nz)

**Abstract.** The early Eocene (56 to 48 million years ago) is inferred to have been the most recent time that Earth's atmospheric
$CO_2$ concentrations exceeded 1000 ppm. Global mean temperatures were also substantially warmer than present day. As such, study of early Eocene climate provides insight into how a super-warm Earth system behaves and offers an opportunity to evaluate climate models under conditions of high greenhouse gas forcing. The Deep Time Model Intercomparison Project (DeepMIP) is a systematic model-model and model-data intercomparison of three early Paleogene time slices: latest Paleocene, Paleocene-Eocene thermal maximum (PETM) and early Eocene climatic optimum (EECO). A previous article outlined the
model experimental design for climate model simulations. In this article, we outline the methodologies to be used for the compilation and analysis of climate proxy data, primarily proxies for temperature and $CO_2$. This paper establishes the protocols for a concerted and coordinated effort to compile the climate proxy records across a wide geographic range. The resulting climate "atlas" will be used to constrain and evaluate climate models for the three selected time intervals, and provide insights into the mechanisms that control these warm climate states. We provide version 0.1 of this database, in anticipation that this
will be expanded in subsequent publications.

## 1 Introduction

Over much of the last 100 million years of Earth history, greenhouse gas levels and global temperatures were higher than present (Zachos et al., 2008; Foster et al., 2017). Because greenhouse gas levels are currently well above anything experienced during the modern natural climate state, ancient climate archives hold important clues to our possible future climate (IPCC,
2013). This is particularly true for those past times when climate was considerably warmer and greenhouse gas levels considerably higher than present. For instance, such intervals can provide information about the sensitivity of the climate to greenhouse gas forcing (e.g. Rohling et al., 2012; Zeebe, 2013; Caballero and Huber, 2013; Anagnostou et al., 2016) or reveal the behaviour of carbon cycle feedbacks under super-warm climate states (e.g. Carmichael et al., 2017). These times of past warmth also provide a powerful means to test the outputs of climate models because they represent actual realisations of how
the Earth system functions under conditions of greenhouse forcing comparable to the coming century and beyond. If the models can match the geological evidence of the prevailing climatic conditions, we can have greater confidence in their skill in

predicting our future climate. Similarly, differences between models and data could indicate aspects of models and/or data that require further development.

This is the rationale behind DeepMIP – the Deep-time Model Intercomparison Project (www.deepmip.org) – which brings together climate modelers and paleoclimatologists from a wide range of disciplines in a coordinated, international effort to improve understanding of the climate of these time intervals, to improve the skill of climate models, and to improve the accuracy and precision of climate proxies. The term "Deep-time" as applied here refers to the history of Earth prior to the Pliocene, or before 5 million years ago (Ma). DeepMIP is a working group in the wider Paleoclimate Modelling Intercomparison Project (PMIP4), which itself is a part of the sixth phase of the Coupled Model Intercomparison Project (CMIP6). In DeepMIP, we focus on three warm greenhouse time periods in the latest Paleocene and early Eocene (~57–48 Ma), and for the first time, carry out formal coordinated model–model and model-proxy intercomparisons.

We previously have outlined the model experimental design of this project (Lunt et al., 2017). Here we outline the recommended methodologies for selection, compilation and analysis of climate proxy data sets for three selected time intervals: latest Paleocene (LP), Paleocene-Eocene thermal maximum (PETM) and early Eocene climatic optimum (EECO). Section 2 outlines previous compilations, and Section 3 formally defines the time periods of interest. Sections 4 and 5 describe the proxies for sea surface and land air temperature (SST and LAT), respectively. Section 6 describes the proxies for atmospheric carbon dioxide ($CO_2$). For each proxy, we highlight the underlying science, the strengths and weaknesses, and recommendations for analytical methodologies. We focus on temperature in this article because it is the most commonly and readily reconstructed climatic variable and it is one of the most accurately represented variables in climate models. When combined with $CO_2$, it allows assessment of climate sensitivity, a key metric that integrates the behavior of the whole Earth system. However, the DeepMIP database will provide a structure for future compilations of other climate proxies, such as precipitation, evaporation, salinity, upwelling, bathymetry, circulation, currents and vegetation cover, but these are not discussed or compiled here. Section 7 outlines the structure of the planned DeepMIP database, which will accommodate the data presented in Supplementary Data Files 2–7 (which constitute Version 0.1 of the database) and additional datasets as they become available. Section 8 presents a preliminary synthesis of the paleotemperature data from these tables and includes a discussion of the geographic coverage and quality of existing paleotemperature data for the three selected time intervals.

**2 Previous climate proxy compilations**

The first global climate proxy compilations for the Cenozoic were based on deep-sea records of stable isotopes from benthic foraminifera (e.g., Shackleton, 1986; Miller et al., 1987, 2005, 2011; Zachos et al., 1994, 2001, 2008; Cramer at al., 2009; Bornemann et al., 2016), which are proxies for deep-sea temperature and ice volume (oxygen isotopes, expressed as $\delta^{18}O$) and carbon cycle changes (carbon isotopes, expressed as $\delta^{13}C$). This work has culminated in studies in which $\delta^{18}O$ and Mg/Ca ratios of benthic foraminifer tests (Lear et al., 2000) were combined to derive an independent estimate of bottom water

temperature (BWT) and thus separate the temperature and ice volume/sea level components in the $\delta^{18}$O record (Cramer et al., 2011).

Early attempts at comparable compilations for sea surface temperature (SST) (e.g., Shackleton and Boersma, 1981) were complicated by seafloor alteration of the oxygen isotope composition of planktic foraminifera (Schrag et al., 1995; Schrag, 1999). For the early Paleogene, the discovery that robust SST reconstructions could be derived from well-preserved foraminifera in clay-rich sediments (Pearson et al., 2001; Sexton et al., 2006), coupled with the development of the organic biomarker-based TEX$_{86}$ SST proxy (Schouten et al., 2002), shifted attention away from the deep sea to continental margin settings where $\delta^{18}$O-based SST reconstructions could be compared with SSTs derived from Mg/Ca, clumped isotopes and TEX$_{86}$ (Zachos et al., 2006; Pearson et al., 2007; John et al., 2008; Hollis et al., 2009; Keating-Bitonti et al., 2011). These relatively few Paleogene sites have formed the basis of several SST compilations, which were undertaken as part of previous model-proxy intercomparison efforts (Sluijs et al., 2006; Bijl et al., 2009; Hollis et al., 2009, 2012; Lunt et al., 2012; Dunkley Jones et al., 2013). In recent years, new compilations have been presented as part of targeted efforts to fill the geographic gaps identified by this earlier work (e.g. Frieling et al., 2014, 2017, 2018; Cramwinckel et al., 2018; Evans et al., 2018a). In comparison, there have been fewer compilations of land air temperature (LAT) proxy data (Greenwood and Wing, 1995; Huber and Caballero, 2011; Jaramillo and Cardenas, 2013; Naafs et al., 2018a). In our study, we have compiled and reviewed existing datasets for SSTs and LAT, calibrated them to a consistent time scale in order to identify the time intervals of interest, and recalculated SST and LAT using the methodologies outlined below (Supplementary Data Files 2–7). This represents the most comprehensive compilation of early Paleogene paleotemperature data published to date.

Compilations of atmospheric CO$_2$, the only greenhouse gas for which any proxy-based constraints exist, have tended to deal with the Cenozoic in its entirety (e.g. Beerling and Royer, 2011) or as part of longer compilations focused on the Phanerozoic (e.g. Royer et al., 2001; Royer, 2006; Foster et al., 2017). Here we review the latest understanding of the available proxies and summarize estimates of atmospheric CO$_2$ for the three focus intervals. These estimates will provide constraints for climate simulations and climate sensitivity studies.

### 3 Time intervals, correlation and variability

We have chosen to focus on three time intervals in the latest Paleocene and early Eocene (Fig. 1). These time intervals are the climate state immediately before (latest Paleocene, LP) and at the peak of a short-term but high-amplitude warming event (Paleocene-Eocene thermal maximum, PETM) and the subsequent long-term peak of Cenozoic warmth (early Eocene climatic optimum, EECO). The latter two time intervals are selected because they are the most extreme warm climates of the Cenozoic, but of very different durations, and so represent warm climate end-members for PMIP model experiments. They are also advantageous as they are readily identifiable in the stratigraphic record, the climate signal has a high signal-to-uncertainty ratio, the uncertainties in non-greenhouse gas boundary conditions (e.g. ocean gateways etc.) between the three intervals are

thought to be small, and a large amount of climate data has been generated by numerous studies over the last two decades. The latest Paleocene provides a reference "background" point for both the PETM and the EECO.

*Latest Paleocene (LP)*: In the most complete deep-sea isotope records (Westerhold et al., 2011, 2017, 2018; Littler et al., 2014), a gradual warming trend starting about 59 Ma leads up to the PETM. Some studies also indicate that the carbon isotope excursion (CIE) is directly preceded by warming (e.g. Sluijs et al., 2007a; Bowen et al., 2014). However, there is a ~1 million year (my) interval between this end-Paleocene warming (<10 ky duration – Sluijs et al., 2007b) and the base of magnetochron C24r in which the benthic foraminiferal $\delta^{18}O$ record is relatively stable (Westerhold et al., 2011, 2017, 2018; Littler et al., 2014). This is the preferred interval to adopt as representing background latest Paleocene conditions, although it is recognized that hiatuses occur in the uppermost Paleocene in several shelf sections (e.g. Hollis et al., 2012) and that many terrestrial sections lack the age control to identify this interval. In practice, many datasets for this LP interval come from the pre-PETM stratigraphy of studies that have good recovery of the onset of the PETM climate event (e.g. Dunkley Jones et al., 2013). These sections can be screened for any rapid warming associated with the onset of the PETM (Dunkley Jones et al., 2013), and although the age of the base of these successions is sometimes poorly constrained, they are assumed to be well within the ~1 my LP interval defined here. For reference, the LP interval is shown on Fig. 1 in relation to the benthic foraminiferal stable isotope compilation for North Pacific ODP Site 1209 (Westerhold et al., 2011, 2017, 2018) and South Atlantic IODP Site 1262 (Littler et al., 2014).

*PETM*: The PETM spans the first ~220 ky of the Eocene (55.93–55.71 Ma – Röhl et al., 2007; Westerhold et al., 2017) and is associated with a negative excursion in the $\delta^{13}C$ of the global exogenic carbon pool (Koch et al., 1992; Dickens et al., 1995; Zachos et al., 2008). Although the magnitude and shape of the CIE exhibits variation between sites and measured substrate (Röhl et al., 2007), in the most complete records it is characterized by a rapid shift to peak negative values within the first ~20 ky of the event (Westerhold et al., 2018). Peak negative $\delta^{13}C$ values within the CIE are closely coupled to PETM peak temperatures, as evident from geochemical proxies (Fig. 2) and the abundance of warm-climate fossil species at higher latitudes (Sluijs et al., 2007a, 2011; McInerney and Wing, 2011; Sluijs and Dickens, 2012; Dunkley Jones et al., 2013; Eldrett et al., 2014; Suan et al., 2017) and the disappearance of some fossil groups, such as reef-building corals and dinoflagellates, in low latitudes (Speijer et al., 2012; Frieling et al., 2017). We recognise that each record will comprise onset, peak and recovery intervals that may vary in timing and duration (Fig. 2). In keeping with Dunkley Jones et al. (2013) and Frieling et al. (2017), our compilation identifies the peak PETM interval in each record based on the shape of the $\delta^{13}C$ excursion (interval of minimum values) and temperature proxies (interval of maximum values).

*EECO*: The EECO is identified in global compilations of benthic foraminiferal $\delta^{18}O$ as a prolonged episode of deep-sea warming between ~53 and ~50 Ma (Zachos et al., 2008; Cramer et al., 2009; Kirtland-Turner et al., 2014). The effect of local temperature regime complicates site-to-site correlations based on $\delta^{18}O$ alone, and for this reason, several studies have also

used $\delta^{13}C$ stratigraphy for high temporal resolution correlations. Studies of pelagic carbonate sequences in New Zealand (Slotnick et al., 2012, 2015) place the base of the EECO at the "J" event (Cramer et al., 2003), a rather subdued CIE that lies between the well-defined hyperthermals ETM2 (H event) and ETM3 (K event), which coincides with a marked increase in terrigenous clay abundance (Fig. 3). Lauretano et al. (2015) show that the J event is associated with a small negative $\delta^{18}O$

excursion in benthic foraminifera that is followed by a baseline negative shift in $\delta^{18}O$ at south Atlantic ODP Sites 1262 and 1263 (Fig. 1). The J event is also associated with a rapid turnover from planktic foraminiferal assemblages dominated by the genus *Morozovella* to assemblages dominated by *Acarinina* (Frontalini et al., 2016; Luciani et al., 2016, 2017). The J event equates to CIE C24n.2rH1 in the notation developed by Sexton et al. (2011) and corresponds approximately to the onset of the EECO previously indicated in global benthic $\delta^{18}O$ compilations (Zachos et al., 2008; Cramer et al., 2009; Kirtland-Turner et

al., 2014). Slotnick et al. (2012, 2015) suggested that the top of the EECO may coincide with the top of the clay-rich interval in the New Zealand sequence, which lies within Chron 22n (Dallanave et al., 2015) (Fig. 3). This is consistent with the global benthic foraminiferal $\delta^{18}O$ record, in which cooling begins at ~50 Ma (Zachos et al., 2008; Cramer et al., 2009; Kirtland-Turner et al., 2014; Lauretano et al., 2015, 2018; Bornemann et al., 2016). At mid-Atlantic ODP Site 1258, the termination of the EECO is placed at the base of a positive shift in benthic $\delta^{18}O$ that follows a hyperthermal event identified as CIE C22nH3

(Sexton et al., 2011).

A new high-resolution, astronomically-calibrated, benthic foraminiferal record at Site 1209, in the North Pacific (Westerhold et al., 2017, 2018), provides further support for these correlations (Fig. 1). Based on these studies, we use a wide definition of the EECO interval as the benthic foraminiferal $\delta^{18}O$ minimum that extends from the J event (CIE CH24n.2rH1; 53.26 Ma) to

the uppermost Chron C22n CIE, C22nH5 (49.14 Ma), an interval of 4.12 Myrs. The top of the EECO is well-defined by the onset of a cooling trend that follows CIE C22nH5. The base of the EECO is less well-defined. Oscillations in $\delta^{18}O$ occur from ~54 to ~52 Ma, with the most distinct negative shift in $\delta^{18}O$ coinciding with the M event (CIE C23rH2) at 51.97 Ma. It is acknowledged that choice of which of multiple CIEs we use to define the base and top of the EECO is somewhat arbitrary and serves to highlight a particular issue with this time slice. In addition to broad-scale warming, the succession of orbitally-paced

hyperthermals that begin in the late Paleocene continue through the EECO (Kirtland et al., 2014; Westerhold, 2018). Consequently, some may argue that averaging proxy data for the EECO is analogous to averaging data from glacial-interglacial cycles for a Pleistocene climate reconstruction. Where possible, it is important to differentiate background EECO conditions from the significantly warmer climatic conditions that characterize hyperthermals within the EECO (Westerhold et al., 2018).

All of the above intervals are defined with timescale-independent stratigraphic markers: the LP from the base of magnetochron C24r to the first sign of pre-PETM warming; the PETM interval based on the identification of the CIE and associated characteristic warming patterns; and the EECO is bounded by CIEs CH24n.2rH1 (J event) and C22nH5. For an absolute timescale, our study benefits from ongoing efforts to complete the astronomical tuning of the geological timescale for the Paleocene and early Eocene (Lourens et al., 2005; Westerhold et al., 2008, 2017, 2018; Littler et al., 2014; Lauretano et al.,

2015), as shown in Figs. 1 and 3. We recognise, however that most existing age models and biostratigraphic schemes are referenced to the GTS2012 timescale (Gradstein et al., 2012), and this is what we use for the purposes of data compilation. However, the DeepMIP database includes stratigraphic levels (depth or height) for all samples and information on age control for each site, which will facilitate updates to new age models as they become available.

## 4 Marine proxies for sea temperature

In this section, we outline the four main approaches for reconstructing Paleocene and Eocene sea temperatures: oxygen isotopes, Mg/Ca ratios, clumped isotopes and TEX$_{86}$. For each proxy, we outline (1) the underlying theoretical background, (2) strengths, (3) weaknesses, and (4) recommendations on methodologies. We focus on sea surface temperatures (SST) and have compiled data from published studies of planktic foraminiferal δ$^{18}$O (Supplementary Data File 3) and Mg/Ca ratios (Supplementary Data File 4), clumped isotopes from benthic foraminifera and molluscs (Supplementary Data File 5) and TEX$_{86}$ (Supplementary Data File 6). These data comprise 1701 samples from 40 drillholes or onshore sections.

### 4.1 Oxygen isotopes

### 4.1.1 Theoretical background of oxygen isotopes

Oxygen isotope paleothermometry is based on the temperature-dependent fractionation of the stable isotopes of oxygen ($^{16}$O and $^{18}$O, expressed as δ$^{18}$O) into biogenic calcite from its growth medium, typically seawater. Pearson (2012) provides more details on the development of the δ$^{18}$O proxy. The temperature dependence of δ$^{18}$O is a combination of thermodynamic and vital effects (e.g., Kim and O'Neil, 1997; Bemis et al., 1998). Most δ$^{18}$O paleotemperature reconstructions use the shells of foraminifera, a group of protists that live and calcify either as plankton in the ocean's surface mixed layer, thermocline or sub-thermocline depths, or as benthos on or just below the sea floor. Temperature is calculated by an empirical calibration to quantify the fractionation between the δ$^{18}$O of ancient seawater and biogenic calcite (Bemis et al., 1998).

### 4.1.2 Strengths of oxygen isotopes

Foraminiferal oxygen isotopes have been the primary proxy for reconstructing ocean temperatures spanning the past ~120 million years, largely due to the low analytical cost, small sample size requirements and relative ease of measurement. Analytical uncertainty on δ18O measurements is typically small ±0.1 ‰ (equivalent to <1ºC). Moreover, the theoretical basis for temperature-dependent fractionation is firmly tied to field- and laboratory-based relationships between foraminifer test δ$^{18}$O values and temperature (e.g., Kim and O'Neil, 1997; Bemis et al., 1998; Lynch-Stieglitz et al., 1999). The range of planktic foraminiferal depth habitats also allows for the reconstruction of water column profiles and thermocline structure (e.g., Birch et al., 2013; John et al., 2013), which can be compared to modelled upper ocean structure (e.g., Lunt et al., 2017). The relatively short life span of planktic foraminifera, two weeks to one month for most modern species, can also provide constraints on palaeo-seasonality (Pearson, 2012). Further, based on the assumption that deep-water formation is largely

focussed at high (subpolar) latitudes throughout the Cenozoic (Cramer et al., 2011), benthic foraminiferal $\delta^{18}$O values can (arguably) be used as a SST proxy in these areas.

### 4.1.3 Weaknesses of oxygen isotopes

The dissolution and subsequent replacement of primary biogenic calcite by inorganic calcite (recrystallization/ diagenesis) in pelagic, carbonate-rich sediments during early diagenesis is known to shift planktic foraminiferal calcite to higher values (Schrag et al., 1995), with the effect that most low and mid-latitude $\delta^{18}$O-derived Paleogene SSTs from deep-ocean carbonate-rich successions may be systematic underestimates (Pearson et al., 2001, 2007; Tripati et al., 2003; Sexton et al., 2006; Pearson and Burgess, 2008; Kozdon et al., 2011; Edgar et al., 2015). The effect of seafloor recrystallization and diagenesis on SST

estimates will be proportionally less significant in areas where cooler surface waters are closer to deep ocean temperatures, which may be the case in some Paleogene high-latitude or upwelling regions, and is considered to be insignificant for benthic foraminifera because temperatures during early diagenesis are very close to growth temperatures (Schrag et al., 1999; Edgar et al., 2013; Voigt et al., 2016). During diagenesis, foraminifer tests may also become overgrown and/or infilled with calcite precipitated from sediment pore waters. Where marine sediments are exposed on land, those secondary precipitates can

incorporate oxygen from isotopically light meteoric waters and hence yield artificially warm apparent temperatures. For these reasons, temperature estimates from foraminiferal $\delta^{18}$O are only considered to be reliable where a good state of preservation has been confirmed by SEM examination and illustration. Well-preserved (non-recrystallized or glassy) late Paleocene to early Eocene foraminifera have been reported from low permeability clay-rich facies in shallow marine or hemipelagic settings from Tanzania (Pearson et al., 2004; 2009; Sexton et al., 2006), the New Jersey margin and California (Zachos et al., 2006; John et

al., 2008; Makarova et al., 2017), New Zealand (Hollis et al., 2012), and Nigeria (Frieling et al., 2017).

One key assumption for $\delta^{18}$O-based temperatures is that foraminifera precipitate their tests in isotopic equilibrium with seawater. However, foraminiferal physiology (e.g., respiration, metabolism, biomineralization, photosymbiosis) and ecology (e.g., depth migration during life cycle, seasonality), often termed 'vital effects', commonly lead to isotopic offsets that can

bias temperature reconstructions (e.g., Urey, 1951; Birch et al., 2013). The biology and ecology of foraminifera may also vary in either time or space. For instance, the depth habitat of a species may change in response to rapid environmental change, or due to evolution within a lineage. Foraminifera that host algal photosymbionts may also be subject to bleaching events, which can substantially alter the test micro-environment within which calcification, and the associated isotopic fractionation, occurs (Wade et al., 2008; Edgar et al., 2013; Luciani et al., 2016; Si and Aubry, 2018). In some sites there is evidence that planktonic

foraminifera and other eukaryotes disappeared from the record during peak PETM warming, possibly because environmental conditions became too extreme (Aze et al., 2014; Frieling et al., 2018) in which case peak conditions would go unrecorded by this proxy. Enhanced dissolution in the PETM would have the same effect.

Calculating ancient SST from foraminiferal $\delta^{18}O$ requires an estimation of the oxygen isotopic composition of seawater ($\delta^{18}O_{sw}$) at the time of precipitation. This is not straightforward because $\delta^{18}O_{sw}$ varies spatially in the surface ocean, largely following patterns of salinity (Zachos et al., 1994; Rohling, 2013), and temporally due to changes in the cryosphere (Broecker, 1989; Cramer et al., 2009). Large and spatially variable changes in the intensity of the hydrological cycle are inferred across the PETM (Bowen et al., 2004; Zachos et al., 2006; Pagani et al., 2006; Carmichael et al., 2017), hence it is unlikely that $\delta^{18}O_{sw}$ at any single location remained constant through the Paleocene–Eocene interval. Continental margin settings, where foraminifera are typically well preserved, may be particularly sensitive to changes in $\delta^{18}O_{sw}$ related to the hydrological cycle.

Finally, culture studies and field observations demonstrate that the $\delta^{18}O$ value of foraminiferal calcite decreases as the pH of the culture medium increases (Spero et al., 1997; Bijma et al., 1999; Zeebe, 1999, 2001; Russell and Spero, 2000). This 'pH effect' can influence ancient SST reconstructions if seawater pH varied rapidly or was significantly different from today (Zeebe, 2001). Studies suggest that early Paleogene surface ocean pH was as much as ~0.5 units lower than today (Penman et al., 2014; Anagnostou et al., 2016; Gutjahr et al., 2017), which implies that background $\delta^{18}O$-based SST estimates for this time interval could be ~3°C too low, or even more so for the PETM when pH may have declined by a further ~0.3 units (Uchikawa and Zeebe, 2010; Aze et al., 2014).

### 4.1.4 Recommended methodologies for oxygen isotopes

Here we outline our recommendations for generating $\delta^{18}O$ data from fossil foraminifera, and for converting $\delta^{18}O$ values into temperature estimates. We have compiled available planktic foraminiferal $\delta^{18}O$ data from ten DSDP, ODP and IODP sites and nine onshore sections (Supplementary Data File 3). Using the methods outlined below, we have calculated SSTs and compiled a summary of proxy-specific SST estimates for each time slice (LP, PETM, EECO). SST estimates are based on species that are inferred to have inhabited near-surface waters or the mixed layer. Data for deeper-dwelling (thermocline) planktic species are also included in Supplementary Data File 3. For compilations of benthic foraminiferal $\delta^{18}O$, see Zachos et al. (2008), Cramer et al. (2009, 2011) and Westerhold et al. (2017, 2018) although these data are not considered here.

Depth ecologies of all Paleogene species, as inferred mainly from stable isotope evidence, have been compiled by Aze et al. (2011). The principal groups used for sea surface temperature reconstruction are open-ocean mixed-layer species with or without algal symbionts and high-latitude species (ecogroups 1, 2 and 5 of Aze et al. [2011]). The main mixed-layer genera for the DeepMIP time slices are *Morozovella* and *Acarinina* but other relevant groups are *Igorina*, *Planorotalites* and *Pseudohastigerina*. Different species of *Morozovella* and *Acarinina* may exhibit consistent offsets indicating a degree of depth stratification within the mixed layer and upper thermocline, possibly related to sinking at time of reproduction. Hence SST reconstructions from different 'mixed layer' species may vary by up to several degrees Celsius. Because of this, combining various species of the same genus in analyses (e.g., measuring *Acarinina* spp.) is likely to produce underestimates of SST. In this compilation, we derive an average SST estimates from 'mixed layer' species for each site in a given time slice. This is to

promote consistency and aid inter-site comparisons but is considered to be a conservative approach to estimating SSTs. For future work we recommend more detailed investigation of the various important species and identification of those species which most faithfully record the warmest upper ocean mixed layer conditions.

To expand on the $\delta^{18}O$ database, we propose a three-pronged approach. First, analysis of new and classic carbonate-rich sites containing recrystallized foraminiferal tests should involve the novel technique that uses a Secondary Ion Mass Spectrometer (SIMS). Pioneering SIMS studies by Kozdon et al. (2011, 2013) suggest that areas furthest from the test exterior are less susceptible to diagenetic overprinting. These areas yield SSTs up to ~8°C warmer than conventional analyses from the same sample and are in better agreement with $\delta^{18}O$-based SSTs from glassy tests (Kozdon et al., 2011). On a cautionary note, a

recent study by Wycech et al. (2018) has reported an offset between SIMS and traditional IRMS analyses for modern foraminifera, with SIMS $\delta^{18}O$ values being ~0.9‰ lower. Further study is needed to determine if this offset also affects fossil foraminifera. The SIMS technique provides hope for recovering reliable SSTs from recrystallized foraminiferal tests but it is time intensive and is not a practical approach for the analysis of all samples. Thus, our second recommendation is that whole specimen analyses are undertaken together with the SIMS analysis, in order to constrain the magnitude of diagenetic bias on

IRMS $\delta^{18}O$ values. For lithologically uniform sediments, quantitative estimates of this diagenetic bias in a few widely spaced samples could provide calibration points for higher resolution data generated by conventional whole shell methods. Whole specimen isotopic analyses should be species-specific and use a prescribed size fraction (e.g., 250-300 µm or 300-355 µm) to minimise variations in vital effects (Birch et al., 2013). Our third recommendation is that new sites that contain glassy foraminifera are sought out to provide the material for whole specimen analysis, as well as SIMS analysis of selected samples

to further validate the two methods. All samples included within the database are categorized as either glassy or recrystallized, using the criteria of Sexton et al. (2006) and Pearson and Burgess (2008), as a guide to reconstructed SST reliability.

Empirically derived $\delta^{18}O$–temperature calibrations can differ by several degrees Celsius, , although offsets decrease with increasing temperature (Bemis et al., 1998; Pearson, 2012). Use of multiple equations may capture a range of plausible

temperature values but our preference is the calibration of Kim and O'Neil (1997) for inorganic calcite, which is appropriate for the Paleogene because it is based on inorganic calcite precipitated from water temperatures between 10–40°C. Both epibenthic and asymbiotic planktic foraminifera yield values close to the resulting regression (Bemis et al., 1998; Costa et al., 2006). Field or laboratory studies only include calcite precipitated up to 30°C (e.g., Bemis et al., 1998; Lynch-Stieglitz et al., 1999), yet Paleogene SSTs likely fall close to or above the upper limit of these studies. The recommended calibration, as

modified by Bemis et al. (1998), derives temperature from Eq. (1):

$$T = 16.1 - 4.64(\delta^{18}O_C - \delta^{18}O_{sw}) + 0.09(\delta^{18}O_C - \delta^{18}O_{sw})^2 \tag{1}$$

where T is the water temperature in ˚C, $\delta^{18}O_C$ and $\delta^{18}O_{sw}$ are the $\delta^{18}O$ of calcite (‰ VPDB) and ambient seawater (‰ VSMOW), respectively. This equation may overestimate SST for symbiont-hosting planktic foraminifera by ~1.5°C based on the consistent 0.3‰ offset observed between *Orbulina universa* grown under high vs. low light conditions (Spero and Williams, 1988; Pearson, 2012). This offset is likely caused by algal photosymbionts modifying the pH in the calcifying microenvironment (Zeebe et al., 1999). *Orbulina universa* is inferred to share a similar ecology to the dominant Eocene genera *Morozovella* and *Acarinina* typically analysed for Paleogene SSTs (e.g., Shackleton et al., 1985; D'Hondt et al., 1994) and is currently the best modern analogue for which calibration data are available. However, we do not recommend applying a symbiont correction to these genera because of uncertainties in photosymbiont activity levels in Paleogene conditions.

It is assumed that these $\delta^{18}O$ calibrations are insensitive to evolving seawater chemistry, unlike some trace element proxies (e.g., Evans and Müller, 2012). These calibrations do, however, require an appropriate estimate of $\delta^{18}O_{sw}$ at the time of test formation. Changes in global mean $\delta^{18}O_{sw}$ are largely driven by continental ice volume and isotopic composition. Various estimates have been proposed for the adjustment of the global ocean value for ice-free conditions of the early Paleogene, Shackleton and Kennett's (1975) initial estimate of -1.00‰ can be compared with the more recent estimates of -1.11‰ of L'Homme et al. (2005) and -0.89‰ of Cramer et al. (2011). Differences arise from uncertainty surrounding the mean $\delta^{18}O$ value of the modern ice caps (see Pearson [2012] for discussion). Pending resolution of these discrepancies we here use the value -1.00‰ for early Paleogene $\delta^{18}O_{sw}$ under ice-free conditions which is the mean of the L'Homme et al. (2005) and Cramer et al. (2011) estimates and identical to the estimate of Shackleton and Kennett (1975) upon which many historical temperature estimates have relied. A correction for the local effects of salinity and hydrology on $\delta^{18}O_{sw}$ should also be incorporated into SST estimates where possible. Zachos et al. (1994) developed a correction using present-day $\delta^{18}O_{sw}$ latitudinal gradients, which is widely used despite acknowledged shortcomings. The correction is applied globally but is only based on Southern Hemisphere data and does not incorporate local freshwater runoff effects on continental margins or the influence of boundary currents. It also ignores expected changes in the relationship between latitude and $\delta^{18}O_{sw}$ through time. Isotope-enabled climate models have been used to generate predictions of early Paleogene surface ocean $\delta^{18}O_{sw}$, distributions, following an assumption of mean average $\delta^{18}O_{sw}$ conditions (e.g., Tindall et al., 2010; Roberts et al., 2011), but the $\delta^{18}O_{sw}$ fields generated are dependent on the surface climatology of the model concerned (Hollis et al., 2012). The use of model-derived $\delta^{18}O_{sw}$ to generate proxy estimates of SST also introduces a problematic model-dependency within the proxy data set. Here, we chose to update the approach of Zachos et al. (1994) using the more highly-resolved global gridded (1° x 1°) data set of modern $\delta^{18}O_{sw}$ produced by LeGrande and Schmidt (2006). We calculated $\delta^{18}O_{sw}$ for each site by relating the site's paleolocation to 10° latitudinal bins in the modern data set (see Supplementary Data File 3). Median values and 95% confidence intervals were calculated using paleolocations derived from both paleomagnetic and mantle-based reference frames (see Sect. 7.2). We encourage future studies to improve the empirical fit to the modern spatial variability in $\delta^{18}O_{sw}$ but also recognise that the modern system only provides a first-order estimate of spatial $\delta^{18}O_{sw}$ patterns in deep-time. With improved data coverage, paired $\delta^{18}O$–Mg/Ca and $\delta^{18}O$–$\Delta47$ analyses hold promise for direct reconstructions of ancient $\delta^{18}O_{sw}$ variability.

We do not recommend applying a pH correction to $\delta^{18}O$ data because of significant uncertainty in the magnitude of the effect below pH 8 (Uchikawa and Zeebe, 2010), which is inferred to be the upper limit for most of our Paleogene records (Anagnostou et al., 2016; Penman et al., 2014; Gutjahr et al., 2017). Further, the pH–$\delta^{18}O_C$ sensitivity of asymbiotic planktic and benthic foraminifera is not well known (e.g., Anagnostou et al., 2016; McCorkle et al., 2008; Mackensen, 2008).

## 4.2 Mg/Ca ratios

### 4.2.1 Theoretical background of Mg/Ca ratios

The sensitivity of foraminiferal Mg/Ca ratios to temperature has a basis in thermodynamics (Lea et al., 1999) through the exponential temperature dependence of any reaction for which there is an associated change in enthalpy. However, most species do not conform to theoretical calculations (Rosenthal et al., 1997), being characterised by a temperature sensitivity of Mg-incorporation around 2-3 times greater than for inorganic calcite. The reasons for this remain elusive (e.g., Bentov and Erez, 2006), necessitating empirical calibration of foraminiferal Mg/Ca to temperature (e.g., Anand et al., 2003).

### 4.2.2 Strengths of Mg/Ca

As with $\delta^{18}O$ paleothermometry, Mg/Ca paleothermometry can be applied to both benthic and planktic foraminifera and be used to constrain the past thermal structure of the water column (Tripati and Elderfield, 2005). The long residence time of Ca and Mg in the ocean means that the Mg/Ca of seawater (Mg/Ca$_{sw}$) can be treated as constant over short timescales ($<10^6$ yrs), which removes one source of uncertainty in calculating relative changes in temperature (e.g., Zachos et al., 2003; Tripati and Elderfield, 2004). A major advantage of Mg/Ca is its use in paired measurements with $\delta^{18}O$ on the same substrate, which allows for the deconvolution of $\delta^{18}O_{sw}$ and temperature effects on measured foraminiferal $\delta^{18}O$. As Mg/Ca measurements are increasingly routine in many laboratories, are relatively inexpensive to produce, and have low sample-size requirements, high-resolution time series are readily achievable. Technological developments in laser ablation-ICPMS and other spatially-resolved methodologies, such as Electron Probe Microanalysis (EPMA) and SIMS, allow for Mg/Ca measurements on or within a single individual test, providing new information on foraminiferal ecology and short-term environmental variability (e.g., Eggins et al., 2004; Evans et al., 2013; Spero et al., 2015). Planktic foraminiferal Mg/Ca may also be more robust to shallow burial diagenetic recrystallization than $\delta^{18}O$, based on values measured in the same material (Sexton et al., 2006).

### 4.2.3 Weaknesses of Mg/Ca

A long-standing challenge for the deep-time application of the Mg/Ca temperature proxy is that the seawater Mg/Ca ratio (Mg/Ca$_{sw}$) influences shell Mg/Ca, but there is still debate over how Mg/Ca$_{sw}$ has varied through time (Horita et al., 2002; Coggon et al., 2010; Broecker and Yu, 2011; Evans and Müller, 2012). Non-thermal influences on foraminiferal Mg/Ca ratios can also be difficult to account for, including pH, bottom water carbonate saturation state, and sample contamination (Barker et al., 2003; Regenberg et al., 2014; Evans et al., 2016b). Determining the reliability of Mg/Ca paleotemperatures requires an

understanding of these challenges, and the use of independent paleoenvironmental proxies where available, such as boron isotopes to constrain carbonate system parameters (Anagnostou et al., 2016).

***Impact of foraminiferal preservation on the Mg/Ca paleothermometer***: Many foraminifer tests from deep ocean sediments are affected by diagenetic alteration (Pearson et al., 2001; Edgar et al., 2015), with the type and extent of alteration controlled by original test morphology, taphonomic processes, the characteristics of the host sediment and burial history. For example, planktic foraminifera tests can undergo immediate post-mortem or post-gametogenic dissolution as they sink into deeper, less carbonate saturated waters (Brown and Elderfield, 1996). In general, $MgCO_3$ is more soluble than $CaCO_3$ such that dissolution tends to decrease foraminiferal Mg/Ca. This may result in artificially low Mg/Ca-based temperature estimates if unaccounted for (e.g., Rosenthal and Lohmann, 2002; Regenberg et al., 2014; Fehrenbacher and Martin, 2014), although not in all cases (e.g., Sadekov et al., 2010; Fehrenbacher and Martin, 2014). The effects of dissolution can be minimized by selecting sites with relatively shallow paleodepths, above the calcite lysocline (e.g., <2000 m). Foraminiferal shells can also be subject to diagenetic overgrowths of various mineral phases, including oxy-hydroxides and authigenic carbonates, depending on seafloor and sub-seafloor conditions (e.g., Boyle, 1983). Recent work suggests that some textural recrystallization of planktic foraminiferal tests may occur in semi-closed chemical conditions, potentially allowing original geochemical signals to be retrieved using microsampling techniques (e.g., Kozdon et al., 2011, 2013, see Sect. 5.2.5).

***Challenges for benthic foraminiferal Mg/Ca paleothermometry***: The temperature sensitivity of Mg incorporation into benthic foraminiferal calcite varies between species, necessitating genus- or species-specific temperature calibrations (Lear et al., 2002). Fortunately, some extant species are common throughout the Cenozoic (e.g. *Oridorsalis umbonatus* – Lear et al., 2000) and offer a means to develop calibrations for coeval extinct species. There is also no consensus as to whether benthic Mg/Ca-temperature relationships are best described by linear or exponential fits (Cramer et al., 2011; Evans and Müller, 2012; Lear et al., 2015); we recommend that calibrations are applied with caution where Mg/Ca ratios are outside the range for which temperatures have been empirically determined.

Present-day benthic foraminifera appear to increase their discrimination against magnesium when calcifying in waters with very low carbonate ion saturation state ($\Delta CO_3^{2-}$); a relationship that has been empirically quantified for some species (Elderfield et al., 2006; Rosenthal et al., 2006). Measurements of benthic foraminiferal B/Ca in tandem with Mg/Ca can be used to identify temporal variations in $\Delta CO_3^{2-}$ (Yu and Elderfield, 2007) and may provide a means of correcting for this secondary effect, although it can be difficult to identify the threshold for a $\Delta CO_3^{2-}$ influence within downcore records (e.g., Lear et al., 2010). Mg/Li ratios may have a more consistent empirical relationship with temperature than Mg/Ca (Bryan and Marchitto, 2008). However, limited understanding of how the Mg/Li seawater ratio has varied over geological time means that this proxy can only be used as a guide to relative temperature change in deep time studies (Lear et al., 2010). The calcification of endobenthic foraminifera species within buffered porewaters may make them relatively insensitive to variations in bottom

water $\Delta CO_3^{2-}$ (Zeebe, 2007; Elderfield et al., 2010). However, the saturation state of porewaters is dependent on many factors, and likely also varies through time (Weldeab et al., 2016). We recommend that tandem trace metal ratios that are sensitive to carbonate saturation state (e.g., B/Ca, Li/Ca) are examined to assess downcore $\Delta CO_3^{2-}$ variations, even in infaunal records (e.g., Lear et al., 2010; Mawbey and Lear, 2013; Lear et al., 2015).

Evidence from multiple proxies indicates that early Paleogene Mg/Ca$_{sw}$ was significantly lower than the modern value (Horita et al., 2002; Coggon et al., 2010; Lear et al., 2015; Evans et al., 2018a). Correcting benthic Mg/Ca data for secular changes in seawater chemistry is complicated by the fact that the benthic foraminiferal magnesium partition coefficient ($D_{Mg}$ = Mg/Ca$_{CALCITE}$ / Mg/Ca$_{sw}$) decreases with increasing Mg/Ca$_{sw}$ according to a power function (Ries, 2004; Hasiuk and Lohmann,

2010; Evans and Müller, 2012; Lear et al., 2015). Moreover, the sensitivity of shell Mg/Ca to changes in Mg/Ca$_{sw}$, specifically, the curvature of the power relationship that relates the two factors appears to be genus-specific. Lear et al. (2015) and Evans et al. (2016b) argued for a low sensitivity of shell Mg/Ca to Mg/Ca$_{sw}$ for *Oridorsalis* and the endobenthic genus *Uvigerina*. In contrast, a higher sensitivity is thought to characterise the epibenthic genus *Cibicidoides/Cibicides* (Evans et al., 2016b), which is also widely used in paleoclimate studies. Evaluating this aspect of benthic foraminiferal geochemistry is challenging and

has yet to be assessed in other widely utilized species. As such, best practice would be to report to what extent Mg/Ca temperatures would change when considering the uncertainty in the slopes of these seawater-shell Mg/Ca relationships.

***Challenges for planktic foraminifera Mg/Ca***: The relationship between planktic Mg/Ca ratios and temperature is species- or group-specific (e.g. Regenberg et al., 2009), such that species-specific calibrations should be used whenever possible.

Nonetheless, many planktic foraminifera conform to a broader Mg/Ca-temperature relationship (Elderfield and Ganssen, 2000; Anand et al., 2003), which is one method by which modern calibrations can be applied to extinct taxa. As for benthic foraminifera, when working with pre-Pleistocene samples the control exerted by changes in seawater elemental chemistry over geological time must also be considered. Culture experiments in modified seawater demonstrate not only that Mg/Ca$_{sw}$ impacts planktic foraminifera shell chemistry (Delaney et al., 1985), but also that the slope of the Mg/Ca-temperature relationship may

be sensitive to Mg/Ca$_{sw}$ (Evans, et al., 2016b). In addition, several other non-thermal controls on Mg/Ca should be considered when interpreting data from planktic foraminifera. Culture and core-top studies demonstrate a relatively minor salinity effect (Kısakürek et al., 2008; Hönisch et al., 2013) in which, for example, a 2 PSU salinity increase results in a temperature overestimate of ~1°C. In contrast, the carbonate system has been shown to have a large influence on Mg/Ca in several species (Lea et al., 1999; Russell et al., 2004; Evans, et al., 2016a). Lower pH and/or [$CO_3^{2-}$] results in higher shell Mg/Ca; for example,

a 0.1-unit pH decrease results in a temperature overestimate of ~1°C. The effect of the carbonate system on planktic Mg/Ca has been identified in sediment-trap as well as culture studies (Evans et al., 2016a; Gray et al., 2018), with Gray et al. (2018) demonstrating that the widely used Mg/Ca-temperature sensitivity of ~9%°C$^{-1}$ in *Globigerinoides ruber* is an artefact of the covariance of temperature and pH, through the temperature effect on the dissociation constant of water. Specifically, the pH of seawater decreases with increasing temperature, resulting in an increase in the incorporation of Mg into planktic

foraminiferal calcite due to both processes (see Evans et al., 2018b). The secondary pH effect accounts for around one third of the observed increase in Mg/Ca ratios at higher temperatures, leaving a primary Mg/Ca "temperature only" sensitivity of 6%°C$^{-1}$ (Gray et al., 2018), which is significantly lower than that widely utilized. Whilst these factors may be accurately accounted for in the recent geological past (Gray & Evans, 2019), it is challenging to account for them in deep time because high-

resolution pH records are scarce (see below for detailed recommendations).

### 4.2.4 Recommended methodologies for Mg/Ca

Here we outline our recommendations for generating Mg/Ca data from fossil foraminifera and for converting these data into temperature estimates. We have compiled available planktic Mg/Ca data from five DSDP and ODP sites and six onshore

sections and, using the methods outlined below, calculated SSTs and associated uncertainties for the late Paleocene and early Eocene (Supplementary Data File 4). SST estimates are based on species that are inferred to have inhabited near-surface waters or the mixed layer. Data for deeper-dwelling (thermocline) planktic species are also included in Supplementary Data File 4 but are not discussed here. For compilations of benthic foraminiferal Mg/Ca ratios, see Cramer et al. 2011).

*Sample preparation*: Foraminifera need to be thoroughly cleaned prior to analysis. Clay and organic contaminants are removed using a short oxidative procedure, whereas removal of metal oxide contaminants requires a longer procedure including a reductive step (Boyle and Keigwin, 1985). These two procedures result in offsets in Mg/Ca values, which must be corrected when making comparisons to other records (e.g., Barker et al., 2003; Yu et al., 2007). Cleaning efficacy is assessed using Al/Ca, Mn/Ca and Fe/Ca ratios (e.g., Boyle, 1983; Barker et al., 2003). In some cases, a simple threshold value may be used

to screen samples, e.g., Al/Ca >80 µmol/mol (Mawbey and Lear, 2013). In many cases, the threshold depends on contaminant composition such that Mn/Mg or Fe/Mg ratios may be a more useful indicator (Barker et al., 2003). Cleaned foraminifera are commonly dissolved in acid and analysed by Inductively Coupled Plasma Mass Spectrometry (ICP-MS), taking into account the dependence of measured Mg/Ca on analyte concentration (the matrix effect – Lear et al., 2002), although other techniques may also be employed (see below). Prior to crushing, several representative specimens should be selected for SEM analysis to

record the extent of textural recrystallization on broken chamber walls. Sr/Ca values routinely collected alongside Mg/Ca data can provide one means of monitoring the impact of recrystallization on test geochemistry. Inorganic calcite tends to have lower Sr/Ca and higher Mg/Ca than foraminiferal calcite (Baker et al., 1982), which leads to inverse relationships in diagenetically altered downcore records.

*Recommended steps in converting of planktic Mg/Ca ratios to temperatures*:

i) The possible presence of dissolution should be determined. In the recent geological past, a dissolution correction can be based on the bottom water saturation state at the sample site (Dekens et al., 2002), but uncertainties in ocean carbonate chemistry currently preclude this approach in deep-time studies. We recommend that planktic foraminiferal test weights are

reported along with Mg/Ca data to assess the potential impact of dissolution (e.g., Rosenthal and Lohmann, 2002). Alternatively, chemically resistant domains within individual tests can be selected for analysis (see above).

ii) A salinity correction should be applied if there is independent evidence that the sample site experienced substantial deviations from normal salinity, such as the large changes in the hydrological cycle inferred for the PETM (Zachos et al., 2003). Normalising culture data of three modern species (compiled in Hönisch et al., 2013 and Allen et al., 2016) to the Mg/Ca observed at a salinity of 35 psu for each species results in the following multi-species salinity sensitivity Eq. (2):

$$\text{Mg/Ca}_{\text{CORRECTED}} = (1 - (\text{salinity} - 35) \times 0.042 \pm 0.008) \times \text{Mg/Ca}_{\text{MEASURED}} \qquad (2)$$

(see the Supplementary Information for further details). This sensitivity of $4.2 \pm 0.8\%$ per PSU is in good agreement with global sediment trap and plankton tow data for *G. ruber* (Gray et al., 2018).

iii) A correction for past changes in the carbonate system should be applied. This is complicated, however, because pH and $[CO_3^{2-}]$ are not only driven by long-term changes in the carbon cycle but also by factors such as the temperature effect on the dissociation constant of water ($K_W$). Therefore, whilst the results of Gray et al. (2018) indicate that the Mg/Ca-temperature sensitivity in the modern ocean is $6\%\,^\circ C^{-1}$ when the effects of pH and temperature are fully deconvolved, this can only be applied if the control of temperature on $K_W$ (and therefore pH) is accounted for (ideally through $\delta^{11}B$-derived pH reconstructions using the same material). For instance, if the pH reconstruction available for a given interval was determined at a different site, a temperature sensitivity of $6\%\,^\circ C^{-1}$ should be applied only if the difference in temperature between sites can be estimated, so that the temperature-driven inter-site pH gradient can be accounted for (pH differences between sites may also exist for other reasons). In practice, this requires that the equations for pH and temperature be solved iteratively, given that Mg/Ca is sensitive to both factors. We recommend differing approaches for Mg/Ca data treatment depending on whether a $\delta^{11}B$ pH record is available for the same site (see step (v) below).

Where $\delta^{11}B$-based pH reconstructions are available, Mg/Ca ratios should be corrected for pH. However, this correction should be considered with caution until the controlling carbonate system parameter on foraminifera Mg/Ca is identified, particularly given that pH and $[CO_3^{2-}]$ may be decoupled over geological time (Tyrrell and Zeebe, 2004). If no pH reconstruction is available for the site of interest, then the temporally and spatially closest data should be used and uncertainties in applying this considered. Based on a linear fit through culture data from three modern species (Evans et al., 2016a), the correction is:

$$\text{Mg/Ca}_{\text{CORRECTED}} = (1 - (8.05 - \text{pH}) \times 0.70 \pm 0.18) \times \text{Mg/Ca}_{\text{MEASURED}} \qquad (4)$$

Note that Mg/Ca may relate nonlinearly to pH outside the range 7.7-8.4, and more complex relationships have been suggested (Russell et al., 2004; Evans et al., 2016a). This sensitivity of -7.0 ±1.8% per 0.1 pH unit is in agreement with the -8.3±7.7% derived from a global sediment-trap and plankton tow *G. ruber* dataset (Gray et al., 2018), although we recommend applying the culture-derived expression in deep time because it is calibrated over a much wider pH range.

iv) A correction for $Mg/Ca_{sw}$ is usually applied to the pre-exponential component (B) of the relevant Mg/Ca-temperature calibration of the form $Mg/Ca = Bexp^{AT}$ (Hasiuk and Lohmann, 2010; Evans and Müller, 2012):

$$B_{CORRECTED} = (Mg/Ca_{sw}^{H} / 5.2^{H}) \times B_{MODERN} \qquad (3)$$

where $B_{MODERN}$ is the pre-exponential coefficient derived from modern calibrations, H is the nonlinearity of the relationship between shell and $Mg/Ca_{sw}$, which may be estimated from culture experiments under variable seawater chemistry (Evans et al., 2016b; Delaney et al., 1985), $Mg/Ca_{sw}$ is that of the time interval of interest, and 5.2 is the modern seawater Mg/Ca ratio in mol mol$^{-1}$. However, the observation that the slope of this relationship is sensitive to $Mg/Ca_{sw}$ in culture experiments means

15 that equations describing the change in both constants (A and B) have been reported for modern taxa (Evans et al., 2016b), with the implication that the use of modern calibrations may underestimate relative temperature changes during the early Paleogene. For estimates of early Paleogene $Mg/Ca_{sw}$ we recommend the use of a relatively high-precision, million-year (average) resolution $Mg/Ca_{sw}$ reconstruction from the coupled analysis of Mg/Ca and clumped isotopes in foraminifera (Evans et al., 2018a).

v) Mg/Ca is converted to temperature using an exponential calibration equation:

$$T = \ln (Mg/Ca_{CORRECTED} / B_{CORRECTED} ) / A \qquad (4)$$

25 Where A and $B_{CORRECTED}$ are derived from species or group-specific calibrations. The most appropriate calibration for extinct Eocene species should be chosen based on similarities to extant species in terms of shell chemistry or, for example, position within the water column and the presence or absence of symbionts. Best practice would report the sensitivity of a reconstruction to the choice of calibration equation. Importantly, the sensitivity factor A should be modified depending on how the carbonate system correction described above is performed. If pH is explicitly accounted for at the site of interest through $\delta^{11}B$ then the

30 6%°C$^{-1}$ sensitivity of Gray et al. (2018) should be applied. However, if the best available pH reconstruction is from a different site or time interval or derived from a model which represents the global mean (e.g. Tyrell and Zeebe, 2004), then we recommend applying the apparent sensitivity derived from culture (Kisakürek et al., 2008; Evans et al., 2016a) as this indirectly accounts for the effect of temperature on $K_W$.

These recommendations have been applied to the Mg/Ca analyses included in the DeepMIP database (Supplementary Data File 4). We have compiled Mg/Ca data for planktic foraminifera from five DSDP or ODP sites and six onshore sections. SST has been derived from Mg/Ca ratios as follows: 1000 random draws were performed of salinity (33–37 psu), seawater Mg/Ca (within the 95% CI given by Evans et al.,2018b), pH (+/-0.2 units), and the Mg/Ca-pH sensitivity (anywhere between 0-8.8% per 0.1 unit, i.e. anywhere between not sensitive at all to the upper confidence interval on the modern culture calibrations) for each data point. Calibration uncertainty was assessed by randomly choosing either the laboratory calibrations of Evans et al. (2016b), which define a $Mg/Ca_{sw}$-dependent Mg/Ca–T sensitivity, or the modern calibration with an 'H-factor' applied to the pre-exponential constant (Evans and Müller, 2012), which maintains the modern Mg/Ca–T sensitivity in deep time. The uncertainties on each data point are then the 97.5 and 2.5 percentiles of these 1000 sets of assumptions. The 'best estimate' SSTs are the 50 percentiles of the subset of these 1000 draws that use the calibrations of Evans et al. (2016b), which is preferred because the available evidence suggests that the $Mg/Ca_{sw}$-T sensitivity varies as a function of $Mg/Ca_{sw}$. Note that the data are subject to revision following replication of that study, and that using the 50th percentile of all estimates including both calibrations would result in overall cooler SST. Analytical uncertainty is not considered significant given the magnitude of these uncertainties.

***Intra-test Mg/Ca analysis by LA-ICP-MS and EPMA***: In contrast to solution Inductively-Coupled-Plasma Mass Spectrometry (ICPMS), which enables high-throughput of pooled, dissolved foraminifera, highly spatially-resolved techniques such as Laser-Ablation ICPMS facilitate targeted *in-situ* analysis of carbonates (Eggins et al., 2003) and allow intra-specimen preservation to be assessed (e.g. Creech et al., 2010; Evans et al., 2015). Small samples such as foraminifera may be analysed without embedding or sectioning, and high-sample through-put means that the technique is relatively inexpensive. Laser spot sizes are typically 20-80 μm in diameter, with 5 μm possible (Lazartigues et al., 2014), enabling repeat measurements of individual chambers or seasonality reconstruction in long-lived organisms with incremental growth layers (Bougeois et al., 2014; Evans et al., 2013). Because each laser pulse removes less than 100 nm of material on carbonates (Griffiths et al., 2013), depth-profiling through the sample has an effective resolution of <0.5 μm when using a fast wash-out ablation cell (Müller et al., 2009). Therefore, element profiles through foraminifera chamber walls not only facilitate the characterization of intra-specimen preservation, which can be assessed by the simultaneous collection of Mg/Ca, Sr/Ca, Mn/Ca, Al/Ca and Fe/Ca ratios, among others, but also allow diagenetically affected areas, such as surface overgrowths, to be excluded from the measurement used to calculate the oceanographic variable of interest (e.g. Hollis et al., 2015; Hines et al., 2017). The disadvantage of LA-ICPMS is that overall data acquisition and reduction is typically more time-consuming compared to solution-based techniques and there is relatively large intra-specimen variability.

Electron Probe Microanalysis (EPMA) is a microanalytical technique based on the detection of element-characteristic x-rays produced by bombarding the sample with an accelerated and focused electron beam. For quantitative analyses, the intensity of the element-specific x-rays is compared against those of the same elements from chemically well-characterized standards.

However, the sensitivity of EPMA is limited, and typically only Mg/Ca and Sr/Ca ratios in foraminiferal shells can be quantitatively measured (e.g., Brown and Elderfield, 1996; Anand and Elderfield, 2005). One advantage of this technique is the high spatial resolution; typical beam spot sizes for quantitative analyses in foraminiferal shells is ~2 to 10 µm (e.g., Hathorne et al., 2003). In addition, semi-quantitative elemental distribution maps can be acquired with sub-µm resolution for

a larger suite of elements (e.g., Pena et al., 2008) allowing for the identification of diagenetic phases.

Sample preparation for EPMA is time-consuming, as tests need to be embedded in epoxy and polished until suitable cross sections for analysis are exposed. This method is unique in allowing SEM imaging of chamber walls prior to analysis, including an assessment of contaminant and diagenetic phases. The same epoxy mounts can also be used for *in situ* $\delta^{18}$O analysis by

SIMS, so that chemical and isotopic information can be deduced from adjacent domains just microns apart (Kozdon et al., 2013; Wycech et al., 2018). Semi-quantitative maps of intrashell Mg/Ca variability with sub-µm resolution are possible with EPMA (e.g., Eggins et al., 2004), but for quantitative Mg/Ca measurements, an electron beam with a diameter of several µm is required, which averages between the high- and low Mg bands or domains observed in many planktic (e.g. Eggins et al., 2004; Spero et al., 2015) and benthic (e.g., de Nooijer et al., 2014) foraminifera. The average of multiple Mg/Ca measurements

by EPMA from an individual shell is typically comparable to solution-phase data that consume the whole shell (e.g., Hathorne et al., 2003). A major advantage of this method is that it is non-destructive.

### 4.3 Clumped isotopes

### 4.3.1 Theoretical background of clumped isotopes

The carbonate clumped isotope thermometer is based on the temperature-dependent proportion of $^{13}$C-$^{18}$O bonds in carbonate minerals (Ghosh et al., 2006, Eiler, 2007). The proxy has a direct basis in thermodynamics (Schauble et al., 2006; Hill et al., 2014) and has been applied to a wide range of marine and terrestrial samples, from foraminifera to paleosol carbonates (e.g. Tripati et al., 2010; Snell et al., 2013; Douglas et al., 2014). The zero-point energy of atomic bonds decreases with the mass of the atoms involved, favoring bonds between the rare, heavy isotopes. However, this effect decreases with increasing

temperature, leading to the theoretical and observed decrease of "clumping" of heavy isotopes with increasing formation temperature of the mineral (Eiler, 2007). Excess abundance of $^{13}$C-$^{18}$O bonds is abbreviated to $\Delta_{47}$ and refers to the over-abundance of $CO_2$ with the composition $^{13}$C-$^{18}$O-$^{16}$O relative to a stochastic distribution of all isotopes (Eiler and Schauble, 2004). $\Delta_{47}$ is measured on an isotope ratio mass spectrometer after acidification of the carbonate, in a very similar way as classical $\delta^{18}$O measurements. The only difference is that the abundance of mass 47 is recorded in addition to the traditional

masses 44-46; $\delta^{18}$O and $\delta^{13}$C are obtained as by-products of the measurements and are needed to calculate the expected stochastic abundance of $^{13}$C-$^{18}$O bonds in the respective sample, which is then compared to the observed abundance to calculate $\Delta_{47}$.

### 4.3.2 Strengths of clumped isotopes

There are three key strengths to this carbonate-based paleothermometer: (1) both theory and empirical studies demonstrate that the isotopic composition of water exerts no measurable control on the clumped isotope signature (Ghosh et al., 2006; Schauble et al., 2006); (2) the technique involves the simultaneous measurement of $\Delta_{47}$ and $\delta^{18}O$, enabling the independent and direct calculation of ancient $\delta^{18}O_{sw}$; (3) many, although not all, biogenic carbonates and inorganic precipitates fall on the same calibration line (e.g. Ghosh et al., 2006; Zaarur et al., 2013; Tang et al., 2014; Tripati et al., 2015). Molluscs and brachiopods (Came et al., 2007; Eagle et al., 2013a; Henkes et al., 2013), foraminifera (Tripati et al., 2010; Grauel et al., 2013; Evans et al., 2018a(Tripati et al., 2010; Grauel et al., 2013; Evans et al., 2018a; Peral et al., 2018; Piasecki et al., 2019), paleosols (Passey et al., 2010), land snails (Eagle et al., 2013b), and other forms of carbonate (Eiler, 2007; Kele et al., 2015) all appear to be reliable archives for the measurement of $\Delta_{47}$. Clumped isotope analysis of speleothems and certain coral species (Ghosh et al., 2006; Tripati et al., 2010; Saenger et al., 2012; Affek and Zaarur, 2014; Tripati et al., 2015; Loyd et al., 2016; Spooner et al., 2016) as well as other taxa (Davies and John, 2019) are more uncertain and require further study.

### 4.3.3 Weaknesses of clumped isotopes

Whilst clumped isotope thermometry has been successfully applied to a wide range of sample types, there are several challenges associated with paleoclimate reconstructions. Of these, the most fundamental is the low abundance of doubly substituted ("clumped") carbonate, which typically makes up only ~46 ppm of the total $CO_2$ produced from a sample. Precision is therefore ultimately limited by our ability to cleanly measure mass 47 $CO_2$. In practice, this implies relatively large sample masses, typically ~10 mg of material (>500 planktic foraminifera) although recent advances in instrumentation have seen this reduced by a factor of 5-10 in some laboratories (Meckler et al., 2014; Rablen et al., 2015; Müller et al., 2017). Analytical precision for these sample sizes limits the accuracy of the technique to ±2-3°C (1 sigma), which can be improved by performing a greater number of replicate measurements (e.g., Huntington et al., 2009; Thiagarajan et al., 2011; Tripati et al., 2014), with an obvious trade-off between sample size and precision. The presence of organic carbon, which can contribute to mass 47, may necessitate stringent sample cleaning procedures. In common with many proxies, the potential for seasonal growth of some archives must be considered. This is particularly the case for molluscs. Careful sample selection and geological context are of critical importance when designing studies and interpreting clumped isotope data.

As for $\delta^{18}O$ and Mg/Ca, preservation of foraminifera and other carbonates is a key issue that must be addressed. While the impacts of dissolution are not known, recrystallization at different temperatures or the addition of secondary diagenetic calcite precipitated after deposition can bias clumped isotope measurements of planktic foraminifera (Shenton et al., 2015; Stolper et al., 2018). However, recent work has shown that glassy foraminifera and some non-glassy planktic and benthic foraminifera appear to yield reliable clumped isotope data for paleoceanographic reconstructions (Leutert et al., 2019). Solid-state reordering within the calcite mineral will alter the isotope ordering, although only in samples that have experienced burial temperatures well above 100-150 °C for over ~10 Ma (Passey and Henkes, 2012; Henkes et al., 2014; Shenton et al., 2015).

Several empirical calibrations of $\Delta_{47}$ to temperature have been developed, with differences partly related to laboratory-specific analytical methodology; these include acid digestion temperature, instrumentation, ion corrections and standards. Some discrepancies observed in earlier studies may be due to the limited temperature range and replication when calibrating biogenic carbonates, whereas calibrations with larger temperature ranges (> 90 °C) across different types of carbonates agree well (Bonifacie et al., 2017; Fernandez et al., 2017; Kelson et al., 2017). Material-specific calibrations have also been suggested, for example, for marine molluscs (Eagle et al., 2013b; Henkes et al., 2013). Accurate absolute temperature reconstructions depend on empirical calibrations being developed or checked in each laboratory. Differences between laboratories have been attributed to a range of factors (Dennis et al., 2011; Wacker et al., 2014; Defliese et al., 2015; Daëron et al., 2016; Schauer et al., 2016) but are not yet fully resolved. With the use of rigorous standardization procedures to correct for instrument drift and more consistent methodologies overall, it is hoped that calibrations between instruments and labs will converge (e.g., Bernasconi et al., 2018).

### 4.3.4 Recommended methodologies for clumped isotopes

Here we outline our recommendations for generating clumped isotope data from fossil shells, and for converting these data into temperature. The current data set for the targeted age range is limited to five early Eocene onshore sections (Supplementary Data File 5) and paleosol carbonate data from four continental North American sites (Supplementary Data File 7A). Using the methods outlined below, we have calculated SSTs and error values and compiled a summary of SST estimates for these EECO sites.

We recommend the use of a calibration that covers a wide temperature range and sufficiently replicated analyses of calibration samples from the carbonate-type being analyzed, and performed with the same analytical and data processing procedures as employed for the samples (e.g., in the absolute reference frame as defined by Dennis et al. [2011] and done using either gas-standard or carbonate-standard based reference frames of Dennis et al. [2011]; or the carbonate standard-based reference frame of Bernasconi et al. [2018]). Where possible, we recommend that previously published calibrations be converted to these reference frames.

A recent meta-analysis of synthetic carbonate calibrations (Petersen et al., 2019) explores the causes of inter-laboratory offsets and makes recommendations on best practice. Best practice for measuring, correcting and reporting $\Delta_{47}$ includes: (1) measurement of a large number of gas and/or carbonate standards of different compositions; (2) correction for instrumental non-linearities (Huntington et al., 2009; Dennis et al., 2011), such as those which arise from secondary electrons (He et al., 2012; Bernasconi et al., 2013), or use of instruments with electron suppression; (3) reporting of data on an absolute reference frame (Dennis et al., 2011), and; (4) reporting full methodology, including digestion apparatus, digestion temperature, gas cleaning procedure, mass spectrometer and corrections used, working gas composition, constant sets used for calculations, acid digestion fractionation factor, $\Delta_{47}$ values and errors, temperature calibration, and estimated temperatures. Ideally, the

provision of gas and/or carbonate standard results together with the sample data facilitates the broad use of data and future recalculations. We recommend the open archiving of raw intensity values and datasets, with their own digital object identifiers, to enable the reanalysis of data over the long-term.

Given the required sample amounts and time-intensive measurements, clumped isotope thermometry is most powerful where other proxies are limited by unknown confounding effects. Rather than providing high-resolution reconstructions, clumped isotopes can be used to ground-truth and improve the accuracy of other proxies, including new constraints on seawater compositions (Evans et al., 2018a). Additionally, systems allowing the repeated measurement of small (~100–500 µg) sample aliquots give the required replication rates for $\Delta_{47}$, which are averaged across replicates or adjacent samples, and have the
potential to provide higher-resolution records of standard $\delta^{18}O$ and $\delta^{13}C$ analyses. As for $\delta^{18}O$ and Mg/Ca, samples need to be carefully screened for diagenetic alteration.

### 4.4 Isoprenoidal GDGTs (TEX$_{86}$)

### 4.4.1 Theoretical background of TEX$_{86}$

The tetraether index of tetraethers consisting of 86 carbon atoms (TEX$_{86}$) is an organic paleothermometer based upon the distribution of isoprenoidal glycerol dialkyl glycerol tetraethers (isoGDGTs) in marine or lake sediments (Schouten et al., 2002)., Within marine environments, isoGDGTs are inferred to be mainly derived from marine Thaumarchaeota (Schouten et al., 2002; Wuchter et al., 2004). Marine Thaumarchaeota occupy much of the epipelagic and mesopelagic zone, but cell numbers are highest in the upper few hundreds of meters of the surface ocean (Church et al., 2010), with TEX$_{86}$ correlating
most strongly with sea surface (SST) or shallow subsurface (Sub-T, 50-200 m) temperatures (Tierney and Tingley, 2015). The underlying principal of TEX$_{86}$ is that the number of cyclopentane rings (moieties) in GDGTs increases with growth temperature in order to alter the fluidity and permeability of the cell membrane (Sinninghe Damsté et al., 2012). Laboratory culture and mesocosm experiments confirm this relationship and indicate that TEX$_{86}$ values continue to increase with temperature above 30°C (Wuchter et al., 2004, Schouten et al., 2007; Pitcher et al., 2009; Kim et al., 2010), the upper limit of the modern coretop
dataset (Kim et al., 2010; Tierney and Tingley, 2015). Although these studies suggest that a linear relationship between TEX$_{86}$ and temperature persists at least to 40°C, the form of the relationship has been shown to vary significantly between different strains of Thaumarchaeota (Elling et al., 2015).

TEX$_{86}$ has been widely used to reconstruct SST during the Eocene (Brinkhuis et al., 2006; Pearson et al., 2007; Sangiorgi et
al., 2008; Bijl et al., 2009, 2010, 2013; Sluijs et al., 2011; Hollis et al., 2009, 2012; Douglas et al., 2014; Frieling et al., 2014; Inglis et al., 2015; Cramwinckel et al., 2018) and across the PETM (Sluijs et al., 2006, 2007a, b, 2011, 2014; Zachos et al., 2006; Schoon et al., 2015; Frieling et al., 2017; 2018).

### 4.4.2 Strengths of TEX$_{86}$

TEX$_{86}$ has several key advantages over other paleothermometers. Isoprenoidal GDGTs are relatively resistant to thermal degradation and diagenesis (Schouten et al., 2004) and can be used to reconstruct SST in settings where well-preserved foraminiferal calcite is absent (e.g. Bijl et al, 2009; Hollis et al., 2012; Pross et al., 2012) and where alkenones (e.g. the U$^{K'}_{37}$ paleothermometer) are either absent or outside their calibration range (Brassell, 2014). Furthermore, TEX$_{86}$ is not affected by

changes in pH and salinity (Wuchter et al., 2004; Elling et al., 2015) and does not require a correction to account for past changes in ocean seawater chemistry (see Sections 5.1.3 and 5.1.4). TEX$_{86}$ values are also not substantially impacted by sediment extraction and processing techniques and interlaboratory variation in TEX$_{86}$ values are comparable to those of other temperature proxies (Schouten et al., 2013a).

**4.4.3 Weaknesses of TEX$_{86}$**

The interpretation of TEX$_{86}$ values can be complicated by the input of GDGTs from archaea other than marine Thaumarchaeota. Several indices have been developed to screen for such secondary inputs. The branched-to-isoprenoidal tetraether (BIT) index (Hopmans et al., 2004) is used to assess the impact of terrestrial input upon TEX$_{86}$ values. TEX$_{86}$-derived SSTs may be warm-biased by 2°C or more when BIT indices exceed 0.4 (Weijers et al., 2006). However, the impact of

terrigenous GDGTs is highly variable, being dependent on the nature and temperature of the source catchment (Bijl et al., 2013; Douglas et al., 2014; Inglis et al., 2015). Sedimentary GDGT production can also affect TEX$_{86}$ values. Methanotrophic Euryarchaeota can synthesise GDGTs with up to 3 cyclopentane moieties (Pancost et al., 2001) and may impact TEX$_{86}$ values in settings in which anaerobic oxidation of methane (AOM) has occurred. The Methane Index (MI) can be used to assess AOM, whereby high MIs (>0.5) reflect high rates of AOM and low values (<0.3) suggest normal sedimentary conditions

(Zhang et al., 2011). During the Eocene, MI values rarely exceed 0.5 suggesting that most settings are relatively unaffected by diffusive methane flux and associated AOM. Methanogenic archaea can synthesize GDGT-0 as well as smaller quantities of GDGT-1–3 and may bias TEX$_{86}$ values (Blaga et al., 2009). This can be assessed using the %GDGT-0 index (Inglis et al., 2015). Culturing experiments indicate that a methanogenic source of GDGT-0 is possible when the index is >67%. Methanogenesis appears to have a minor influence on Eocene records as >90% of analyses are below this threshold (Inglis et

al., 2015).

One of the major uncertainties in TEX$_{86}$ palaeothermometry is the impact of non-temperature factors upon thaumarchaeotal GDGT distributions, including growth phase (Elling et al., 2014; Hurley et al., 2016) and oxygen concentrations (Qin et al., 2015). Although these factors remain difficult to assess in the geological record, the Ring Index (Zhang et al., 2016) and

fCren′:Cren′ + Cren (O'Brien et al., 2017) can potentially help to reveal non-temperature controls, or exclude the influence of such biases. There is also evidence that sedimentary GDGTs are not exported solely from surface waters but also shallow subsurface waters (Taylor et al., 2013) and that their export may be seasonally biased (Herfort et al., 2006; Bijl et al., 2010). These factors may help to explain some apparent SST anomalies (e.g. Taylor et al., 2018) and discrepancies with other proxies (Eberle et al., 2010; Hollis et al., 2012; Inglis et al., 2015). The suggestion that these discrepancies may be explained by export

of GDGTs from the deep ocean (Ho and Laepple, 2016) was refuted by Tierney et al. (2017) who argue that the sedimentary TEX$_{86}$ signal is predominantly derived from the upper water column (50–300 m) and that temperatures through the upper water column are strongly correlated, therefore minimizing any calibration bias. Export from the upper water column rather than sea surface will impart some uncertainty because TEX$_{86}$ is calibrated to climatological ocean temperatures, such that the local

dominant depth or season of production and export could introduce bias, particularly in settings with relatively deep, sub-thermocline GDGT production (Taylor et al., 2013) or in middle to high latitudes (Hollis et al., 2012). Future calibration studies should provide detailed information about the prevailing oceanographic conditions at each core-top location. In areas with a shallow and steep thermocline, TEX$_{86}$ may be recording thermocline rather than surface variations (e.g., Huguet et al., 2007). To address this, both SST and sub-T calibrations have been developed (e.g. Schouten et al., 2002; Kim et al., 2008; Tierney

and Tingley, 2015), and should be applied with knowledge of the oceanographic characteristics of the site in question.

A final weakness is that many early Eocene sites (e.g. Frieling et al., 2017, 2018; Cramwinckel et al., 2018) are characterized by TEX$_{86}$ values that exceed the calibration range in modern oceans (0.3–0.8). Whilst some uncertainty may be alleviated by comparison with associated multi-proxy data (e.g. Frieling et al., 2017; Tierney et al., 2017), TEX$_{86}$ values >0.8 require

extrapolation of the relationship between TEX$_{86}$ and temperature beyond the modern calibration dataset or incorporation of limited data from laboratory studies (e.g., Pitcher et al., 2009; Elling et al., 2015). Such high values almost certainly indicate hotter temperatures, but the degree of estimated warming depends on assumptions about the mathematical nature of the temperature-TEX$_{86}$ relationship (see below and discussion in Eley et al, in review).

**4.4.4 Recommended methodologies for TEX$_{86}$**

Here we outline our recommendations for generating TEX$_{86}$ data from marine sediments, and for converting these data into temperature. We have compiled available TEX$_{86}$ data from five ODP and IODP sites, five onshore drillholes and nine onshore sections (Supplementary Data File 6). Using the methods outlined below, we have calculated SSTs and associated uncertainties, and compiled a summary of SST estimates for LP, PETM and EECO time slices.

Preparation of sediments for TEX$_{86}$ analysis is straightforward, although large sample volumes (100g) may be needed where concentrations of organic matter are low. After extraction and purification, the isoprenoidal GDGTs are analysed by HPLC/MS (see Hopmans et al., 2000). Previously, a Cyano column was used for HPLC separation (Hopmans et al., 2000); however, the use of BEH HILIC UHPLC columns has now been shown to yield better separation while not substantially affecting the TEX$_{86}$

index (Hopmans et al., 2016). The TEX$_{86}$ index is calculated as the ratio between isoprenoidal GDGTs as follows:

(GDGT-2 + GDGT-3 + cren')/ (GDGT-1+ GDGT-2 + GDGT-3 + cren')                            (5)

where GDGT-1, GDGT-2 and -3 are characterized by one, two and three cyclopentane moieties and cren' is the regioisomer of crenarchaeol.

Several TEX$_{86}$ calibrations have been developed to estimate sea surface or shallow subsurface temperatures. The original TEX$_{86}$ core-top calibration was a linear relationship to SST (Schouten et al., 2002). However, the correlation between TEX$_{86}$ and SST is weak below 5°C and samples from the Red Sea deviate from the wider calibration dataset (Kim et al., 2008). To resolve this, TEX$_{86}$ was recalibrated to exclude samples from the Red Sea and sites where SST < 15 °C (e.g. Kim et al., 2008; Naafs and Pancost, 2016; O'Brien et al., 2017).

Various non-linear relationships have also been proposed (e.g. Liu et al., 2009; Kim et al., 2010). Of these non-linear calibrations, the most commonly used is TEX$_{86}$$^H$, which assumes an exponential relationship between temperature and GDGT distributions. It uses a log transformation of TEX$_{86}$ and excludes core-top data from the Red Sea as well as from sites where SST < 5 °C. SST is derived from TEX$_{86}$$^H$ as follows:

$$TEX_{86}^{H} = \log_{10}(GDGT\text{-}2 + GDGT\text{-}3 + cren') / (GDGT\text{-}1 + GDGT\text{-}2 + GDGT\text{-}3 + cren') \qquad (6)$$

$$SST = 68.4 \times TEX_{86}^{H} + 38.6 \; (RMSE = 2.5 \; °C) \qquad (7)$$

However, the TEX$_{86}$$^H$ calibration does not account for observational error and is affected by regression dilution because TEX$_{86}$ is treated as the independent variable and therefore assumed to be error-free. The regression dilution causes it to systematically underestimate warm SSTs in the modern ocean and, by inference, may also underestimate SSTs in ancient greenhouse climates (Tierney and Tingley, 2014; O'Brien et al., 2017).

Kim et al. (2010) also proposed another index, TEX$_{86}$$^L$, which is also an exponential calibration but uses the entire core-top dataset (with the exception of the Red Sea) and was developed to reconstruct SST across all temperature ranges. However, TEX$_{86}$$^L$ does not strictly reflect the degree of cyclisation in GDGTs and consequently lacks obvious biological rationale. It is also particularly sensitive to contributions from other archaea, especially those living in the subsurface (Kim et al., 2015; Taylor et al., 2013) and its application within the geological record has been questioned (Inglis et al., 2015; Taylor et al., 2018).

Available experimental evidence (Pitcher et al., 2009; Schouten et al., 2013b; Elling et al., 2015) suggests a linear relationship between TEX$_{86}$ and SST, which is consistent with the modern core-top dataset. However, it remains uncertain if TEX$_{86}$ continues to have a linear relationship with temperature beyond the range of the modern calibration (Cramwinckel et al., 2018). The functional form of the calibration has significant consequences for temperature estimation, especially for values >0.8 (Fig. 4A), which is particularly relevant for the warm Paleogene where such values are common.

The Bayesian regression model (BAYSPAR) of Tierney and Tingley (2014, 2015) applies a linear calibration, but the regression terms can vary spatially to accommodate the modern regional variation in $TEX_{86}$ sensitivity, as in the Mediterranean Sea (Kim et al., 2015). In deep-time settings, BAYSPAR searches the modern core-top dataset for $TEX_{86}$ values that are similar to the measured $TEX_{86}$ value within a user-specified tolerance and draws regression parameters from these modern locations. This assumes that modern $TEX_{86}$ values yield information on the environmental factors that determine the $TEX_{86}$-SST relationship in the ancient oceans, including factors other than mean annual sea surface temperature that might regionally alter the sensitivity of the proxy. This assumption is difficult to test and has potential to introduce additional uncertainties or bias.

BAYSPAR can be determined without excluding core-tops from the Red Sea or where SSTs < 15 °C. This is likely important as Red Sea-like conditions cannot be precluded for many Eocene sites (Inglis et al., 2015). The Bayesian model can also be used to estimate uncertainty beyond the modern calibration range. BAYSPAR can be used by downloading the publicly-available code in Matlab (https://github.com/jesstierney/BAYSPAR) or Python (https://github.com/brews/baysparpy). Both surface ($BAYSPAR_{SST}$) and subsurface ($BAYSPAR_{SubT}$) calibrations are available, the latter being the gamma-weighted average of the temperature range over 0–200 m water depth, with a maximum probability at ~50 m (Tierney and Tingley, 2015).

When applied to $TEX_{86}$ values that exceed the modern calibration range, BAYSPAR typically yields results that lie between the linear and exponential calibrations because it incorporates both "normal tropical marine" and "Red Sea" sensitivities, the latter of which involves a steeper $TEX_{86}$-SST slope that is closer to the exponential assumption (Fig. 4b). It is thus a good "middle ground" choice for the Paleogene data considered here, and so we have used $BAYSPAR_{SST}$ in our compilations (Fig. 8, Supplementary Data File 6). While an exponential model might be a viable alternative, we do not recommend application of the $TEX_{86}^{H}$ calibration because of the known bias from regression dilution.

Given the ongoing evolution of calibrations, we recommend that researchers include the fractional abundances of all isoprenoidal GDGTs as well as $TEX_{86}$ values in all future publications as these will facilitate easy recalculation if and when the functional form of the calibration becomes better constrained. If BAYSPAR is used, we recommend that researchers report the parameters used (the prior mean and standard deviation, the search tolerance, and the version of the calibration dataset) so that the results are reproducible. The issue of extrapolation beyond the temperature range of the modern GDGT calibration dataset is highlighted in Eley et al. (in review), which also proposes an alternative calibration methodology.

### 4.5. Alkenone-based paleothermometry

Although there are no published SST records from the DeepMIP time slices based on alkenone paleothermometry, here we review the potential for new records based on this methodology and give a brief guide to their potential use. The alkenone SST

proxy is based on the temperature-dependence of the relative unsaturation of $C_{37}$ ketones (alkenones), synthesized as energy storage molecules by the Isochrysidales clade of haptophyte algae (Brassell et al., 1986; Marlowe et al., 1984; Volkman et al., 1980; and see Section 6.2 below). For the purposes of the SST proxy, the degree of unsaturation is typically quantified in the $U_{37}^{K'}$ index, defined as:

$$U_{37}^{K'} = \frac{C_{37:2}}{C_{37:2} + C_{37:3}}$$

where $C_{37:2}$ and $C_{37:3}$ are respectively, the di- and tri-unsaturated $C_{37}$ alkenones (Prahl and Wakeham, 1987). In both culture (Prahl et al. 1998) and core-top studies (Müller et al. 1998) there is a strong correlation between culture growth temperature
or mean annual SSTs (core-tops), and the $U_{37}^{K'}$ index. The temperature- $U_{37}^{K'}$ relationship is well-modelled by a linear relationship over most of the calibration range, at least between temperatures of ~3ºC to ~24ºC (Tierney & Tingley, 2018). At high-temperatures, the production of triunsaturated alkenone is increasingly rare, such that the $U_{37}^{K'}$ index approaches unity at temperatures above ~27ºC (Tierney & Tingley, 2018).

In the fossil record, alkenones characteristic of the Isochrysidales are absent prior to the early Eocene (Brassell, 2014), and within lower Eocene sediments are only reported from three locations, the New Jersey Margin ODP Site 612, the Cape Verde Basin DSDP Site 367, and the São Paulo Plateau DSDP Site 356 (Dzvonik [1996] cited in Brassell [2014]), all in the later stages of the EECO (Brassell 2014). At these locations, only the diunsaturated $C_{37:2}$ alkenone is present (Brassell, 2014), indicating production temperatures in excess of ~27ºC (Tierney & Tingley, 2018). Alkenones are reported more consistently,
and from more locations, within middle Eocene successions including the first appearance of triunsaturated alkenones (Brassell, 2014). Although this shift in the composition of alkenone assemblages may be due to the evolution of alkenone-production coupled to the early Eocene origin and early evolution of the major calcifying clade within the Isochrysidales (the Noelaerhabdaceae) (Brassell, 2014; Agnini et al., 2014), it is also consistent with the temperature-control on alkenone unsaturation and the progressive cooling of global SSTs from the early to middle Eocene (Brassell, 2014; Inglis et al., 2015;
Cramwinckle et al., 2018).

Although not included within the current DeepMIP compilation, a reexamination of the stratigraphic age and composition of reported early Eocene alkenones (Dzvonik, 1996) is warranted, especially if new occurrences are documented from additional locations. In this context we would recommend temperature constraints based on the new BAYSPLINE calibration, which
explicitly addresses the reduced temperature sensitivity of $U_{37}^{K'}$ above 24ºC, the high-temperature range most likely relevant to EECO alkenone assemblages (Brassell, 2014). Based on the absence of triunsaturated compounds from the reported

occurrences of early Eocene alkenones (Brassell, 2014), these would infer a predicted SST distribution with a lower limit on the 90% confidence interval of 27°C (Tierney & Tingley, 2018).

## 5 Terrestrial proxies for air temperature

Fewer geochemical options are available for reconstructing temperatures on the land (i.e., within the airmass overlying areas of land) than for reconstructing sea temperatures. In this section, we outline the primary geochemical approach, which is based on the distribution of branched tetraethers in sediments, in addition to two fossil-based approaches: physiognomic analysis of leaf fossils (including Leaf Margin Analysis and CLAMP) and paleobiogeographic analysis of pollen assemblages (NLR, Nearest Living Relative approach).

Stable isotopes have also been used to reconstruct early Paleogene terrestrial temperatures, notably $\delta^{18}O$ of mammalian teeth (Fricke and Wing, 2004) and pedogenic carbonate (Hyland and Sheldon, 2013; Hyland et al., 2017). Temperature proxies have also been developed from elemental ratios with paleosols. Termed climofunctions, the salinization index (SAL; Sheldon et al., 2002) and the paleosol weathering index (PWI; Gallagher and Sheldon, 2013) have been used to estimate temperature trends

in early Eocene paleosols from North and South America (Hyland and Sheldon, 2013; Hyland et al., 2013; 2017). Clumped isotopes have also been applied to pedogenic carbonate and show great promise as an effective means to reconstruct seasonal temperatures (Snell et al., 2013). These proxies are not discussed here but the temperature records are included in the DeepMIP database. We adopt the term Land Air Temperature (LAT) for what is otherwise referred to as Mean Air Temperature (MAT) or Mean Annual Air Temperature (MAAT).

We have compiled terrestrial proxy data from 80 sites, which include four ODP or IODP sites and 76 onshore localities (Supplementary Data Files 7 and 7a). A wide range of proxies provide over 100 LAT estimates from over 400samples. Only the most widely used of these are described below.

**5.1 Branched tetraether (brGDGT) paleothermometry**

**5.1.1 Theoretical background of branched tetraether (brGDGT) paleothermometry and early work**

Branched glycerol dialkyl glycerol tetraethers (brGDGTs) are membrane-spanning lipids produced by bacteria, probably acidobacteria (Weijers et al., 2009; Sinninghe Damsté et al., 2011, 2018). First discovered in a Dutch peat deposit (Sinninghe Damsté et al., 2000), brGDGTs are ubiquitous in terrestrial settings (Schouten et al., 2013b). These compounds have been used

increasingly as paleothermometers since a study of a global suite of mineral soil samples (Weijers et al., 2007) found that the degree of methylation in brGDGTs (MBT index) is correlated to LAT and pH, and that the number of cyclopentane rings in the same brGDGTs (CBT index) is related to soil pH, resulting in the MBT–CBT paleotemperature proxy. The proxy has been subsequently revised and recalibrated as MBT'–CBT (Peterse et al., 2012). The following three equations are used to derive LAT from the MBT'–CBT proxy [Eqs. 8–10].

$$MBT' = \frac{Ia+Ib+Ic}{Ia+Ib+Ic+IIa+IIa'+IIb+IIb'+IIc+IIc'+IIIa+IIIa'} \tag{8}$$

$$CBT = -\log\left(\frac{Ib+IIb+IIb'}{Ia+IIa+IIa'}\right) \tag{9}$$

$$MAT\ (°C) = 0.81 - 5.67 \times CBT + 31.0 \times MBT' \quad (n=176, R^2=0.59, RSME=5.0\ °C) \tag{10}$$

The MBT'–CBT proxy has been used to reconstruct early Paleogene terrestrial temperatures from continental margin marine records from the Southern Hemisphere (Bijl et al., 2013; Pancost et al., 2013; Pross et al., 2012) and the PETM in the Northern Hemisphere (Schoon et al., 2015; Weijers et al., 2007). These studies explicitly assumed that the majority of brGDGTs in marine sediment cores are produced within mineral soils at around sea level close to the core site and transported without

alteration into marine sediments of similar age. The resulting LATs are significantly higher than present day. However, despite being in good agreement with other terrestrial proxies for LAT as well as with modelled LAT (Huber and Caballero, 2011), they are ~5–10 °C lower than SST estimates derived from the same samples. This anomaly is discussed further below.

### 5.1.2 Strengths of brGDGT paleothermometry

Terrestrial temperatures are a crucial climatic variable, but most of our temperature records during the geological past are from the marine realm. The main strength of the MBT'–CBT terrestrial temperature proxy is that it is one of the few geochemical proxies currently available that can provide high-resolution quantitative estimates of terrestrial temperatures during the Cenozoic. In addition, brGDGTs are ubiquitous in ancient soils (e.g. Chinese loess), peats and lignites from across the Cenozoic (Peterse et al., 2012; Naafs et al., 2018a, b), making it possible to apply this proxy across a range of archives and depositional

environments.

### 5.1.3 New developments in brGDGT paleothermometry

Recent studies suggest that caution should be applied to the interpretation of existing MBT'–CBT data. De Jonge et al., (2013; 2014) used refined analytical techniques to show that the 5-methyl penta- and hexamethylated brGDGTs, which were used to

calculate the original CBT and MBT' indices, coelute with newly identified 6-methyl brGDGTs that have a strong pH dependence. Re-evaluation of the global mineral soil dataset has led to new indices and calibrations based on 5-methyl brGDGTs only, which are dependent on temperature alone (De Jonge et al., 2014).

$$MBT'_{5ME} = \frac{(Ia+Ib+Ic)}{(Ia+Ib+Ic+IIa+IIb+IIc+IIIa)} \tag{11}$$

$$MAT = -8.57 + 31.45 \times MBT'_{5ME} \quad (n=231, R^2=0.64, RMSE=4.9\ ^oC) \tag{12}$$

As well as a multiple linear regression (MAT$_{mr}$) calibration:

$$MAT_{mr} = 7.17 + 17.1 \times \{Ia\} + 25.9 \times \{Ib\} + 34.4 \times \{Ic\} - 28.6 \times \{IIa\} \tag{13}$$

$$(n = 231, R\textasciicircum2 = 0.67, RMSE = 4.7°C)$$

Naafs et al. (2017a) extended the calibration dataset and developed a new calibration for LAT as well as one to reconstruct Growing Degree Days above freezing (GDD$_0$), a measure of cumulative temperature over the growing season. GDD$_0$ is a guide to warm season temperatures and helps to constrain seasonal temperature variations, which is particularly important in high latitudes with high seasonality.

$$MAAT_{soil} \, (^oC) = 40.01 \times MBT'_{5me} - 15.25 \; (n = 350, R^2 = 0.60, RMSE = 5.3 \, ^oC) \tag{14}$$

$$GDD_{0\,soil} = 14344.3 \times MBT'_{5me} - 4997.5 \; (n = 350, R^2 = 0.63, RMSE = 1779) \tag{15}$$

### 5.1.4 Weaknesses of brGDGT paleothermometry

Recent studies have identified challenges in applying brGDGT paleothermometry in marine settings. The assumption that
brGDGTs in marine sediments are mainly derived from mineral soils formed at sea level may not be valid in some cases. Recent work has shown that brGDGTs are also produced *in situ* in the marine water column and in the hosting marine sediments (Sinninghe Damsté, 2016; Weijers et al., 2014), in rivers (Zell et al., 2014), and in lakes (Naeher et al., 2014; Weber et al., 2015). To illustrate this, we plot the relative abundance of the tetra-, penta-, and hexamethylated brGDGTs of the published early Paleogene data from five sites together with the modern mineral soil and peat data (Naafs et al., 2017a, b) and early
Paleogene data from lignites (Naafs et al., 2018a, b), following the approach of (Sinninghe Damsté, 2016) (Fig. 5a). The result demonstrates that most of the marine samples analyzed in this dataset have brGDGT distributions that are different from those found in any modern mineral soil or peat with a higher relative contribution of hexamethylated brGDGTs. This difference in brGDGT distribution suggests that these brGDGTs may not originate exclusively from mineral soils or peat. However, it is worth noting that the LATs derived from MBT'/CBT from some of these sites are in close agreement with LATs derived from
pollen assemblages using the Nearest Living Relative methods (Pross et al., 2012; Pancost et al., 2013).

Even when derived from mineral soils, the brGDGT assemblage in marine sediment will reflect varying combinations of settings across an entire terrestrial catchment, including different elevations and microclimates (Bendle et al., 2010). Moreover, brGDGT distributions have also been found to be influenced by soil moisture content in some settings, which needs to be
considered in paleoclimate reconstruction, especially in arid regions (Dang et al., 2016; Menges et al., 2014).

Although brGDGTs appear resistant to initial degradation in the water column, thermal maturation of brGDGTs on geological time scales does influence the proxy. Schouten et al. (2013b) used artificial maturation experiments to demonstrate that the degree of cyclization of brGDGTs (e.g. CBT-indices) changes at temperatures above 240 °C and brGDGTs are completed

degraded at 300 °C. However, the influence of thermal maturity on the degree of methylation (MBT indices) appears to be limited as long as brGDGTs are not fully degraded.

A further weakness of this proxy is that it fails to capture the super-warm temperatures of the early Paleogene because the calibrations saturate at ~25°C (e.g., Peterse et al., 2012; Naafs et al., 2017a). Many of the proxies for terrestrial temperature discussed here suffer from this problem because the modern environments that form the analogues for the calibration do not extend into tropical settings >25°C. For this reason, data sets should be scrutinized for evidence of LAT plateaus, which suggest the calibration has saturated.

Because the co-elution of newly identified 6-methyl brGDGTs may compromise MBT'–CBT estimates for LAT, earlier datasets should be re-analyzed using the latest analytical methods and calibrations. This problem may not be widespread because the brGDGT distributions reported in several Paleogene studies (e.g. Bijl et al., 2013) are dominated by tetramethylated brGDGTs, which have no 6-methyl homologues.

### 5.1.5 Recommended methodology for brGDGTs

If the brGDGT-proxy is to be applied to reconstruct terrestrial temperatures during the Paleogene the latest set of calibrations (De Jonge et al., 2014; Naafs et al., 2017a) and analytical techniques that separate the 5- and 6-methyl brGDGTs should be used (e.g., Hopmans et al., 2016). In addition, if applied to marine sediments to reconstruct terrestrial temperatures, supporting evidence should be provided (Fig. 5) to demonstrate a dominant input of brGDGTs from mineral soils or peat.

Another option is to avoid the source uncertainty associated with analyses from marine sediments by directly analyzing terrestrial sediments. For example, Inglis et al. (2017) used the latest analytical techniques and mineral soil calibrations to reconstruct early Eocene terrestrial temperatures from lignites in northern Germany. Although determining accurate ages for lignites can be challenging, these systems are dominated by *in situ* production of brGDGTs, resulting in reconstructed temperatures with a higher fidelity to local LAT.

Some studies have questioned whether soil-based calibrations should be applied to peat and lignite (Huguet et al., 2013, 2014; Weijers et al., 2011). Consequently, Naafs et al. (2017b) recently developed new peat-specific brGDGT-based temperature and pH calibrations, following Eqs. (16–19)

$$MBT'_{5ME} = \frac{(Ia+Ib+Ic)}{(Ia+Ib+Ic+IIa+IIb+IIc+IIIa)} \tag{16}$$

$$LAT_{peat} \ (°C) = 52.18 \times MBT'_{5me} - 23.05 \qquad (RMSE = 4.7 \ °C) \tag{17}$$

$$CBT_{peat} = \log\left(\frac{Ib+IIa'+IIb+IIb'+IIIa'}{Ia+IIa+IIIa}\right) \tag{18}$$

$$pH = 2.49 \times CBT_{peat} + 8.07 \qquad (RMSE = 0.8) \qquad\qquad (19)$$

In addition, the relative abundance of H-brGDGTs (brGDGTs with an extra covalent bond between the two alkyl chains) and the degree of methylation of H-brGDGTs is also correlated with LAT (Naafs et al., 2018b). Although the correlation is not

strong enough to provide quantitative temperature estimates, a high abundance of H-GDGTs in ancient lignites could provide additional evidence for elevated terrestrial temperatures. Applying the peat-specific brGDGT temperature calibration to the lignite record from North Germany confirms that LAT was significantly higher than present day, and 2-3 °C higher than those obtained with the mineral soil calibrations (Naafs et al., 2018a). These peat-specific calibrations were recently applied to a range of early Paleogene lignites from across the globe and indicate that LAT were significantly higher than previously thought

and are more consistent with the marine-based temperature estimates (Naafs et al., 2018a).

In summary, the application of peat-specific brGDGT-based temperature calibrations to early Paleogene lignites is in the early stages of development but holds great potential to provide new insights into terrestrial temperatures during this key period of the Cenozoic (Naafs et al., 2018a). MBT-based approaches are still judged useful to provide temperature constraints, especially

for multi-proxy comparisons in well-dated marine sediments but must be carefully assessed for calibration saturation at high temperatures, sampling of wide catchment areas, and inputs of brGDGTs that are not sourced from non-mineral soils or peat.

**5.2 Leaf morphology-based approaches to estimating paleotemperature**
**5.2.1 Theoretical background of leaf morphology-based approaches**

Methods of estimating paleotemperatures and other climate parameters from the physiognomy of fossil leaves have been explored since the early 1900s (Bailey and Sinnott, 1915, 1916). They are based on the general premise that plants adapt their physical features to tolerate the climate in which they germinate due to their immobility. Early observations of Southeast Asian floras identified a positive correlation between temperature and the percentage of woody dicotyledonous (dicot) species that had smooth-margined leaves (Bailey and Sinnott, 1915). This correlation was later developed into the widely-applied method,

Leaf Margin Analysis (LMA; Wolfe 1979; Wing and Greenwood, 1993). Numerous modifications, different calibration data sets and measures of uncertainty related to LMA have since been explored (e.g. Wilf, 1997; Greenwood et al., 2003; Kowalski and Dilcher, 2003; Greenwood et al., 2004; Hinojosa and Villagran, 2005; Miller et al., 2006; Spicer et al., 2011). Essentially, LMA involves assessment of the margin character (whether smooth [entire] or toothed [non-entire]) of all dicot leaf morphotypes in a fossil assemblage. The percentage of entire margins is calculated, and temperature determined using an LMA

calibration equation.

Several shortcomings were recognized quite early in the development of the LMA method. For example, the relationship between the percentage of entire margins and temperature was found to vary between geographic regions with no robust means of ascertaining which modern calibration should be used for a fossil leaf flora. In response to these shortcomings and in

recognition that other leaf characters may be correlated with climate variables, multivariate methods were explored. CLAMP (Climate Leaf Analysis Multivariate Program; Wolfe, 1993; Spicer et al., 2009; Yang et al., 2011, 2015) is the most widely used of these methods. In this method, 31 leaf character variables are measured for each fossil leaf morphotype and estimates of several climate parameters are produced, including temperature, precipitation and humidity. CLAMP relies on comparison

of fossil leaf data with large calibration datasets of modern dicot leaf morphology data collected using rigid protocols. The initial calibration dataset (Wolfe, 1993) was dominated by Northern Hemisphere assemblages. Subsequently, calibrations from other geographic areas have been developed (Steart et al., 2010; Jacques et al., 2011; Kennedy et al., 2014; Khan et al., 2014; Yang et al., 2015). Corresponding climate data sets are either based on the closest available long-term climate station measured data, or model-generated gridded data sets (Spicer et al., 2009). Other multivariate methods, such as the Nearest Neighbor

Approach (Stranks and England, 1997) and Canonical Correspondence Analysis (Hinojosa and Villagran, 2005; Hinojosa et al., 2006) also use the CLAMP datasets. More information on calibration updates on datasets and methodology, and an online analysis are available on the CLAMP website (http://clamp.ibcas.ac.cn/).

Another multivariate method, digital leaf physiognomy (DiLP), applies multivariate linear regression models based on 92

calibration sites to fossil leaf morphology data captured digitally (Huff et al., 2003; Royer et al., 2005; Peppe et al., 2011). In addition to these leaf morphology-based methods, leaf size has also been used as a univariate climate proxy, particularly for precipitation (Dolph and Dilcher, 1980; Wilf et al., 1998; Wiemann et al., 1998; Carpenter et al., 2012).

### 5.2.2 Strengths of leaf morphology-based approaches

The strengths and weaknesses of leaf morphology-based approaches to temperature reconstruction have been discussed extensively (e.g. Wolfe, 1993; Wilf, 1997; Uhl et al., 2003; Spicer et al., 2004; Traiser et al., 2005; Spicer et al., 2005; Green, 2006; Greenwood, 2007; Yang et al., 2007; Spicer and Yang, 2010; Jaques et al., 2011; Peppe et al., 2011; Spicer et al., 2011; Kennedy et al., 2014; Khan et al., 2014; Yang et al., 2015, Greenwood et al., 2017; Lowe et al., 2018). A key strength is that they provide quantitative estimates that do not depend on correct assignment of fossil leaf forms to modern taxa and their

current climate affiliations (e.g. Uhl et al., 2003). Each leaf morphology-based method also has its own particular strengths. Univariate methods such as LMA are relatively easy to apply (e.g. Yang et al., 2015) whereas a strength of a multivariate method like CLAMP is that the inclusion of several leaf morphology characters alleviates some of the problems associated with taphonomic loss of character information in fossils. An advantage of DiLP is the minimization of any subjectivity or ambiguity in the leaf character scoring process because computer algorithms rather than individuals process the measurements

(Peppe et al., 2011).

### 5.2.3 Weaknesses of leaf morphology-based approaches

A number of assumptions are made when applying leaf morphology-based paleoclimate methods, including that calibrations between leaf morphology and climate parameters are stable back through time and that fossil leaf collections are sufficiently

representative of the source vegetation. Each method has its own set of weaknesses, many of which are discussed in the references given in section 5.2.2. Univariate methods such as LMA are potentially strongly affected by taphonomic biases because they are reliant on the representative preservation of a single leaf character. LMA is also strongly influenced by geographic variability, with vegetation from different parts of the world yielding significantly different LMA calibrations.

CLAMP is less influenced by these weaknesses but is more complicated and time-consuming to apply. As with the MBT'–CBT proxy, the calibration for leaf morphology proxies saturates at ~25°C for most calibrations, perhaps at ~28°C for CLAMP (Utescher et al., 2014). An exception is LMA using the Kowalski and Dilcher (2003) calibration, where LATs approach 30°C. This calibration endeavours to account for a cool bias in many other approaches, which is inferred to be due to the over-representation of toothed leaves in the wet environments that typify most settings where fossil leaves are preserved. However,

this is still considered a provisional calibration dataset because it includes only ten sites from a restricted geographic area (Spicer et al., 2011).

**5.2.4 Recommended methodology for leaf morphology-based approaches to estimating paleotemperature**

A multi-proxy approach to reconstruction of land-based climate parameters is advocated where possible, including alternative

leaf morphology-based methods, taxonomic (Nearest Living Relative) and geochemical approaches.  A statistically assessed consensus approach (Ballantyne et al., 2010; Greenwood et al., 2017; Lowe et al., 2018) is recommended where feasible. For LMA, careful consideration should be given to the choice of calibration dataset, which have significant geographic variation. For CLAMP, the collecting and scoring protocols outlined on the CLAMP website should be followed to ensure the most robust estimates are produced. Updates to the calibration datasets and CLAMP methodology are also available through the

CLAMP website (http://clamp.ibcas.ac.cn/). It should be noted that both LMA and CLAMP are only applicable to leaf assemblages of woody dicot species and should be applied to fossil assemblages with at least 20 morphotypes (e.g. Wolfe, 1979; Povey et al., 1994; Spicer et al., 2004, 2005).

**5.3 Nearest Living Relative analysis for estimating paleotemperature**

**5.3.1 Theoretical background of Nearest Living Relative (NLR) analysis**

Nearest Living Relative (NLR) approaches are widely used techniques to derive quantitative climate parameters from pre-Quaternary fossil pollen or floral assemblages. In contrast to leaf morphology-based methods that relate physical attributes of plants to climate, NLR uses a taxonomic approach to derive climate parameters from fossil pollen assemblages based on the climatic ranges of their present-day nearest living relatives. The NLR approach includes several different methods such as the

Mutual Climate Range technique (e.g. Pross et al., 2000; Thompson et al., 2012), Bioclimatic Analysis (e.g. Kershaw and Nix, 1988; Greenwood at al., 2005; Eldrett et al., 2009; Pross et al., 2012; Pancost et al., 2013), Climatic Amplitude Method (e.g. Faquette et al., 1998; Suan et al., 2017), the Coexistence Approach (Mosbrugger and Utescher, 1997; Carpenter et al., 2012; Pross et al, 2012; Pancost et al., 2013; Utescher et al., 2014) and variations of the Probability Density Function method (Kühl et al. 2002; Greenwood et al. 2017; Harbert and Nixon, 2015; Hyland et al.2018).

All these NLR methods apply similar principles. Each method requires a modern vegetation database that can be linked to modern climatology to calculate climate requirements of fossil taxa derived from the modern distribution of their NLR. The climate parameters usually include mean annual air temperature (MAAT or LAT as used here), coldest/warmest month mean temperature and the mean annual precipitation range. To generate a paleoclimate estimate, the climate ranges in which the NLRs of a given fossil assemblage can coexist must be usually defined for each parameter. The climate estimates are usually shown as climate ranges of equally possible values within which a fossil plant community could have co-existed or as point values with confidence intervals. The latter should be the preferred format for further comparisons with other proxies and climate model outputs.

Concerns about the analytical approach of the Coexistence Approach (CA), one of the most widely used methods to derive quantitative climate estimates from pre-Quaternary palynological data, have recently been raised in regard to the treatment of outliers, and reliance on single taxa to define upper and lower limits for climate parameters (Grimm and Denk, 2012; Grimm and Potts, 2016). The latter significantly increases the uncertainty in CA-based climate reconstructions of "deep-time" environments that yield many extinct and relict taxa with NLRs whose climate envelope might have changed over time (e.g. Reichgelt et al. 2018). Probability Density Function (PDF) methods (Chevalier et al., 2014; Kühl et al. 2002; Greenwood et al. 2017) or the coexistence likelihood estimation method (Harbert and Nixon, 2015), which statistically constrain the most likely climatic co-occurrence envelope, offer a statistical solution to reduce the potential impact of wrongly defined climate tolerances.

### 5.3.2 Strengths of NRL analysis

NLR analysis uses the presence or absence of individual taxa in fossil assemblage rather than relative abundance, which reduces the likelihood of taphonomic biases. This facilitates, to some extent, the reconstruction of past non-modern analogue climates and environments (Pross et al., 2000; Utescher et al., 2014). Due to the use of presence-absence data, the method also does not require a calibration dataset using modern plant communities. Other methods that assess the direct relationship between pollen abundance and climate (e.g. transfer function techniques), require large calibration datasets from modern environments and their use for deep-time climate reconstruction is therefore very limited (e.g. Bartlein et al., 2011; Garreta et al., 2012). NLR can be applied to a wide range of plant fossils, including seeds, woods, pollen/spores and leaves. Despite the limitations outlined below, NLR-derived temperatures are generally in good agreement with estimates derived from other paleobotanical and geochemical approaches, including CLAMP and LMA(e.g. Uhl et al., 2003; Roth-Nebelsick et al., 2004; Pound and Salzmann, 2017; Pross et al., 2012).

### 5.3.3 Weaknesses of the NRL analysis

Quantitative climate estimates from the pre-Quaternary fossil plant record are always accompanied by large uncertainties. The greatest weakness that affects all NLR approaches is the assumption of uniformitarianism, namely that the climate tolerances of modern species can be extended into the past. This assumption inevitably introduces uncertainty that increases with the age of the geological record. Also, the modern distribution of a plant species is not only controlled by its climate envelope, but by a range of other factors such as biogeographic history and competition. By incorporating as many taxa as possible into the NLR analyses, uncertainties relating to such factors can be minimized to some extent (Harbert and Nixon 2015; Willard et al. 2019).

The accuracy of estimates derived from NLR depend on the quality of the modern climate and plant distribution database, uncertainties in the identification of a fossil taxa and their nearest living relatives, as well as on the total number of taxa within a fossil assemblage (Utescher et al., 2014). Accuracy is greatly reduced in sites with low taxa diversity (such as semi-deserts), large catchments (such as marine records), or non-modern analogue vegetation (such as the Arcto-Tertiary relict flora). NLR methods do also not account for changes in the relative abundance of fossil taxa or long-distance transport of palynomorphs resulting in "mixed" assemblages from different habitats or vegetation zones. This latter factor places an important caveat on applying NLR to terrestrial assemblages in marine sediments. By its nature, NLR cannot accurately predict environmental conditions beyond the present-day range. For instance, maximum LAT estimates plateau at ~25 for CA (Utescher et al., 2014).

### 5.3.4 Recommended methodology for NLR Analysis

When deriving climate estimates from NRL analyses, we recommend, whenever possible, a statistically assessed ensemble approach, where multiple terrestrial climate proxies, including geochemical and paleobotanical methods, are combined and assessed using probability density analysis (e.g. Ballantyne et al., 2010, Pross et al., 2012; Greenwood et al., 2017, Lowe et al., 2018). Probability density functions should also be used to overcome the problematic use of absolute limits to describe climate response ranges (Kühl et al., 2002). NLR-derived climate estimates should be presented with confidence ranges including the uncertainty associated with the estimate of climate tolerances (Grimm et al., 2016). For comparisons using larger datasets, each climate estimate should be rated using a qualitative ranking system (i.e. confidence assessment) that addresses major influencing factors, such as taphonomy, diversity and age control (Salzmann et al., 2013). The confidence level should be used in addition to the indicated temperature range as a guide to assess the robustness of a temperature estimate.

The quality of the databases used to derive modern distribution and climate from the NLR is of vital importance for the accuracy of the paleoclimate estimates. Preference should be given to databases with clear policies on data quality and ownership, global coverage and long-term secured funding. We therefore recommend that NLR methods use the GBIF (Global Biodiversity Information Facility) database for sourcing distributions of extant taxa (http://gbif.org), and that climate range data is derived from a global climate surface model such as WorldClim2 (Fick and Hijmans, 2017). Most NRL methods require at different stages of their analyses and database queries a pre-selection of modern and fossil data, which is necessary, for

example, to exclude introduced garden plants from the modern dataset, redundant occurrences or taxa with doubtful assignments (e.g. Willard et al. 2019). Fossil datasets are often reduced to focus the analyses on a selected set of reliable climate indicators (e.g. Chevalier et al. 2019). To warrant transparency of the NLR process and reproducibly of the final estimates, it is essential that all steps of the data selection process are accurately described and published.

## 6 Proxies for atmospheric CO₂

In this section, we outline the four main approaches for reconstructing atmospheric $CO_2$ concentrations: the boron isotopic composition of marine carbonate, the carbon isotopic composition of marine phytoplankton, stomatal morphology and the carbon isotopic composition of land plants, and the carbon isotopic composition of pedogenic carbonates. We also include in

our $CO_2$ compilation (Fig. 6) one estimate from the nahcolite proxy, which is based on mineral phase equilibria (Lowenstein and Demicco, 2006; Jagniecki et al., 2015).

### 6.1 Boron-based CO₂ reconstructions

### 6.1.1 Theoretical background of boron-based CO₂ reconstructions

Boron isotopes within foraminiferal calcite ($\delta^{11}B_c$) have been used extensively to reconstruct atmospheric $CO_2$ concentrations on a variety of different timescales (Foster, 2008; Hönisch et al., 2009, 2012; Pearson et al., 2009; Seki et al., 2010; Anagnostou et al., 2016; Gutjahr et al., 2017). The $\delta^{11}B$-pH proxy has a strong grounding in theory (Hönisch et al., 2009; Foster and Rae, 2016) and is based on several foraminiferal calibration studies (e.g. Sanyal et al., 2001; Henehan et al., 2013; Foster and Rae, 2016).

### 6.1.2 Strengths of boron-based CO₂ reconstructions

Boron-based $CO_2$ reconstructions spanning the last ~800,000 years show excellent agreement with ice core $CO_2$ estimates (Hönisch et al., 2009). The results of boron isotope studies have also been found to be highly reproducible between different laboratories (Foster et al., 2017; Gutjahr et al., 2017), allowing the integration of multiple datasets and providing high

confidence in the fundamental measurement of boron isotopes in biogenic carbonates. In common with other marine $CO_2$ proxies (e.g., alkenones below), age constraints are also typically strong for $\delta^{11}B_c$ and it is easy to correlate the resultant $CO_2$ estimates with marine-based climate proxies, some of which can be measured on the same material (e.g. $\delta^{18}O$, $\Delta_{47}$ and Mg/Ca in planktic foraminifera). Finally, detailed studies comparing $\delta^{11}B_c$ from contemporaneous diagenetically recrystallized foraminifera and "pristine" glassy foraminifera show no influence of diagenesis on the recorded values (Edgar et al., 2015),

indicating that $\delta^{11}B_c$ is more diagenetically robust than $\delta^{18}O_c$; the reasons for this are not yet fully understood (Edgar et al., 2015).

### 6.1.3 Weaknesses of boron-based CO₂ reconstructions

The complications of the boron isotope proxy can be separated in two categories, based on the two main steps required to estimate $CO_2$ from $\delta^{11}B_c$ values: (i) the calculation of pH from measured $\delta^{11}B_C$ values that requires additional constraints on seawater $\delta^{11}B$ ($\delta^{11}B_{sw}$), the vital effects in extinct species, the calcification temperature, the relative depth habitat of foraminifera and the extent of any diagenetic overprint on primary $\delta^{11}B_c$ values, and (ii) the translation of the pH estimate into one of atmospheric $CO_2$, which first requires an estimate of a second carbonate system parameter to infer aqueous $CO_2$ concentrations at specific seawater concentrations for Mg and Ca, and determination of $CO_2$ disequilibrium between surface waters and the atmosphere at the study site. The influence of the latter can be reduced by careful site selection targeting oligotrophic settings.

Offsets between the boron isotopic composition of ambient seawater borate ion ($\delta^{11}B_{borate}$) and that of foraminiferal calcite are generally known as 'vital effects' following the usage of Urey et al. (1951) for oxygen isotopes. Many different processes could, in principle, be responsible for such offsets which could also differ between species. Of particular concern is the modification of pH in the diffusive boundary layer around foraminifera by the photosynthesis of algal symbionts (Hönisch et al., 2003). Other possibilities include physiological controls of the local pH during calcification (Rollion-Bard and Erez, 2010) and the incorporation of boron from aqueous boric acid (Vengosh et al., 1991).

### 6.1.4 Recommended methodology for boron-based $CO_2$ reconstructions

The recommended procedure for boron isotope reconstructions of atmospheric $CO_2$ is designed to constrain unknowns and limit uncertainties associated with the proxy method. First, $\delta^{11}B_{sw}$ can be constrained in a number of ways: (i) by modelling the pH profile of the water column using multiple species (Pearson and Palmer, 1999, 2000; Anagnostou et al., 2016); (ii) using pairs of planktic/benthic foraminiferal $\delta^{11}B_c$ (Greenop et al., 2017); (iii) utilizing existing records of benthic foraminiferal $\delta^{11}B_c$ coupled with assumptions regarding deep water pH evolution (Raitzsch and Hönisch, 2013); (iv) using geochemical box models of the boron cycle (Lemarchand et al., 2000). All of these approaches indicate that the maximum rate of $\delta^{11}B_{sw}$ change is most likely ~0.1 ‰/my, which is consistent with the long residence time of boron in the ocean (10–20 my), its relatively large oceanic budget and the modern fluxes of boron in and out of the ocean (Lemarchand et al., 2002). Future studies should reconstruct surface ocean pH depth profiles using foraminiferal $\delta^{11}B_c$ to constrain changes in $\delta^{11}B_{sw}$ on shorter time scales, especially during periods when abrupt changes in weathering inputs of boron are possible. It is unlikely that these would be the main control on the evolution of $\delta^{11}B_{sw}$ over longer time scales (e.g. Lemarchand et al., 2002). Alternatively, carefully spaced, short (< 1 my) time series of relative changes in pH, and hence $CO_2$ (Foster and Rae, 2016), could bypass the need of accurate $\delta^{11}B_{sw}$ constrains, as they will encompass minimal change in seawater boron composition, typically of a magnitude less than analytical precision (i.e. <0.2 ‰).

Second, the issue of vital effects can be directly investigated within modern species by controlled culturing experiments (Sanyal et al., 1996, 2001; Foster, 2008; Henehan et al., 2013, 2016). For extinct species, vital effects can only be assessed

indirectly. Recent work suggests that vital effects of Paleogene foraminifera play a less significant role in determining $\delta^{11}B_{borate}$ than in more recent times (Penman et al., 2014; Anagnostou et al., 2016). Analysis of monospecific (or similarly behaving) foraminiferal species in terms of $\delta^{18}O_c$, $\delta^{13}C_c$ and $\delta^{11}B_c$ – will reduce uncertainties in pH and $CO_2$ reconstructions.

Third, estimates of salinity and temperature exert a relatively minor influence on estimates of pH (0.005 pH per psu change and 0.01 pH per $^o$C) in carbonate system calculations. Nevertheless, seawater temperature estimates are commonly derived from paired analyses of $\delta^{18}O_c$ (Pearson et al., 2007; John et al., 2013; Anagnostou et al., 2016) or Mg/Ca (Pearson et al., 2009; Foster et al., 2012; Martinez-Boti et al., 2015.

Finally, an additional carbonate parameter is required to calculate $CO_2$ concentrations at any given seawater salinity and temperature. For pre-Quaternary applications, this is typically provided by assumptions regarding the saturation state of calcium carbonate or alkalinity estimates constrained using geochemical models (Ridgwell, 2005; Pälike et al., 2012; Anagnostou et al., 2016; Gutjahr et al., 2017). Modelling studies indicate that the saturation state of calcium carbonate in surface waters (surface $\Omega_{calc} = [Ca]_{sw} * [CO_3^{2-}]/K_{sp}$) remains essentially constant through the Cenozoic, independent of model 15   boundary conditions (Tyrrell and Zeebe, 2004; Ridgwell and Zeebe, 2005; Hönisch et al., 2012; Pälike et al., 2012). This has recently been confirmed empirically, at least for the last 20 Myrs (Sosdian et al., 2018). Estimates of $[Ca]_{sw}$ are, however, required to derive carbonate ion concentration ($[CO_3^{2-}]$), which is used as the second carbonate system parameter, and these estimates are typically derived from fluid inclusion analysis (e.g. Horita et al., 2002).

At this stage, mean, upper and lower bounds of atmospheric $CO_2$ are calculated using a Monte Carlo approach to solve the relevant carbonate system equations. All simulations are iterated assuming Gaussian distribution of these parameters within their $2\sigma$ uncertainties. For pre-Quaternary time scales, propagated estimates of $CO_2$ carry an uncertainty of ~25% (95% confidence intervals). Relative $CO_2$ estimates, however, carry a much-reduced uncertainty of 2-10% (Foster and Rae, 2016).

**6.2 Carbon isotopic composition of marine phytoplankton**
The carbon isotopic composition of marine phytoplankton, as recorded by bulk organic matter, chlorophyll-related compounds (porphyrins, pristane and phytane) or organism-specific biomarkers, has long been used to reconstruct changes in atmospheric $CO_2$ (i.e. Freeman and Hayes, 1992). This method relies on the concentration of $CO_2$ in seawater being the primary influence on the magnitude of carbon isotopic fractionation between dissolved inorganic carbon in the growth medium and the primary 30   photosynthate of the algal cell (Rau et al., 1989). The fractionation factor, $\varepsilon_p$, is also influenced by algal cell geometry (Popp et al, 1998) and physiology, including specific growth rates (Laws et al., 1995; Bidigare et al., 1997) and the type of carbon acquired and lost from the cells (Sharkey and Berry, 1985). The most widely applied proxy of this type is based on the $\delta^{13}C$ composition of haptophyte algal alkenone biomarkers (Freeman and Pagani, 2005; Pagani, 2014), which have been used to reconstruct atmospheric $CO_2$ from the middle Eocene to modern (Pagani et al., 2005; Bijl et al., 2010; Zhang et al., 2013).

Alkenones are typically scarce in sediments older than ~40 Ma (Brassell, 2014), which has encouraged the development of proxies based on similar principles using dinoflagellate cysts (Hoins et al., 2015; Wilkes et al., 2017) and other types of primary photosynthetic biomarkers with longer geological ranges (Freeman and Hayes, 1992; Naafs et al., 2016). Although unpublished, recent reports of the haptophyte-specific C37:2 alkenone and measurements of its carbon isotopic composition ($\delta^{13}C_{37:2}$) from early Eocene sediments in the northeast Atlantic (Eley et al. 2018) justify a consideration of the alkenone-$CO_2$ proxy as it might be applied to the DeepMIP EECO time slice in the near future.

### 6.2.1 Theoretical background of alkenone-based CO$_2$ reconstruction

The saturation state and isotopic composition of alkenone biomarkers, unsaturated ethyl and methyl ketones with carbon chain lengths of C37–C39, are the basis for SST (Brassell et al., 1986; Conte and Eglinton, 1993; Conte et al., 2006) and salinity proxies (e.g., van der Meer et al., 2007; Warden et al., 2016), as well as $CO_2$ estimation (e.g., Pagani, 2002; Zhang et al., 2013). In the modern oceans, alkenones are produced by a relatively limited number of calcifying haptophyte algae ("coccolithophores"), all within the family Noelaerhabdaceae and including *Emiliania huxleyi* and *Gephyrocapsa* species (Volkman et al., 1980; Conte et al., 1998; Pagani, 2014). Where alkenone-bearing sediments predate the first appearance of the modern alkenone-producing species, it is assumed that ancient relatives within the Noelaerhabdaceae, such as the early Eocene to modern genus *Reticulofenestra*, are the primary alkenone-producers (Pagani, 2014). Alkenone-based estimates of $CO_2$ rely on the relationship between $\varepsilon_p$ and [$CO_{2(aq)}$] in the surrounding growth medium, which is related to atmospheric $CO_2$ via Henry's Law (Freeman and Hayes, 1992; Pagani, 2014). Calculation of $\varepsilon_p$ requires determination of the difference between the $\delta^{13}C$ of $CO_{2(aq)}$ in the growth medium ($\delta^{13}C_{CO2(aq)}$) and the $\delta^{13}C$ of biosynthesised cellular organic matter ($\delta^{13}C_{org}$). The latter is measured on the haptophyte-specific C37:2 alkenone ($\delta^{13}C_{37:2}$) (Pagani, 2014; Zhang et al., 2013).

As lipids are depleted in $^{13}C$ relative to the carbon isotope composition of the whole cell a correction of +4.2‰ is applied when calculating $\delta^{13}C_{org}$ from $\delta^{13}C_{C37:2}$ (Schouten et al., 1998; Pagani, 2014). Alkenone-based estimates of ancient $CO_2$ assume diffusive transport of inorganic carbon across algal cell membranes (discussed below), with the total carbon isotope fractionation (Zhang et al., 2013; Pagani, 2014) being described by:

$$\varepsilon_{p37:2} = \varepsilon_t + (\varepsilon_f - \varepsilon_t)\frac{c_i}{c_e} \qquad\qquad (20)$$

where the carbon isotope fractionation associated with photosynthesis and carbon transport are denoted by $\varepsilon_f$ – typically ~25 to 28 ‰ (Goericke and Fry, 1994; Farquhar and Richards, 1984) - and $\varepsilon_t$ respectively. $C_e$ is the ambient, and $C_i$ the intracellular, concentrations of aqueous $CO_2$. The physiological factors that control the $C_i$ term are complex and poorly known, and so a simplified version of this relationship is typically used (Jasper et al., 1994; Rau et al., 1996):

$$\varepsilon_{p37:2} = \varepsilon_f - \left( \frac{b}{[CO_{2(aq)}]} \right) \tag{21}$$

where the $b$ term integrates a suite of physiological variables, including cellular growth rate and cell geometry, that can influence photosynthetic carbon isotope fractionation (Pagani, 2014 and references therein). Studies reconstructing ancient

$CO_2$ typically assume that the $b$ term is correlated to nutrient levels, represented by phosphate concentrations, with a range of regressions presented by Pagani et al. (2014). Ultimately, $CO_2$ is calculated from $[CO_{2(aq)}]$ by applying Henry's Law:

$$CO_2 = [CO_{2(aq)}]/k_H \tag{22}$$

$k_H$ is defined as:

$$k_H = (-0.00004 * T^3 + 0.0048 * T^2 - 0.2313 * T + 6.2665)/100 \tag{23}$$

For a salinity of 35‰ (Weiss, 1974) and where 'T' is an independently-derived estimate of sea surface temperature, best

constrained by the $U_{37}^{K\prime}$ index derived from the same alkenone assemblage, or else another DeepMIP recommended SST proxy methodology if SSTs are beyond the limit of $U_{37}^{K\prime}$.

.

### 6.2.2 Strengths of the alkenone $CO_2$ method

The relatively high abundance of alkenone biomarkers throughout the Neogene, and the relative ease of analysis of $\delta^{13}C_{37:2}$,

makes alkenone $CO_2$ estimation suitable for high temporal-resolution studies of $CO_2$ dynamics. The alkenone $CO_2$ method is advantageous over other estimates of $\varepsilon_p$, such as the use of bulk organic carbon or other less specific biomarkers, because of the high specificity of marine alkenone production to one family within the haptophyte algae (Volkman et al., 1980; Conte and Eglinton, 1993). This source-specificity also allows for independent constraints on other factors that influence $\varepsilon_p$ including cell geometry and physiology, based on the coccolithophore fossil record (Henderiks and Pagani, 2007; Pagani, 2014; Bolton et

al., 2016). Finally, as the distribution ratio of alkenones produced by haptophyte algae is closely related to sea surface temperature (SSTs) ($U_K^{37}$ index; applicable below ~28°C) (Conte et al., 2006), similar to $\delta^{11}B$ methods, it is possible to directly generate parallel records of two key parameters for $CO_2$ estimation - SSTs and $\varepsilon_{p37:2}$.

### 6.2.3 Weaknesses of the alkenone $CO_2$ method

There are several uncertainties and assumptions in translating measured $\delta^{13}C_{37:2}$ first into a value for $\varepsilon_{p37:2}$, and then into an estimate of $[CO_{2(aq)}]$ and $CO_2$. In particular, robust proxy estimates of $\delta^{13}C_{CO2(aq)}$ and SST are required. These have not always been available in previous alkenone-based estimates of Paleogene $CO_2$ (Pagani et al., 2005). Inaccurate estimates of SST can cause substantial error in alkenone-based $CO_2$ estimates (Super et al., 2018), especially at high $CO_2$ where $\varepsilon_p$ approaches the

assumed value for $\varepsilon_f$ (Bidigare et al., 1997; Riebesell et al., 2000; Pagani et al., 2011; Pagani, 2014). Although the '$b$' term integrates the effects of a number of physiological variables (e.g., specific growth rate, cell geometry) on net carbon isotope fractionation during photosynthesis (Bidigare et al., 1997; Riebesell et al., 2000; Pagani et al., 2011; Pagani, 2014) it is typically derived from the strong empirical relationship with surface ocean phosphate concentrations [$PO_4^{3-}$] in the modern oceans

(Bidigare et al., 1997). Although there have been attempts to disaggregate cell geometry effects (e.g. Henderiks and Pagani 2007), most existing records of ancient $CO_2$ rely on assumed ranges of the $b$ term based on estimates of ancient [$PO_4^{3-}$] (Pagani, 2014).

Alkenone-based $CO_2$ estimates also assume the dominance of diffusive $CO_2$ transport across the algal cell membrane (Zhang

et al., 2013). Some studies have shown the operation of carbon concentrating mechanisms (CCMs), including the potential use of bicarbonate carbon (Holtz et al., 2015, 2017), under conditions of $CO_2$ limitation, resulting in non-linear relationship between $\varepsilon_p$, growth rate and $CO_{2aq}$ (Riebesell et al., 2000; Zhang et al., 2013). Such CCM activity could impact the value of $\varepsilon_p$ through active transportation of carbon species with varied isotope compositions (Zhang et al., 2013). Ongoing empirical and modelling studies of carbon uptake and partitioning (Bolton and Stoll, 2013; Holtz et al., 2017, 2015; McClelland et al., 2017),

and those examining the role of environmental conditions on CCM behaviour (Raven and Beardall, 2014; Raven and Hurd, 2012) will hopefully provide constraints on the ubiquity, importance and impact of these mechanisms within the alkenone-producing haptophytes. Finally, changes in growth rate can impact the isotopic offset ($\Delta\delta$) between alkenone $\delta^{13}C_{C37:2}$ and the $\delta^{13}C_{org}$ of the whole cell. $\Delta\delta$ is commonly set at 4.2‰, although culture study estimates typically range from 3.1 to 5.9‰ (Pagani, 2014 and references therein), up to a reported maximum of 7-12‰ (Benthien et al., 2007). Where $\varepsilon_p$ is high, changes

of this order in $\Delta\delta$ can exert a significant impact on reconstructed $CO_2$ values (Pagani, 2014).

### 6.2.4 Recommended methodologies for the alkenone $CO_2$ method

The alkenone $CO_2$ method requires estimation of several physiological ($\varepsilon f$, cell size, '$b$') and environmental ([$PO_4^{3-}$], SST, $\delta^{13}C_{CO2(aq)}$) parameters. Reconstruction of SSTs should follow best practice from one of the established proxies outlined in

Section 4 and, where possible, SST uncertainty should be propagated through to final estimates of $CO_2$. The most reliable estimates of $\delta^{13}C_{CO2(aq)}$ are derived from the measurement of test $\delta^{13}C$ compositions of well-preserved, mixed-layer dwelling foraminifera, within the narrow 250 – 300μm size window, which are closest to equilibrium with modern $\delta^{13}C_{DIC}$ (Birch et al., 2013). Although very low $\varepsilon_f$ values for *Emilliania huxleyi* of ~11‰ have been published (Boller et al., 2011), these results are yet to be replicated, and we recommend the existing approach of providing propagated $CO_2$ uncertainty based on a 25-28‰

range of $\varepsilon_f$ for algae using C3 pathways (Pagani, 2014).

As with $\varepsilon_f$, establishing robust estimates for the $b$ factor in ancient communities is non-trivial, and we recommend a similar propagation of uncertainty based on a range of $b$ factors drawn from surface water [$PO_4^{3-}$] measurements in analogous modern environments (Zhang et al., 2013). Given the potential importance of cell size within the physiological factors that make up

the *b* term, some studies have sought to use an explicit cell-size 'correction' to alkenone $CO_2$ reconstructions (Bolton et al., 2016). However, the precise application of a cell size correction and its relation to the standard *b* term remains unclear. Here, we recommend that reticulofenestrid coccolith size is measured along with the determination of $\delta^{13}C_{37:2}$, at least for the purposes of screening out the impact of major shifts in cell-size on $CO_2$ records (Pagani, 2014), and for potential future use to improve quantitative estimates of the *b* factor.

### 6.2.5 Other algal substrates for $CO_2$ estimation

*Dinoflagellates*: Significant progress has been made towards a $CO_2$ proxy based on the carbon isotope fractionation of phototrophic dinoflagellates and their preservable cysts (dinocysts) (Hoins et al., 2015, 2016a, b, Wilkes et al., 2017). A strong correlation between $\varepsilon_{pDINO}$ and $CO_2$ was found in four cultured dinoflagellate species (Hoins et al., 2015). Importantly, there appears to be species-specific effects on $\varepsilon_{pDINO}$ based on uptake and leakage of different carbon species ($HCO_3^-$ and $CO_{2(aq)}$) and light intensity, making single-species records preferable (Hoins et al., 2016a, b). Dinocysts are found in the geological record from the Triassic onwards (MacRae et al., 1996) and several extant species are remarkably long ranging (e.g. Williams et al., 2004). This includes two species used in culture experiments (Hoins et al., 2015), *Protoceratium reticulatum* and *Gonyaulax spinifera*, which are preserved in the fossil record as dinocyst species *Operculodinium centrocarpum* and *Spiniferites ramosus*, respectively. The geological record for *O. centrocarpum* and *S. ramosus* spans the last 60 my and 130 my, respectively (e.g. Williams et al., 2004). The morphology and ecological affinity of these ubiquitous species is well known and apparently unchanged since the early Paleogene (Zonneveld et al., 2013, Sluijs et al., 2005, Frieling and Sluijs, 2018).

Unlike biomarker-based $CO_2$ proxies, individual dinocysts of a single species are selected based on morphological characteristics prior to analysis to maximize the similarity in size and shape between analysed individuals, both within and between samples. A recently developed laser-ablation nano-combustion isotope ratio mass spectrometry (LA-nC-IRMS) method (van Roij et al., 2017) allows for accurate analyses of the carbon isotope composition of single dinocysts (Sluijs et al., 2017).

While results from culture experiments are promising, it should be noted that several, potentially important, factors should be constrained before $\varepsilon_{pDINO}$ can be widely applied as a $CO_2$ proxy. For example, current efforts aim to resolve the potential offset between the cultured motile cell and the preservable cyst as well as the relation of $\varepsilon_{pDINO}$ to parameters other than $CO_2$, both in culture and in their natural environment. Although these and perhaps other issues still need to be addressed, the $CO_2$ dependent $\varepsilon_{pDINO}$ in several dinocyst species may represent an opportunity to reconstruct $CO_2$ in geological deep time, excluding any morphology, species or genus-related factors in $CO_2$-dependency that may cause significant uncertainty in $CO_2$ estimates based on less specific algal compounds.

***Phytane and general biomarkers***: The carbon isotopic composition of phytoplankton has also been tested in a variety of general biomarkers, compounds which are representative of overall phytoplankton community. Chlorophyll, the primary pigment in all photoautotrophs, is of particular interest for reconstructing secular trends given its spatially and temporally ubiquitous presence in the geologic record. For this reason, studies on both its porphyrin head (Popp et al., 1989) and its side-
chain phytol preserved as the diagenetic product phytane (Freeman and Hayes, 1992) are some of the earliest uses of $\varepsilon_p$ for $CO_2$ reconstruction. Here, we will discuss phytane as an example of using general biomarkers for estimating past $CO_2$.

The overall theoretical background for using general biomarkers, such as phytane, is the same as the alkenone-based reconstructions. Like other lipids, phytane is depleted in $^{13}C$ relative to the carbon isotopic composition of the whole cell; a
correction of $3.3 \pm 1.3$‰ standard deviation is applied when calculating $\delta^{13}C_{org}$ from phytane, based on the average of twenty-two cultured species (Witkowski et al., 2018). As seen with the other $\varepsilon_p$-based proxies, the physiological variable $b$ is difficult to constrain. Studies for phytane have used a constant $b = 170$‰ kg $\mu M^{-1}$ (Bice et al., 2006; Damsté et al., 2008; Naafs et al., 2016), consistent with average $b$ found in modern marine sediments for total organic matter at $168 \pm 43$ ‰ kg $\mu M^{-1}$ (Witkowski et al., 2018) and for alkenone at $165 \pm 53$ ‰ kg $\mu M^{-1}$ (Pagani, 2014). All other parameters are treated as seen in the alkenone-
based reconstructions.

Based on these assumptions, a recent study shows that phytane yields a robust and consistent trend over the Phanerozoic, the longest record for $CO_2$ from a single proxy (Witkowski et al., 2018). This record has the potential to span even further in time given that the $\delta^{13}C$ of phytane has been measured in 1.73 Ga sediments (Li et al., 2003). This opportunity to observe changes
in $CO_2$ from a single proxy is beneficial to understanding secular trends. In addition to its span in preservation, phytane is one of the most abundant biomarkers in both marine sediments and oils. This availability of phytane is critical during periods where other proxies are rare, extending beyond the alkenone and boron records, most notably used for reconstructing Cretaceous $CO_2$ (Bice et al., 2006; Sinninghe Damsté et al., 2008; Naafs et al., 2016). The ubiquity of phytane throughout time, location, and sample type lends to the possibility for amassing large data sets which can be used for developing robust average estimations
for $CO_2$.

Although the phytane-based reconstructions show remarkably similar trends to other $CO_2$ proxies over the Phanerozoic, this biomarker requires more investigation. Nearly all studies regarding the mechanisms behind $\varepsilon_p$ for $CO_2$ reconstructions have been species-specific and almost exclusively for alkenone-producing Haptophytes. It is unknown whether integrating the entire
phytoplankton community, as opposed to several species, has an effect on the observed signal. Studies testing the different parameters and other potential effects on fractionation are still needed in order to properly constrain the uncertainties on using phytane.

**6.3 Leaf stomatal and gas-exchange proxies**

### 6.3.1 Theoretical background, strengths, and weaknesses of leaf stomatal and gas-exchange proxies

Plants interact directly with the atmosphere, making them well suited as climate proxies. Woodward (1987) and subsequent work (e.g., compilation in Royer, 2001) demonstrated that the stomatal density and stomatal index (stomatal density / stomatal + epidermal cell density) in many plant species respond inversely to the partial pressure of atmospheric $CO_2$; these responses are usually species-specific. If the stomatal response to $CO_2$ is known (usually from herbaria specimens and elevated $CO_2$ experiments), it can be applied to fossils of the same species to reconstruct paleo-$CO_2$. The genetic (Casson and Gray, 2008), signaling (Lake et al., 2002), and functional bases of the stomatal-$CO_2$ response are known reasonably well: theoretical considerations of the latter predict an inverse power law between stomatal density and $CO_2$ (Wynn, 2003; Franks et al., 2014; Konrad et al., 2017). The stomatal proxy is popular, having been applied in dozens of studies to produce over 300 estimates of paleo-$CO_2$ across the Phanerozoic, including the early Paleogene (Royer, 2014). A related proxy called the stomatal ratio relates the ratio of paleo-atmospheric $CO_2$ to present-day $CO_2$ to the ratio of stomatal density in a fossil to that in its nearest living equivalent (i.e., not necessarily the same species) via a one-point calibration and an assumed exponent of -1 for the power law. Because of these features, the stomatal ratio is best used for tracking relative (not absolute) changes in $CO_2$ (McElwain and Chaloner, 1995).

Some limitations of the stomatal proxy are: 1) only species whose stomatal density responds to $CO_2$ can be used; 2) extant calibrations are laborious to construct, and most fossil plant species are not alive today; 3) the method is empirical, not explicitly mechanistic, meaning that it is difficult to exclude the possibility that the extant relationships differed in the geologic past; and 4) in many species the upper error limit on the $CO_2$ estimate becomes very large or unbounded when the estimated $CO_2$ is moderately high or higher. An alternative approach that partly addresses these limitations is a mechanistic model based on the fundamentals of leaf gas-exchange (Konrad et al., 2008, 2017; Franks et al., 2014). The core of these models is the well-known relationship between leaf assimilation rate and the product between total leaf conductance (which partly reflects the "supply" side of assimilation) and the $CO_2$ gradient between the atmosphere and intercellular spaces (the "demand" side) (Farquhar and Sharkey, 1982). The critical measurements needed to estimate atmospheric $CO_2$ are the leaf $\delta^{13}C$, stomatal size and density, and an estimate of assimilation rate from a nearest living relative. The gas-exchange proxy of Franks et al. (2014) typically has 95% confidence intervals within -25% to +35% of the median $CO_2$ estimate, on par with other popular paleo-$CO_2$ proxies.

### 6.3.2 Recommended methodology of leaf stomatal and gas-exchange proxies

Both the stomatal ratio and leaf-gas-exchange proxies can be applied to most fossil leaves, even if the taxa are extinct today. Because the $CO_2$ estimates from stomatal ratios are "semi-quantitative" (McElwain and Chaloner, 1995), they are best used when complementary $CO_2$ information is present (e.g., Montañez et al., 2016). The stomatal density and stomatal index proxies work well when calibrated taxa are present as fossils and when estimated $CO_2$ is low to moderate. At higher $CO_2$ (above ~500 ppm for many species; e.g., Smith et al., 2010; Doria et al., 2011), the upper error limit can be unbounded, meaning that only

minima can be quantified. One challenge with this proxy is the construction of robust calibrations, especially at the high end of the calibrations, because plants may need multiple growing seasons in experimentally elevated $CO_2$ concentrations before their stomatal densities and indices fully equilibrate (Royer, 2003; Hincke et al., 2016). The leaf-gas-exchange proxies are comparatively more recent, and so have been tested less. We encourage rigorous testing in nearest living relatives, especially for sensitive model inputs that cannot be measured directly in fossils, for example the assimilation rate at a known $CO_2$ concentration and the scaling between maximum and operational stomatal conductance to $CO_2$ (Maxbauer et al., 2014; McElwain et al., 2016a, 2016b, 2017; Franks and Royer, 2017).

In cases where multiple stomatal proxies have been applied to the same fossils, the corresponding $CO_2$ estimates are broadly similar (e.g., Maxbauer et al., 2014; Montañez et al., 2016; Richey et al., 2018), conferring mutual support for the methods. Generating $CO_2$ estimates from multiple taxa typically leads to a more accurate consensus estimate outcrop (e.g., Montañez et al., 2016; Kowalczyk et al., 2018; Richey et al., 2018; Steinthorsdottir et al., 2018). Routines for propagating all known uncertainties associated with the leaf-gas-exchange and stomatal density / index proxies have been developed (Beerling et al., 2009; Franks et al., 2014) and should be used.

### 6.4 Pedogenic carbonate proxy for atmospheric $CO_2$

For soils developing on carbonate-free bedrock, soil $CO_2$ is a mixture between two sources: atmospheric $CO_2$ that has diffused into the soil and $CO_2$ produced in-situ via biological respiration. In arid and semi-arid climates, carbonate commonly precipitates in soils and its $\delta^{13}C$ value reflects that of the mixed soil $CO_2$. Because the $\delta^{13}C$ of the two sources is different, the concentration of atmospheric $CO_2$ can be calculated with a two-end-member mixing model if the atmospheric $\delta^{13}C$, pedogenic carbonate $\delta^{13}C$, and concentration and $\delta^{13}C$ of biological $CO_2$ is known (Cerling, 1991, 1999). This pedogenic carbonate proxy has been widely applied across the Phanerozoic: over 500 estimates from dozens of studies (Royer, 2014). One difficulty has been the estimation of the biological $CO_2$ concentration, $S(z)$. It is clear now that $S(z)$ was overestimated in most early studies because pedogenic carbonate mostly forms during the dry season, when biological productivity is lowest (Breecker et al., 2009, 2010). Several promising quantitative proxies for $S(z)$ have been proposed in recent years (Cotton and Sheldon, 2012; Montañez, 2013; Breecker and Retallack, 2014). The 95% confidence intervals typically range from -30/+40% to -50/+100% of the median estimate, depending on the characteristics of the paleosol in question (Breecker, 2013).

### 6.5 C₃ land plant carbon isotope proxy for atmospheric $CO_2$
### 6.5.1 Theoretical background of C₃ land plant proxy method

Plants are differentiated into three main categories based in the method by which they fix carbon: $C_3$, $C_4$ and CAM. The carbon isotope composition ($\delta^{13}C$) of $C_3$ plants, which use the RuBisCO enzyme (ribulose-1,5-bisphosphate carboxylase-oxygenase) to catalyze $CO_2$ fixation, has long been suspected to be a function of atmospheric $CO_2$ concentration (Farquhar et al., 1982; Krishnamurthy and Epstein, 1990; Körner et al., 1991; Leavitt and Danzer, 1992; Feng and Epstein, 1995). Subsequent

controlled growth chamber experiments conducted across a wide range of $CO_2$ show that carbon isotope discrimination between the $\delta^{13}C$ value of the atmosphere ($\delta^{13}C_{CO2}$) and the resulting plant tissue ($\delta^{13}C_{org}$) [$\Delta^{13}C = (\delta^{13}C_{atm} - \delta^{13}C_{org})/(1 + \delta^{13}C_{org}/1000)$] increases with increasing $CO_2$, according to the following equation (Schubert and Jahren, 2012):

$$\Delta^{13}C = [(A)(B)(CO_2 + C)]/[(A + (B)(CO_2 + C)) \tag{24}$$

Where A = 28.6, B = 0.22 and C = 23.9 (Schubert and Jahren, 2015). These data were consistent with other field and chamber experiments measured across smaller changes in $CO_2$ and revealed a unifying relationship between changes in $CO_2$ and changes in $\Delta^{13}C$ value (Schubert and Jahren, 2012). Using a large data compilation of terrestrial organic matter spanning the Pleistocene-Holocene transition, Schubert and Jahren (2015) showed how this relationship could be used to reconstruct $CO_2$ within the fossil record by relating changes in $CO_2$ to changes in $\Delta^{13}C$ value [i.e., $\Delta(\Delta^{13}C) = \Delta^{13}C_{(t)} - \Delta^{13}C_{(t=0)}$] between a time of interest ($t$) and a reference time ($t = 0$):

$$\Delta(\Delta^{13}C) = [(A)(B)(CO_{2(t)} + C)]/[(A + (B)(CO_{2(t)} + C)) - [(A)(B)(CO_{2(t=0)} + C)]/[(A + (B)(CO_{2(t=0)} + C) \tag{25}.$$

Excellent agreement between the proxy results and ice core data validated changes in $\Delta^{13}C$ value as an accurate proxy for $CO_2$ in the fossil record (Schubert and Jahren, 2015). Much of the application of this work has focused on the early Paleogene greenhouse and accompanying hyperthermals (Schubert and Jahren, 2013; Cui and Schubert, 2016, 2017, 2018). These studies have all revealed background $CO_2$ across the late Paleocene and early Eocene (prior to the EECO) less than ~3.5 times preindustrial levels (<1000 ppm), but with significantly higher levels at the hyperthermals.

### 6.5.2. Strengths of C₃ land plant proxy

The C₃ land plant proxy has the potential for wide applicability in the fossil record because it relies on measurements of the $\delta^{13}C$ value of terrestrial organic matter, an abundant and widely measured substrate that can integrate carbon contributions across a large number of plants. Notably, the proxy: (i) has been validated through comparison with ice core data (Schubert and Jahren, 2015); (ii) includes a robust uncertainty analysis for all input parameters and accounts for potential contemporaneous environmental change (Cui and Schubert, 2016); (iii) can resolve differences in carbon isotope excursion magnitude between terrestrial and marine substrates (Schubert and Jahren, 2013); (iv) provides a method for producing nearly continuous records of $CO_2$ at resolution sufficient for resolving early Paleogene CIEs, such as the PETM (Cui and Schubert, 2017); (v) can help identify potential sources of the CIEs (Cui and Schubert, 2018); (vi) is applicable across the entire range of Cenozoic $CO_2$, including the early Paleogene greenhouse (Cui and Schubert, 2016), and (vii) includes an understanding of the mechanism driving the observed increase in discrimination with increasing $CO_2$ (Schubert and Jahren, 2018) that fits within the framework for carbon isotope discrimination developed by Farquhar et al. (1982).

### 6.5.3 Weakness of C₃ land plant proxy

Weaknesses include: (i) potential for different responses among different plant groups (Hare et al., 2018; Porter et al., 2017); (ii) the effect of $O_2$ on $\Delta^{13}C$ is poorly constrained (Beerling et al., 2002; Berner et al., 2000; Porter et al., 2017), and (iii) the proxy saturates at very high $CO_2$, yielding positive uncertainties of >120% above 1000 ppmv (Cui and Schubert, 2016). Large uncertainties in $CO_2$ can be reduced if changes in other environmental parameters that are known to affect $\Delta^{13}C$ value (e.g., water availability, plant community shifts) are constrained (Cui and Schubert, 2016). The proxy also requires estimates of $\delta^{13}C_{CO2}$, which for the Paleocene and Eocene, can be calculated from the $\delta^{13}C$ value of benthic foraminifera (Tipple et al., 2010). Because the proxy is based on relative changes in $\Delta^{13}C$ value, quantitative estimates of $CO_2$ require either: (i) knowledge of $\Delta^{13}C$ value at a known $CO_2$ level (Schubert and Jahren, 2015; Eq. 25), or (ii) independent estimates of the change in $CO_2$ across the two time periods of interest (i.e., $\Delta CO_2 = CO_{2(t)} - CO_{2(t=0)}$; as shown in Schubert and Jahren, 2013). Marine carbonate records, which can be used to quantify $\Delta CO_2$, might be affected by dissolution (Cui and Schubert, 2017); changes in the $\Delta 13C$ value of paleosol carbonate mimic changes in the $\Delta^{13}C$ value of terrestrial organic matter, but might also be affected by diffusion of increased $CO_2$ into the soil and increased productivity (Schubert and Jahren, 2013). Therefore, careful consideration of the selected substrate and site is required; when information on environmental change is not available, use of multiple records might lead to better $CO_2$ estimates (Cui and Schubert, 2017).

### 6.5.4 Recommended methodology of C₃ land plant proxy

This proxy is ideal for reconstructing the wide range of $CO_2$ characteristic for the early Eocene and associated hyperthermals. Like many other $CO_2$ proxies, uncertainties increase with increasing $CO_2$; positive and negative errors, however, are less than ~120% and 40%, respectively, at the $CO_2$ levels predicted for the late Paleocene and early Eocene (Cui and Schubert, 2016). The choice of record is important for producing robust $CO_2$ estimates via Eq. (25) because there is potential for systematic bias in the calculated $CO_2$ estimate resulting from: (i) sampling resolution, (ii) substrate type, (iii) diagenesis, and (iv) environmental change (Cui and Schubert, 2017). For example, working in sites with high mean annual precipitation may help to reduce variability in median $CO_2$ estimates, as $\Delta^{13}C$ value is less responsive to changes in MAP in ever-wet sites (e.g., Diefendorf et al., 2010). However, Lomax et al. (2019) demonstrated experimentally that plants grown under water stress showed the same relative change in $\Delta^{13}C$ value with increasing $CO_2$ concentration as observed in well-watered experiments (e.g., Schubert and Jahren, 2012), but with lower $\Delta^{13}C$ values overall. Their experimental data are consistent with results that modelled the predicted effect of water stress on $\Delta^{13}C$ value across multiple levels of $CO_2$ (see Fig. 4a within Schubert and Jahren, 2018). Nevertheless, both chamber (Lomax et al., 2019) and deep-time (Barral et al., 2017) studies have tried to use Eq. (24) to quantify $CO_2$ and failed to account for the effects of growing conditions and plant species type on $\Delta^{13}C$ value. Such attempts highlight the need to use Eq. (25) with robust estimates of $\Delta(\Delta^{13}C)$ (after Schubert and Jahren, 2015) in order to accurately quantify $CO_2$.

### 7 DeepMIP data compilation

We have compiled data files for sea surface and land air temperatures (SST and LAT) through the late Paleocene and early Eocene (Supplementary Data Files 3–7) and atmospheric $CO_2$ reconstructions for the early Paleogene (Supplementary Data File 8) as the first stage of creating a DeepMIP database and to show how the methodologies outlined above can be applied to the three time slices. These datasets comprise 1701 records of SST, 510 of LAT and 539 of $CO_2$ (Table 1). As has been discussed above, each proxy is used with varying degrees of confidence. Temperature estimates from some proxies are inferred to be unreliable or have low confidence based on diagenetic alteration (e.g. planktic foraminiferal $\delta^{18}O$) and mixed provenance for GDGTs (TEX86, MBT-CBT). Further assessment of the reliability of specific proxy records is beyond the scope of this study but is an essential part of follow-up studies. Once the online platform has been completed, we encourage others to add to the database to better characterize and understand the climate of this warm, high-$CO_2$ time in earth history. The geographic coverage of sites with marine and terrestrial paleotemperature data is shown on early Eocene (52 Ma) paleogeographic maps (Fig. 7a, 7b). Fig. 7c illustrates how choice of plate tectonic reference frame has a significant effect on paleogeographic locations of the sites from which climate proxy data are derived (Matthews et al., 2016). Offsets in paleolatitude approach 10° in some regions. As noted previously (Lunt et al., 2017), we endorse the recommendation of van Hinsbergen et al. (2015) to use a paleomagnetic reference frame based on reconstructed polar wander paths because it can be directly related to the Earth's spin axis in preference to a mantle-based reference frame. In general, the paleomagnetic reference frame shifts Pacific sites northward and Atlantic sites southward. Because some models have already adopted a mantle-based reference frame (e.g. Herold et al., 2014), we include paleolocations derived from both reference frames in Supplementary Data File 2.

Although the number of sites for which robust proxy climate reconstructions are available continues to increase through the concerted efforts of paleoclimate research teams, the global coverage remains restricted, especially for marine proxy records of the EECO and terrestrial proxy records for the PETM and LP. For marine sites, we have compiled data for 24 LP, 27 PETM and 16 EECO sites (Fig. 7a). Best coverage is in the Atlantic but there are large tracts in the Pacific Ocean that lack data, especially in the central and northern west Pacific and the central and southern east Pacific. The Arctic Ocean remains represented by a single site. Our LAT data are mainly from terrestrial sites although several data sets are derived from marine sites, which have the added value of good chronology and accompanying data from marine proxies. Otherwise our data compilation is hampered by poor age control for many terrestrial sites, with numerous proxy records excluded because they have a broadly defined late Paleocene-early Eocene age range. For this reason, our compilation is dominated by later early Eocene records, which we include in a broadly defined EECO time-slice comprising 41 sites, with only five PETM and nine LP sites. There are vast tracts of the planet that remain poorly represented by terrestrial proxies, especially Asia, Africa, South America and Antarctica.

Once model-data comparisons are carried out with this new dataset, we anticipate that new regions and environmental settings will be identified for new data acquisition. These will include regions where there is disagreement between models and data, between different models, or between different proxies. A particular focus may be regions where significant climate gradients

are suggested either by models or data, such as the Southern Ocean (Douglas et al., 2014). We also anticipate ongoing focus on those environmental settings judged to have greatest potential for yielding robust temperature reconstructions, such as clay-rich marine settings (e.g., Pearson et al., 2007; Frieling et al., 2017, 2018) and ancient terrestrial peat deposits (Naafs et al., 2017b).

### 7.1 Data organization and management

At present, we include this version 0.1 of the DeepMIP database as supplementary files to this paper (Supplementary Data Files 1–8), but we intend to import these and other datasets into a bespoke online database. A comprehensive data-model and data-data comparison effort requires new tooling for data organization, manipulation and management. It is critically important

to provide full traceability of data sources and reproducibility of methods to produce derived data sets, both in terms of stratigraphic correlations, age-model development, and calculation of derived values from raw measurements. This effort has two main components: data organization and scientific workflow management.

We will organize data in a publicly accessible database that has a common format allowing easy data retrieval and

manipulation. This database is part of the EARTHSEQUENCING project and is currently in the process of incorporating primarily data from the various ocean drilling efforts (DSDP, ODP, IODP). The specific database system chosen allows much faster access than traditional relational databases while still preserving the typical cross-table relationships that allow relating parameters, e.g., core-depths and sample identifiers to derived properties like age models. This backend system stores the relevant data and meta-data and allows programmatic and external access to data and derived properties. In order to avoid the

slow-access and cumbersome retrieval of data from existing systems, where each individual measurement is stored in its own row, the EARTHSEQUENCING database stores data aggregated on a *per-Location* and *per-Analysis/Datatype* basis, with supporting structural meta-data, consistently formatted and heavily compressed. Overall this approach results in access times per hole and analysis on the order of seconds, thus allowing near real-time data retrieval and analysis. The actual data tables are flexible and can be imported from traditional text files that can be exported from all major spreadsheet applications. Where

necessary, existing stacks and compilations can be imported (e.g., Zachos et al., 2008).

Whilst the database system was primarily designed for IODP data sets, it is flexible enough to incorporate data and measurements from land-based sections. The primary requirement for data input and traceability are unique measurements and sample identifiers, following existing IODP standards, i.e. each measurement requires complete information about sample

depths and offsets, measurement units, and related meta-information such as site location, and publication reference where applicable. The database can also store additional information, such as biostratigraphic or paleomagnetic age constraints to facilitate easy and reproducible age model development, assigning of ages to measurements, and re-mapping between different age models. We also strongly encourage a full documentation of uncertainties in proxy records. As such, for all climate estimates in the DeepMIP database, the aim is that the following will be recorded: (a) analytical uncertainty, (b) calibration

uncertainty, and (c) uncertainties associated with generating mean or median values for data records within time slices. This should include separate estimates for effects such as diagenesis, dating uncertainty, the use of extinct species, seasonality effects, etc. We recognize that these additional uncertainties are subjective and hard to quantify, but they are crucially important for giving context to model-data comparisons. Calibration uncertainty includes estimation of poorly constrained parameters,

especially the chemical composition ($\delta^{18}O$, Mg/Ca) of ancient seawater. New approaches are moving towards a more comprehensive treatment of uncertainty on these parameters, including Monte Carlo methods. Further work on the best practice in this area would be strongly supported within or outside of the current DeepMIP working group. Where data already exist in other databases, a URL or DOI should be provided (e.g. PANGAEA). A current (but incomplete) overview of datasets already imported is available at https://paloz.marum.de.

### 7.2 Data management, and scientific workflow development

In addition to the database system described above, we have also designed a scientific workflow system, which allows for full reproducibility of all data manipulation steps and to allow for repeated re-analysis of derived data, such as applying alternative age models or computational steps for calculations as described earlier. Specifically, the chosen system is based on an existing

open-source system (https://kepler-project.org). In this system, individual parts of a complete scientific workflow are connected through so-called Actors, where each actor performs a specific task in a sequence of steps, connected through inputs and outputs. We have augmented the Kepler system with specific Actors that implement, for example, data splicing, data correlation, data visualization, and the ability to apply domain-specific tools such as Astrochron (Meyers, 2014) or the running of arbitrary scripts in Matlab, R, Python or other scripting tools. Individual data analysis steps can then be saved as a project

and re-applied by the scientific community while easily changing specific analysis parameters. These project files will be separately archived and curated, providing an easily accessible archive of DeepMIP data analysis efforts, ideally with a citable and versioned DOI reference.

### 8 Preliminary synthesis

### 8.1 Comparisons between climate proxies and time slices

The proxies for atmospheric $CO_2$ exhibit considerable variability through the early Cenozoic and uncertainty is high for individual measurements, well above 100-200 ppm in most cases (Fig. 6). However, LOESS smoothing by Foster et al. (2017) produced a long-term trend that is broadly consistent with the temperature trend recorded in benthic foraminiferal $\delta^{18}O$, with marine-based $CO_2$ proxies generally above the LOESS curve and the terrestrial proxies on or below the curve. Taken together

it appears that background $CO_2$ levels are ~500 ppm in the latest Paleocene and rise to ~1000 ppm during the EECO whereas the PETM is associated with a short-lived rise in $CO_2$ to ~2000–3000 ppm. Thus, we can have some confidence that our three time slices allow us to investigate two types of $CO_2$-modulated global warming scenarios: rapid warming associated with the PETM and longer term warming associated with the EECO. A finer determination of $CO_2$ during our time slices is however

clearly desirable and a number of community efforts are underway to achieve this (e.g. www.p-CO2.org), including a number of new data generation efforts (www.deepmip.org/sweet/).

Preliminary illustrations of the distribution of SST and LAT data by latitude are shown in Figs. 8 and 9. These figures serve to represent the overall data coverage and the level of agreement between proxies. The data points plotted are mean or median values for samples that span the three time slices. Error bars are 95% confidence intervals for sample ensembles unless noted otherwise. Full sample suites, analytical results, temperature determinations and summary metrics are provided in Supplementary Data Files 3–7. More detailed data-model comparisons and interrogation of sources of error and uncertainties will be the focus of future work.

As expected, the SST compilation shows that temperature estimates derived from the $\delta^{18}O$ values of recrystallized foraminifera are significantly cooler than estimates derived from the $\delta^{18}O$ of well-preserved foraminifera, foraminiferal Mg/Ca ratios, and clumped isotope values from larger benthic foraminifera (Fig. 8a). This is considered to reflect diagenetic modification of foraminiferal calcite. Estimates derived by SIMS measurements of foraminiferal $\delta^{18}O$ at central Pacific ODP Site 865 (Kozdon et al., 2011, 2013) are more in line with other proxies for the LP and PETM time slices. However, a large offset between SSTs derived from SIMS and Mg/Ca for the EECO suggests that the SIMS approach cannot always counter the effects of recrystallisation. Another large offset between SSTs derived from the $\delta^{18}O$ values in well-preserved foraminifera in the New Zealand Hampden section and those derived from Mg/Ca at the same section, as well as other New Zealand sections, is harder to explain (Hollis et al., 2012; Hines et al., 2017). The sample from which the $\delta^{18}O$ measurements were obtained is slightly younger (lower Heretaungan New Zealand Stage ~49 Ma) than the other samples (upper Mangaorapan, ~50 Ma) and possibly records post-EECO cooling at this site or, alternatively, local hydrological factors at this coastal site may have affected the $\delta^{18}O$ values. A similarly large offset between the two clumped isotope records from Belgium – 29.8°C in Ampe Quarry but only 20°C in Kester borehole (Evans et al., 2018a) – is also noted. In other respects, the carbonate proxies exhibit a relatively consistent trend for all time slices: a shallow latitudinal gradient of <5°C for the Southern Hemisphere (to ~55°S) and a somewhat steeper gradient of 10–15°C in the Northern Hemisphere (to ~45°N). Carbonate proxy data are too sparse to generalize temperature differences between the time slices.

When TEX$_{86}$-derived SSTs (BAYSPAR$_{SST}$ calibration) are added to the compilation (Fig. 8b), the range in SST increases markedly: the Southern Hemisphere gradient increases to >10°C and the Northern Hemisphere gradient increases to >15°C. There is also a distinct separation between the time slices: the PETM is ~5–10°C warmer than the LP and the EECO tends to be >5°C warmer than the LP, at least in middle–high latitudes. Interestingly, TEX$_{86}$ values for the LP and EECO time slices overlap in low-latitude sites. Average SST values based on TEX$_{86}$ tend to be ~5°C warmer than estimates derived from carbonate proxies at the same or nearby sites, notably in New Zealand and New Jersey. TEX$_{86}$-derived SSTs for low-latitude sites are in better agreement with carbonate proxies, especially for the EECO.

Data coverage for land air temperature (LAT) is reasonable for high–middle southern and northern latitudes (Fig. 9) but there are only four low-latitude records, all in India. Available datasets for the PETM and LP are too locally restricted to observe any latitudinal patterns other than to note general similarities with the EECO. These EECO records exhibit a pattern similar to the SST record; in general LAT estimates are warmer in southern mid-latitudes than in northern mid-latitudes. However, there is considerable variability within estimates based on leaf physiognomy, some of which relates to the wide variety of methods employed. The nine different methods used for early Eocene leaf fossil assemblages range from LMA and CLAMP (each with multiple calibrations) to single or multiple linear regressions, and other statistical manipulations of leaf shape and size. Some methods consistently yield higher LATs than others (e.g. LMA using the Kowalski and Dilcher [2003] calibration) but there is as yet no consensus on a preferred methodology. Moreover, considerable local variability is also evident with the MBT/CBT proxy. As discussed previously, age control is poor in some terrestrial sections and some of the variability may relate to this; some sites may prove to be outside the EECO time slice.

Most of the proxies indicate that LAT ranges from ~5–10°C in high latitudes to ~25–28°C in middle and low latitudes during the EECO, albeit with only two low-latitude sites to constrain the latitudinal range from 30°S to 30°N. LATs derived from paleosol or mammal $\delta^{18}O$ lie at the lower end of this range at several sites, notably Salta Basin, Argentina (Hyland et al., 2017), Wind River Basin, Wyoming (Hyland et al., 2013) and Ellesmere Island (Fricke and Wing, 2004). This suggests that these proxies may suffer from a cool bias in some circumstances. Conversely, LAT estimates from clumped isotopes are much higher than this range, which is consistent with the interpretation that they represent summer temperatures, at least in the Bighorn Basin (Snell et al., 2013). Low- and middle-latitude EECO records for the new brGDGT-based proxy ($LAT_{PEAT}$) also yield LATs significantly higher than other LAT estimates from the same region or similar latitudes (Naafs et al., 2018b). In New Zealand, a $LAT_{PEAT}$ value of ~28°C at Otaio River contrasts with LATs of 16–20°C based on NLR at nearby mid-Waipara River. In the Northern Hemisphere, a $LAT_{PEAT}$ value of ~27°C from the Schöningen Lignite, Germany, is 2°C and 5°C warmer than the warmest LAT estimates derived by MBT'/CBT and LMA, respectively, at an equivalent latitude. These differences are within the error ranges for the proxies and this $LAT_{PEAT}$ estimate is consistent with a LAT estimate of 28°C derived from mammal $\delta^{18}O$ from a site in the San Juan Basin, New Mexico (Fricke and Wing, 2004). It is also consistent with the summer temperatures inferred for the Bighorn Basin (Snell et al., 2013). This agreement between proxies indicates that the $LAT_{PEAT}$ proxy may help to resolve the long-standing issue of an apparent cool bias in other LAT proxies (Huber and Caballero, 2011; Pancost et al., 2013; Naafs et al., 2018b). Moreover, the two $LAT_{PEAT}$ values for the EECO are within the range of SST estimates for the EECO (Fig. 9).

For both SST and LAT estimates, the use of averaged values for the three time slices requires a cautionary note. Average values inevitably dampen the temperature signal and have potential to obscure the true scale of warming, especially in the PETM and EECO. For planktic foraminifera, Pearson et al. (2007) argued that the specimens with the most negative $\delta^{18}O$

values most likely represented the seasonal maximum SST for Tanzania. Similarly, the potential for seasonal bias in many of the temperature proxies considered here needs to be considered for all middle- to high-latitude sites.

**9 Conclusions and priorities for new data gathering**

We have reviewed the primary ways in which temperature and atmospheric $CO_2$ may be reconstructed in the early Cenozoic. These proxies are in widely different stages of maturity, with several proxies in need of concerted effort to improve their reliability and reduce uncertainties. After considering the strengths and weaknesses of each of these climate proxies, we have provided recommendations on their future use, both in general terms and specifically in the context of the DeepMIP objective to reconstruct climate across three time slices: the latest Paleocene (LP), Paleocene-Eocene thermal maximum (PETM) and

early Eocene climatic optimum (EECO). We applied these recommendations in a preliminary way to a compilation of existing sea surface and land air temperatures (SST and LAT).

We have determined that existing data coverage is very patchy and that there are considerable inconsistencies between temperature proxies. The same is true for the available $CO_2$ proxies. Over the past few years, researchers associated with the

DeepMIP project have made great advances in recovering paleotemperature records from poorly sampled regions, both in high (Frieling et al., 2014; Hollis et al, 2015; Schoon et al., 2015; Hines et al., 2017) and low latitudes (Inglis et al., 2015; Frieling et al., 2017, 2018; Cramwinckel et al., 2018; Evans et al., 2018a; Naafs et al., 2018a). Several of these studies have applied new methodologies to deep sea cores collected over 20 years ago (Hollis et al., 2015; Inglis et al., 2015; Cramwinckel et al., 2018). There is huge potential to apply new methods in $\delta^{18}O$ and Mg/Ca analysis (Kozdon et al., 2011, 2013; Evans and Müller,

2012) to many other cores in our database that so far have only been analyzed by conventional $\delta^{18}O$ methods. There is also value in searching for new continental margin records in poorly sampled regions, where multi-proxy approaches to climate reconstruction can be applied (e.g. Sluijs et al., 2014; Frieling et al., 2017). For such records, the value of improving on existing calibrations for branched and isoprenoidal GDGTs (MBT–CBT and $TEX_{86}$) present in marine sediments cannot be overestimated as they have the potential to provide coupled SST and LAT records in well dated successions that can be

compared with other proxies. For MBT–CBT, the challenges are to address the issue of calibration saturation at the upper temperature range and the effects of mixed sources of brGDGTs. For $TEX_{86}$, we have shown that the preferred alternative approach to BAYSPAR, $TEX_{86}^H$, is poorly formulated mathematically. It lacks a physiological basis as there is little evidence for the exponential increase in archaeal cyclopentane moieties in the upper temperature range, which is implicit in the calibration (Fig. 4a). Therefore, the challenge for $TEX_{86}$ is to understand why BAYSPAR temperature estimates are

consistently higher than other proxies in middle and high latitudes (Fig. 8b).

For terrestrial records, a major challenge is to improve age models through integrated stratigraphy – utilizing miospore biostratigraphy, carbon isotopes and magnetostratigraphy. This is a long-term objective, requiring significant taxonomic effort.

Researchers associated with the DeepMIP project have also generated several new $CO_2$ proxy records for the studied time intervals (Anagnostou et al., 2016; Foster et al., 2017). As with the temperature proxies, there is clearly value in finding locations where multiple $CO_2$ proxies can be applied to add confidence in the data generated. Work should also focus on reducing the structural uncertainty in the associated proxies (e.g., vital effects for $\delta^{11}B$).

The DeepMIP database (v. 0.1) is the first step towards a comprehensive compilation of climate proxy data for the latest Paleocene, PETM and EECO. In its current form, it improves upon previous datasets that have been used to identify areas of agreement as well as mismatches between Paleogene climate data and climate models (e.g. Lunt et al. 2012; Huber and Caballero, 2011). With further interrogation, using methodologies similar to those of Caballero and Huber (2013), we aim to

10 derive estimates for global mean sea surface, land surface and overall surface temperature for the three time slices, which will help to advance our understanding of climate sensitivity under high $CO_2$ conditions. The next challenge is then to add the full range of available climate data (e.g. salinity, precipitation, humidity, seasonal variability) to the DeepMIP database. With a recent study indicating that high growth/low mitigation scenarios herald a return to early Eocene-like conditions by the end of the century (Burke et al., 2018), the goals of DeepMIP are increasingly important and timely.

**Author contributions**

CJH and TDJ compiled the text; CJH prepared Figs. 1-4 and 8–9; KME, JCZ, PP, BW, RK prepared the text on oxygen isotopes; DE, CHL, RK and AT prepared the text on Mg/Ca ratios; DE, AT and NM prepared the text on clumped isotopes; GNI, JT, AS, JF, MC, PKB, KL and RDP prepared the text on $TEX_{86}$; BDAN and RDP prepared the text on brGDGTs and

20 Fig. 5; EMK and UZ prepared the text on plant fossil-based temperate proxies; GLF, EA, YC, YE, DLR, BS, CW and YGZ prepared the text on CO2 proxies; GLF prepared fig. 6; HS compiled the paleolocations for all sites and prepared Fig. 7; TW, UR, LL, VL and KL contributed to the text on geochronology; HP prepared the text on the DeepMIP database; CJH, TDJ, KME, DE, GNI, JT, BDAN, EMK, UZ and GLF compiled the data files; many of the above in addition to GRD, RPS, PS, MH and DL contributed to the overall text.

**Competing interests**

The authors declare that they have no conflict of interest.

**Data availability**

30 The data referenced in this manuscript are provided as Supplementary Data Files 1 to 8. In the final version, these files will form DeepMIP database version 0.1 and will be accessible online via a citable DOI reference.

**Acknowledgements**

DeepMIP is supported by the UK government through NERC grants NE/P01903X/1 and NE/N006828/1. Individual contributors acknowledge support from the following grants and agencies: CJH, EMK and HS (GNS Science Global Change through Time Programme); TDJ, KME and YE (NERC grant NE/P013112/1); CL (NERC grant NE/P019102/1); YC and BAS (NSF EAR #1603051); RPS and PS (KU Leuven BOF #3E170442). Several datasets utilized in this study were uploaded from the World Data Center, PANGAEA, and are based on samples collected and provided by the International Ocean Discovery Program (IODP). The authors acknowledge helpful reviews of the manuscript by Erica Crouch, Sebastian Naeher, David Greenwood, Marci Robinson and Navjit Sagoo.

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

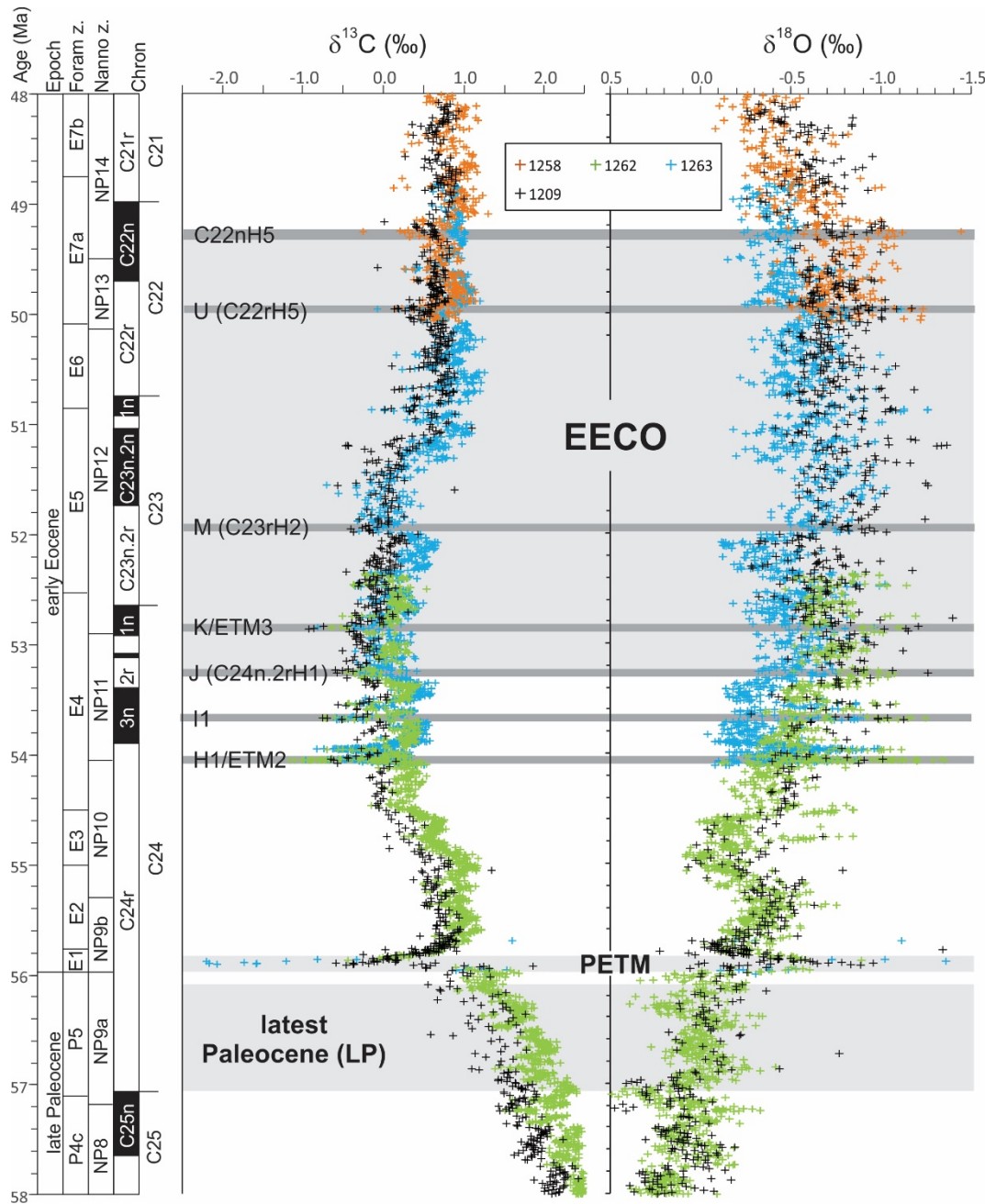

**Figure 1: Benthic foraminiferal carbon and oxygen stable isotope records from ODP sites 1209, 1258, 1262 and 1263 (Westerhold et al., 2011, 2017, 2018; Littler et al., 2014; Lauretano et al., 2015, 2018; Barnet et al., 2019) calibrated to the timescale of Westerhold et al. (2017). Calcareous nannofossil and planktic foraminiferal biozone boundaries are recalibrated from Gradstein et al. (2012).**

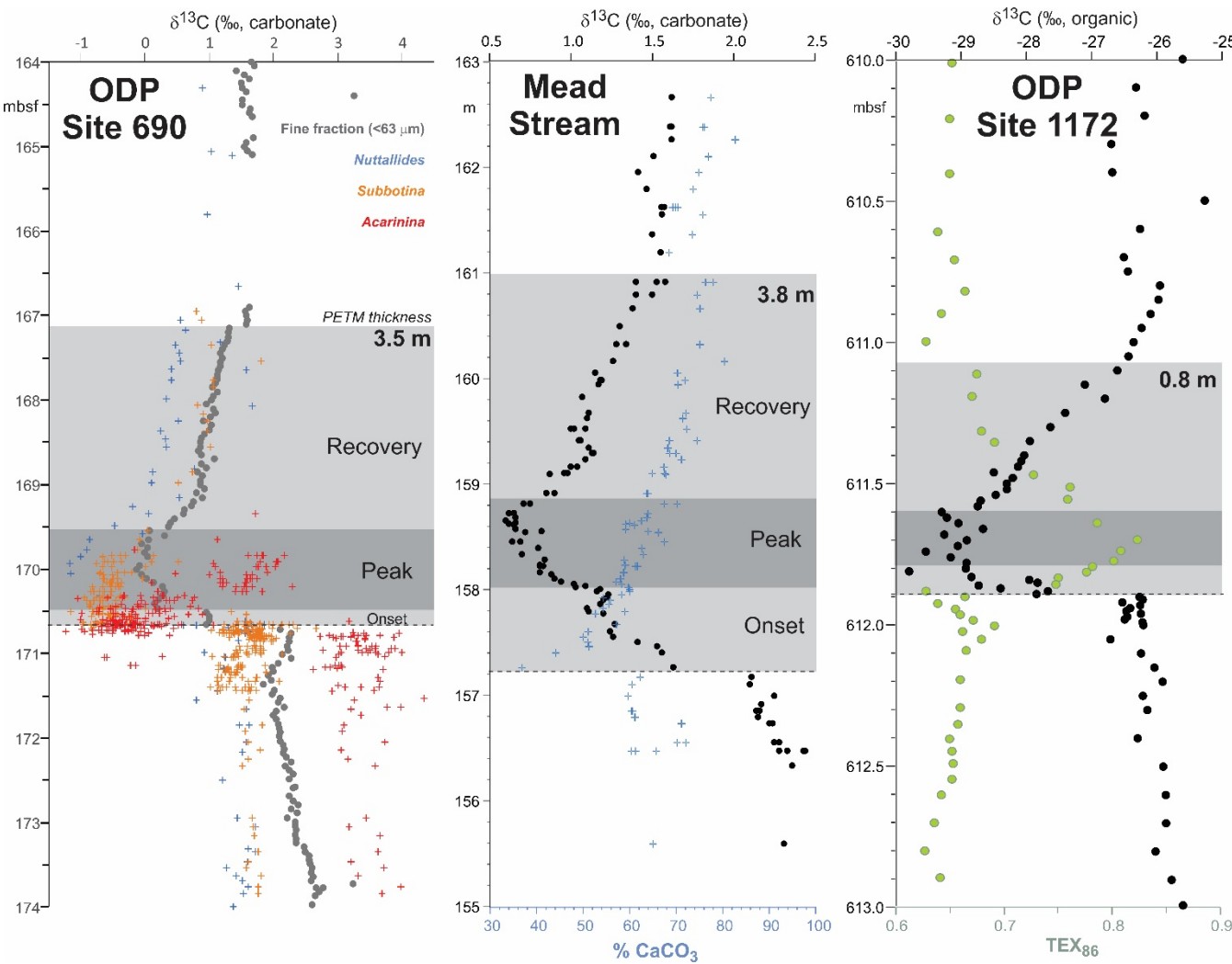

Figure 2. Three high-resolution records through the Paleocene-Eocene thermal maximum (PETM): ODP Site 690, Maud Rise, South Atlantic (Kennett and Stott, 1991; Bains et al., 1999; Thomas et al., 2002; Nunes and Norris, 2006); Mead Stream, New Zealand, South Pacific (Nicolo et al., 2010); ODP Site 1172, East Tasman Plateau, Tasman Sea (Sluijs et al., 2011).

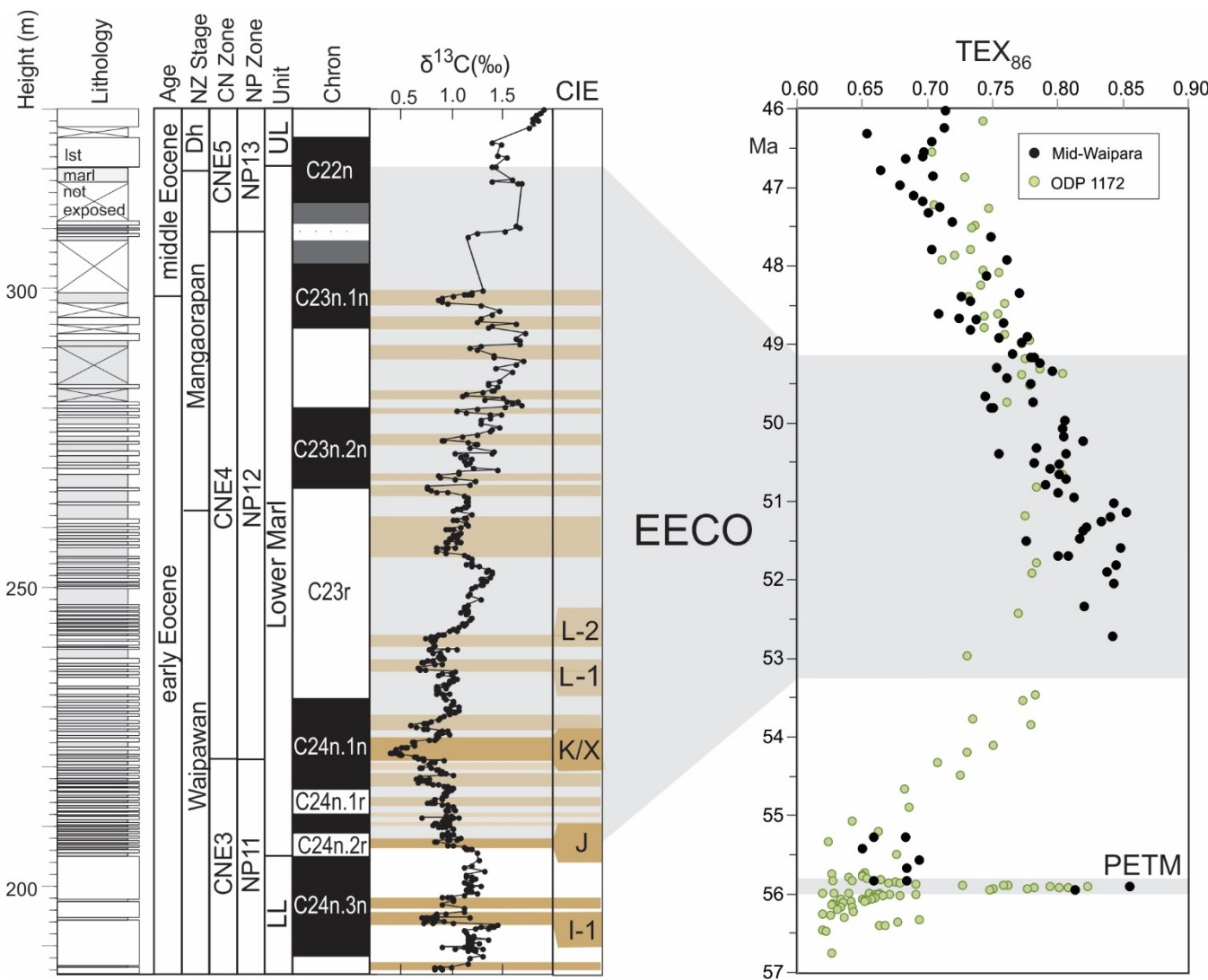

**Figure 3. Eocene carbon isotopes and lithostratigraphy at Mead Stream, New Zealand (Slotnick et al., 2012), compared with indicators of relative changes in sea surface temperatures: TEX$_{86}$ for ODP Site 1172 (Bijl et al., 2009) and mid-Waipara (Hollis et al., 2012; Crouch et al., submitted.). Grey shading = EECO interval as defined by Westerhold et al. (2018).**

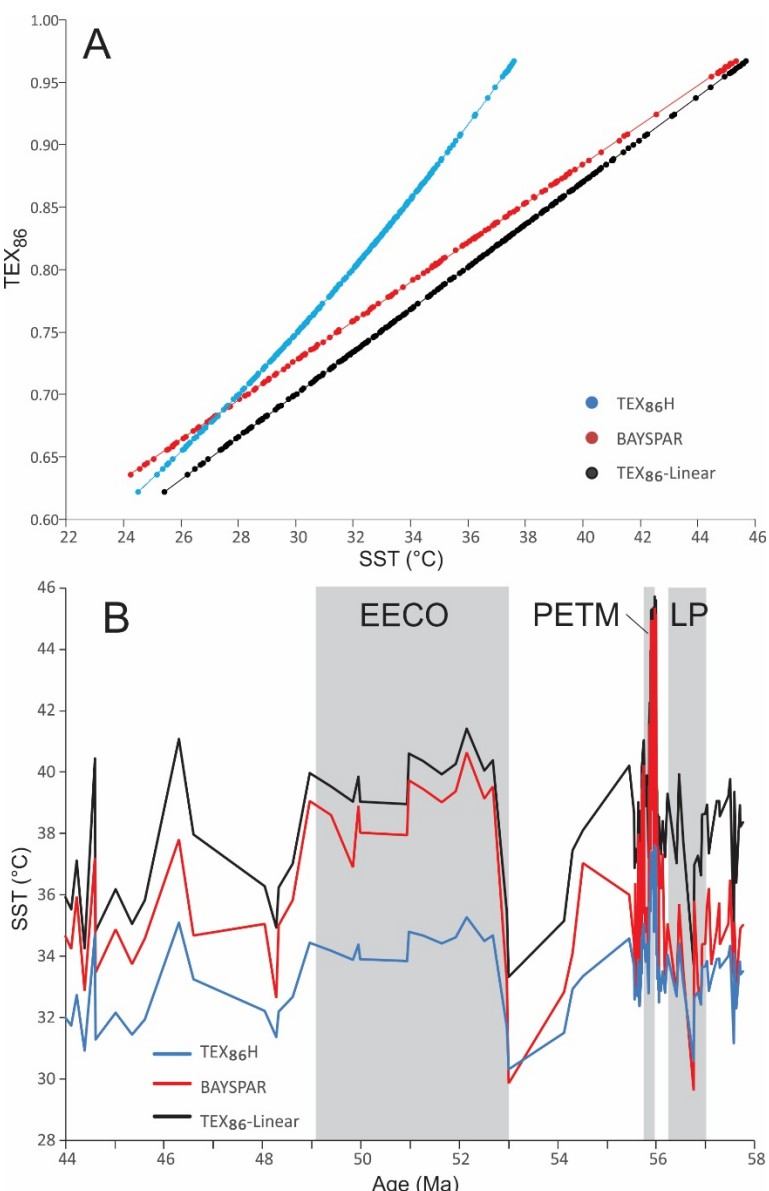

**Figure 4: Relationships between TEX$_{86}$ calibrations and sea surface temperature for low-latitude Paleocene-Eocene sites (data from Cramwinckel et al., 2018). A) Comparison of the exponential TEX$_{86}$$^H$ calibration (Kim et al., 2010), the linear TEX$_{86}$ (O'Brien et al., 2017) and BAYSPAR calibrations (Tierney and Tingley, 2014, 2015). Because of the spatially varying regression used for BAYSPAR only one site (ODP Site 959) is plotted for this calibration. B) Comparison of SST reconstructions from late Paleocene to early middle Eocene for the three calibrations. The timescale of Gradstein et al. (2012) is used in this figure. Approximate locations of the three targeted time intervals are also shown: latest Paleocene (LP), Paleocene-Eocene thermal maximum (PETM) and early Eocene climatic optimum (EECO).**

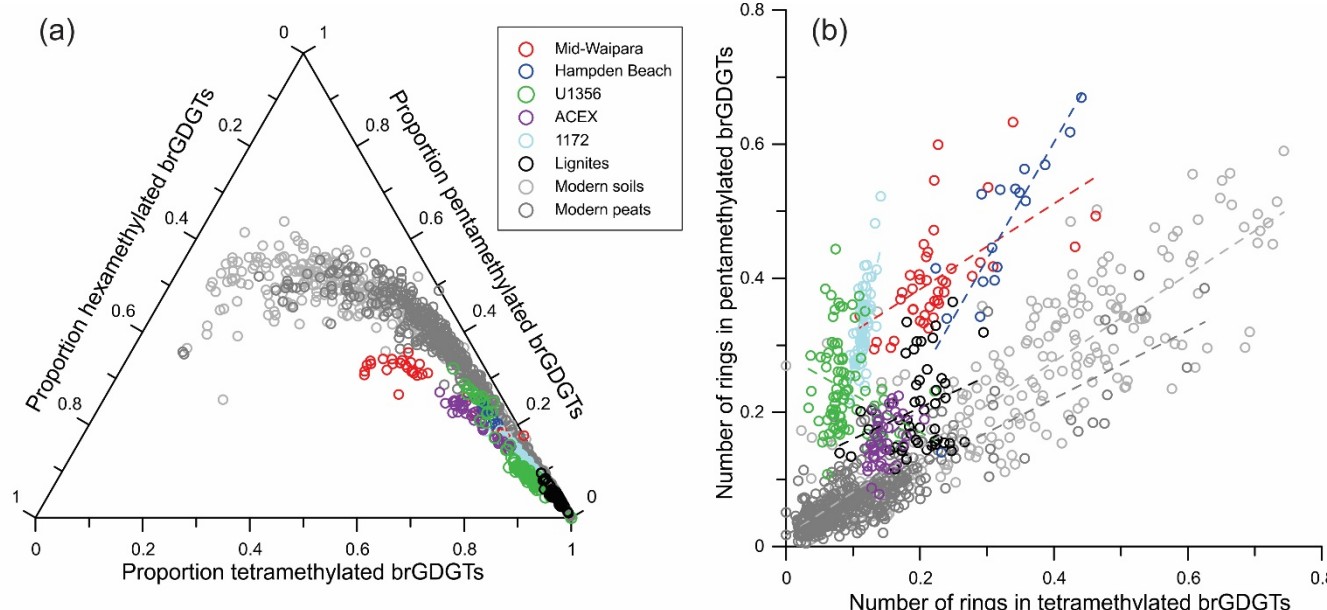

**Figure 5. Distribution of brGDGTs in present-day mineral soils (Naafs et al., 2017a) and peats (Naafs et al., 2017b), compared to that in early Paleogene marine sediments (Weijers et al., 2007; Bijl et al., 2013; Pancost et al., 2013) and terrestrial lignites (Naafs et al., 2018): (a) a ternary plot showing relative proportions of tetra-, penta- and hexamethylated brGRGDTs; (b) a cross plot showing the average number of cyclopentane moieties (rings) for tetra- and pentamethylated brGDGTs.**

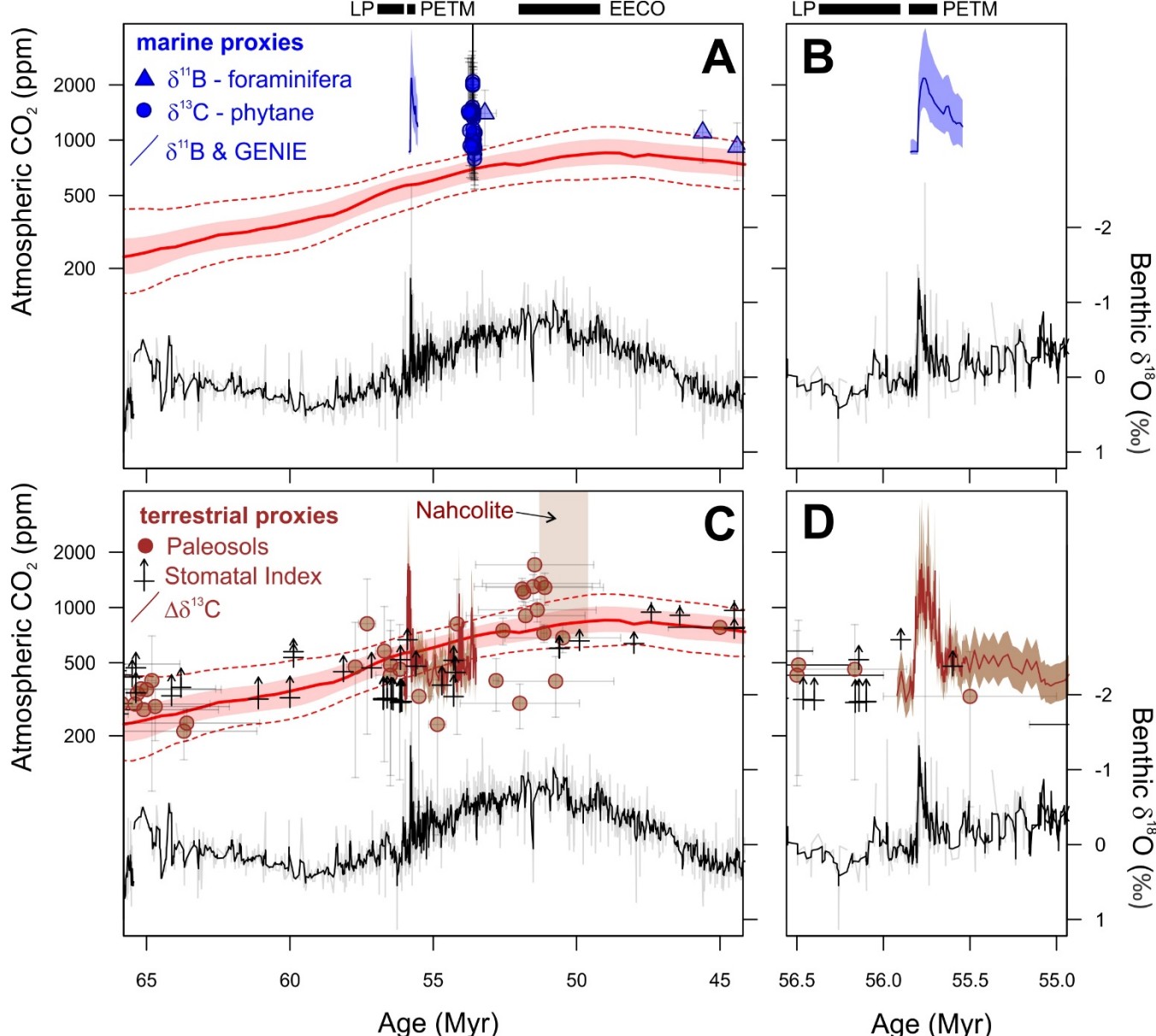

**Figure 6.** Reconstruction of atmospheric CO₂ from marine (A and B) and terrestrial (C and D) proxies for the early Paleocene to early middle Eocene (A and C), and an expanded reconstruction across the Paleocene-Eocene boundary (B and D). The CO₂ estimates plotted here are summarised with appropriate references in Supplementary Data File 8. The red line is a LOESS smoother from Foster et al. (2017), the red band is 68% confidence and the dashed red lines are 95% confidence around that long-term trend. The timescale of Gradstein et al. (2004) is used in this figure. Approximate locations of the three targeted time intervals are also shown: latest Paleocene (LP), Paleocene-Eocene thermal maximum (PETM) and early Eocene climatic optimum (EECO). Note that stomatal index and the nahcolite are plotted with an unbounded upper CO₂ and so represent minimum estimates. See the text for a discussion of the methodologies used.

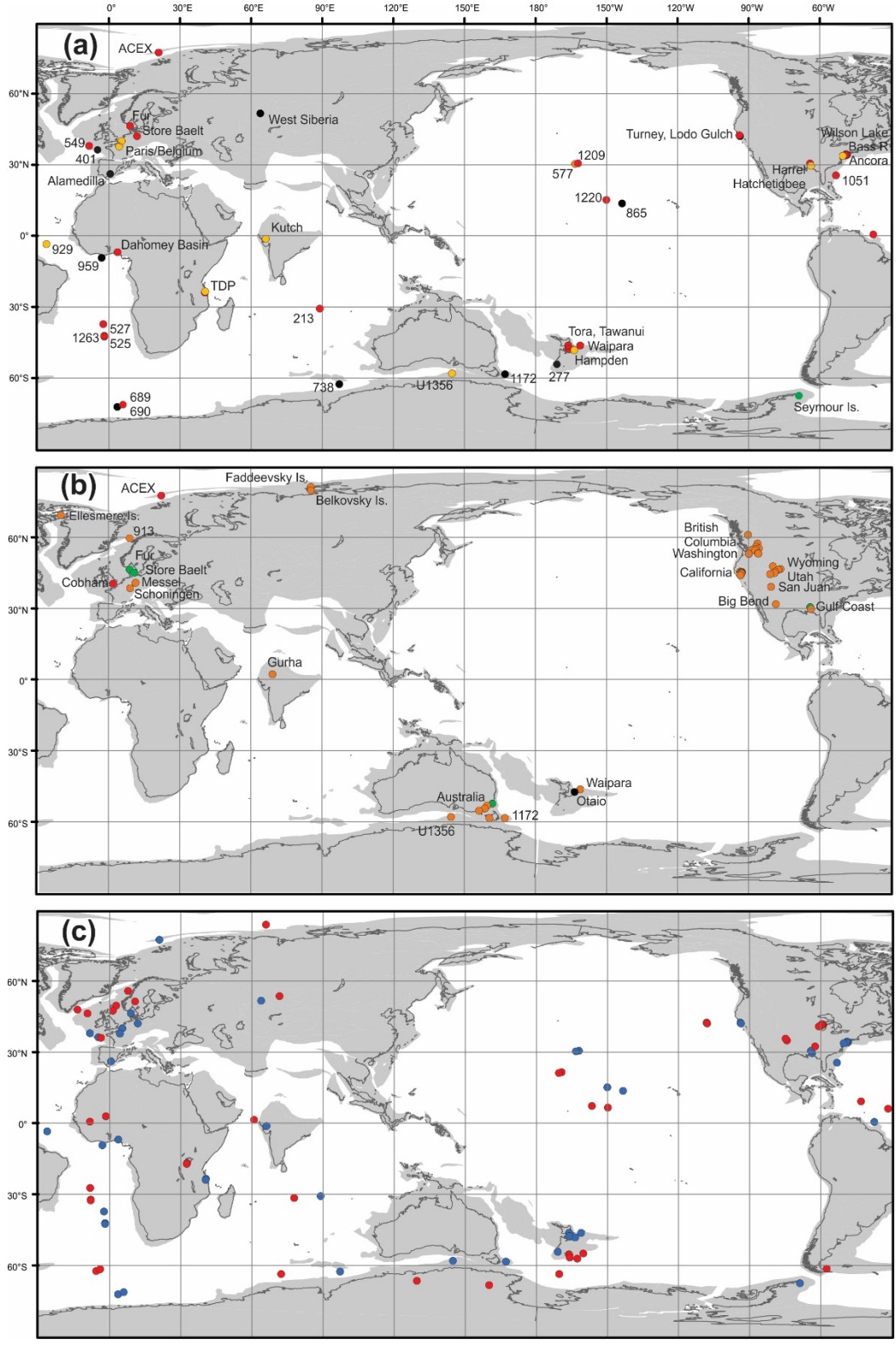

**Figure 7. Sites with latest Paleocene–early Eocene temperature estimates on a 52 Ma paleogeographic reconstruction using relative plate motions from Matthews et al. (2016) with the paleomagnetic reference frame of Torsvik et al. (2012). (a) Marine sites with time intervals indicated as follows: black circles, all three time intervals represented; red circles, PETM ± LP intervals; orange circles, EECO interval; green circle, middle Eocene (Seymour Island). (b) Terrestrial sites with time intervals indicated as in (A) and green circles, LP only. (c) Comparison between reference frames; blue circles are marine sites plotted using the same paleomagnetic reference frame as in (A); red circles are plotted using a moving hot-spot reference frame (Matthews et al., 2016). Sites locations and paleolocations are tabulated in Supplementary Data File 2.**

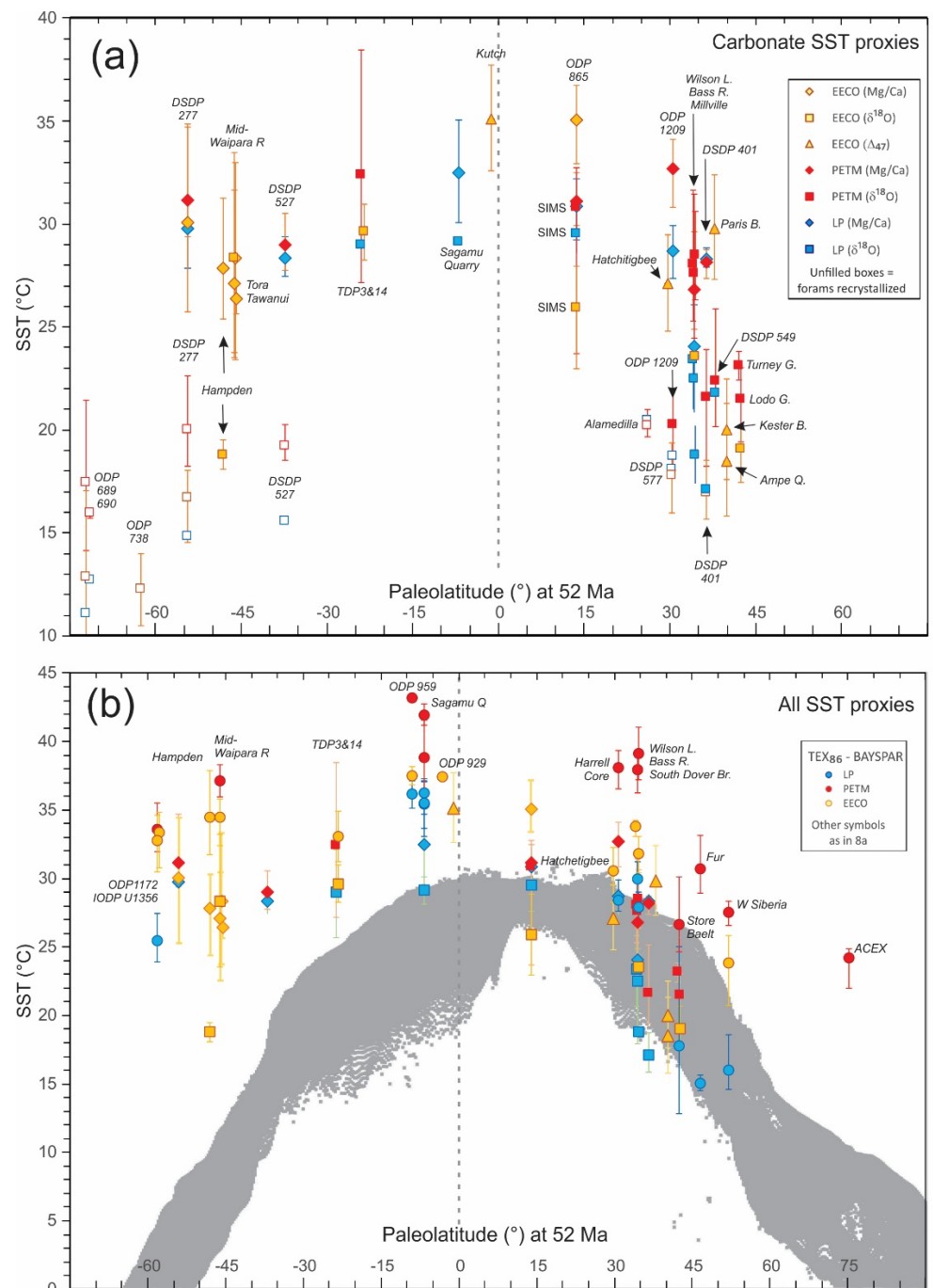

**Figure 8. SST estimates derived from (a) carbonate proxies and (b) all proxies for three time slices: latest Paleocene (LP), PETM and EECO. Deep sea $\delta^{18}O$ records that are inferred to be affected by diagenesis are omitted from Fig. 8a. These data are tabulated in Supplementary Data Files 3–6. Grey data cloud in Fig. 8b represents modern SST measurements derived from European Centre for Medium-Range Weather Forecasts (2017), ERA5 Reanalysis Monthly Means, doi:10.5065/D63B5XW1, Research Data Archive at the National Center for Atmospheric Research, Computational and Information Systems Laboratory, Boulder, Colo. (Updated monthly.) Accessed 8-11-2018.**

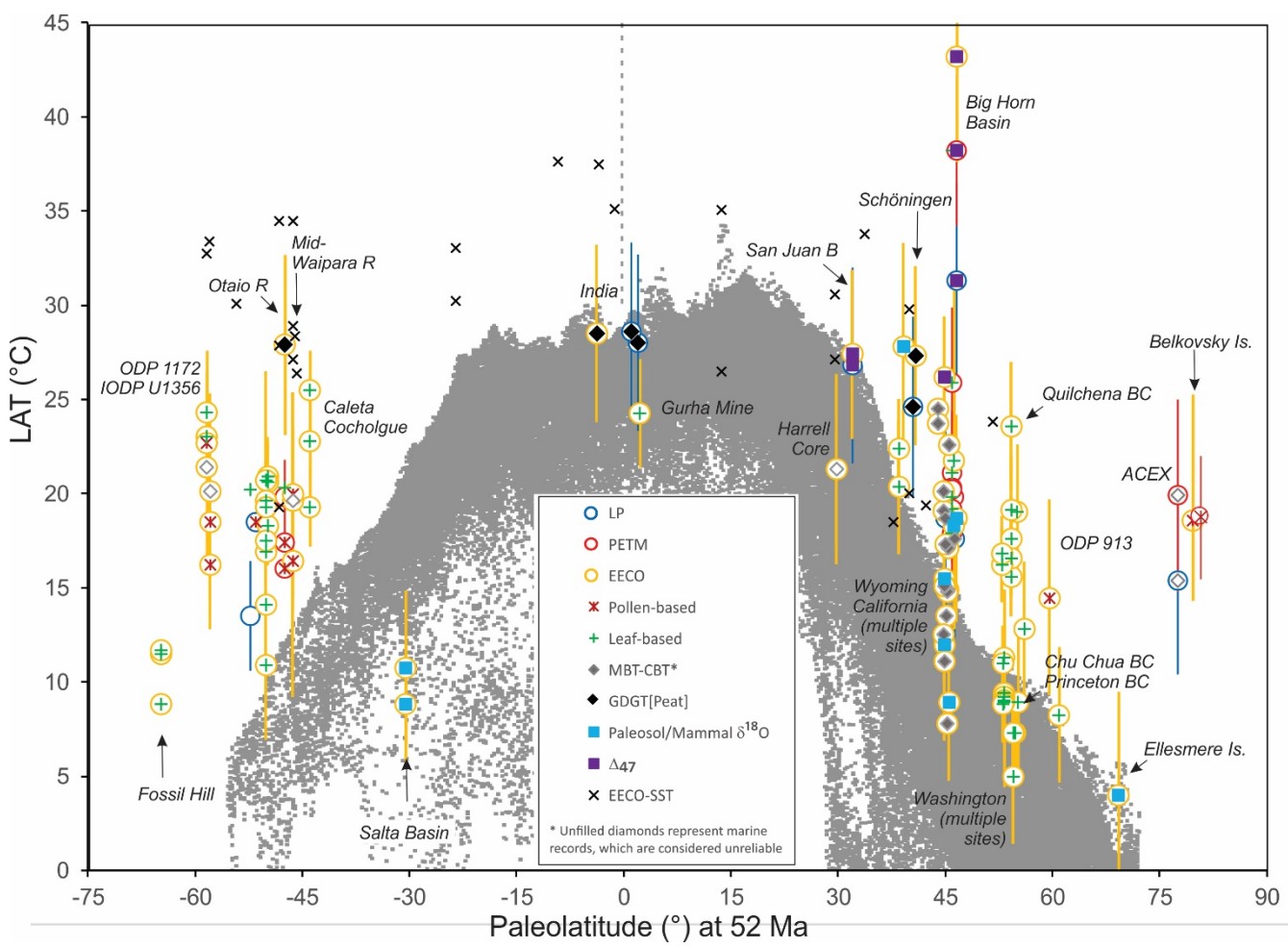

**Figure 9. LAT estimates for the three time slices differentiated by broad proxy categories. These data are tabulated in Supplementary Data File 7. The SST data for the EECO is shown for comparison. Grey data cloud represents modern LAT measurements derived from same source cited for Figure 8b.**

| Proxy | Climate Parameter General | This study | Total | DeepMIP Samples Moderate to high confidence All | LP | PETM | EECO | Low confidence | Comment |
|---|---|---|---|---|---|---|---|---|---|
| Foraminiferal $\delta^{18}$O | Sea temperature | SST | 894 | 507 | 169 | 296 | 42 | 387 | Diagenesis |
| Foraminiferal Mg/Ca | Sea temperature | SST | 321 | 321 | 124 | 130 | 67 | 0 | |
| Clumped isotopes ($\Delta47$) | Sea temperature | SST | 7 | 7 | 0 | 0 | 7 | 0 | |
| TEX$_{86}$ (isoGDGTs) | SST | SST | 479 | 456 | 140 | 113 | 203 | 23 | High BIT index |
| Subtotal | | | 1701 | 1291 | 433 | 539 | 319 | 410 | |
| brGDGTs (MBT-CBT, MBT'$_{m5e}$) | LAT | LAT | 157 | 73 | 11 | 1 | 61 | 84 | Marine record |
| Leaf Margin Analysis | LAT, Precipitation | LAT | 40 | 40 | 1 | 5 | 34 | 0 | |
| CLAMP | LAT, Precipitation | LAT | 15 | 15 | 0 | 1 | 14 | 0 | |
| Other Leaf-based proxies | LAT, Precipitation | LAT | 16 | 16 | 0 | 2 | 14 | 0 | |
| Nearest Living Relative | LAT, Precipitation | LAT | 198 | 198 | 1 | 9 | 188 | 0 | |
| Mammal/Paleosol $\delta^{18}$O | LAT | LAT | 24 | 24 | 0 | 0 | 24 | 0 | |
| Clumped isotopes ($\Delta47$) | LAT | LAT | 21 | 19 | 4 | 4 | 11 | 2 | Diagenesis |
| Paleosol Climofunction | LAT, Precipitation | LAT | 39 | 39 | 0 | 0 | 39 | 0 | |
| Subtotal | | | 510 | 424 | 17 | 22 | 385 | 86 | |
| Boron isotopes | Atmospheric $CO_2$ | $CO_2$ | 44 | 44 | na | na | na | 0 | |
| Phytane | Atmospheric $CO_2$ | $CO_2$ | 34 | 34 | na | na | na | 0 | |
| C$_3$ land plants | Atmospheric $CO_2$ | $CO_2$ | 341 | 341 | na | na | na | 0 | |
| Leaf stomata | Atmospheric $CO_2$ | $CO_2$ | 38 | 38 | na | na | na | 0 | |
| Pedogenic carbonate | Atmospheric $CO_2$ | $CO_2$ | 82 | 82 | na | na | na | 0 | |
| Subtotal | | | 539 | 539 | | | | | |
| TOTAL | | | 2750 | 2254 | 450 | 561 | 704 | 496 | |

**Table 1. Number of sea surface temperature (SST), land air temperature (LAT) and atmospheric CO₂ estimates in the DeepMIP database (v. 0.1), including the number of samples in each time slice which meet criteria for use as temperature estimates (moderate to high confidence) and low confidence temperature estimates, which are either included data summaries with caution (diagenesis, calibration uncertainty) or excluded (high BIT index).**