# Peer review of "The DeepMIP contribution to PMIP4: methodologies for selection, compilation and analysis of latest Paleocene and early Eocene climate proxy data, incorporating version 0.1 of the DeepMIP database"

_Geoscientific Model Development, 2018_

## Referee Comment (RC1) · David Greenwood (Referee) · 15 Feb 2019

Hollis et al. 2019. The DeepMIP contribution to PMIP4: methodologies for selection, compilation and analysis of latest Paleocene and early Eocene climate proxy data, incorporating version 0.1 of the DeepMIP database. Geosci. Model Dev. Discuss.

1. General comments.

The paper by Hollis et al. provides a well written summary of the proxy methods used to derive past climate states and greenhouse gas levels from the geological record – specifically for the latest Paleocene (LP), Paleocene-Eocene thermal maximum (PETM) and early Eocene climatic optimum (EECO). The focus on this geological interval is appropriate - as stated by the authors – as at around 56 Ma the world experienced a geologically short ($\sim$220 ky) and strong warming, the PETM, that was followed by a series of additional hyperthermals (e.g., ETM-2) and the longer duration EECO from $\sim$53 to $\sim$50 Ma. The PETM in particular offers a possible analog for current and forecast climate warming due to its short duration and the range of pCO2 reconstructed for the PETM spanning current to projected greenhouse gas levels under different projections.

The authors provide a comprehensive yet succinct review of both paleontological proxies as well as geochemical proxies of temperature of waters and of the air, past CO2 levels, as well as a range of paleoclimate parameters including annual and seasonal precipitation based primarily on terrestrially derived proxies. Further, their review highlights the challenges – strengths and weaknesses – posed by such a wide range of methods and approaches, ranging from concerns over precision versus accuracy, alternative methods of analysis and interpretation, taphonomy, limited geographic coverage, and the issues around compiling such disparate data into a database that can be used by the paleoclimate modelling community (model-model and model-data comparisons). As such, the article largely succeeds, offering a set of guidelines – as the article states – for the selection, compilation and analysis of such proxy data.

The article closes with a proposal for an "atlas" of climate conditions arrayed on an agreed paleogeography for the three selected time intervals to constrain and provide insight into the mechanisms controlling past hyperthermals via 'database ver. 0.1'. The article draws on the expertise of a large set of authors – the DeepMIP team – whose research foci span these topics. The manuscript therefore represents a substantial contribution to modelling science and is appropriate for the journal. I have few

concerns, and these mainly reflect topics dealt with in less depth than I would consider necessary, or restricting discussion to particular research teams, as detailed in the next section. In my specific comments I have focused on my area of expertise; paleobotanical reconstruction of past pCO2 and climate, so "5 Terrestrial proxies for air temperature."

2. Specific comments

Section 5 in its title only states 'Terrestrial proxies for air temperature', when this section also considers precipitation (e.g., p. 31 line 24, in section 5.2 or in section 5.3 at line 15 on p. 32), and should be re-titled as 'Terrestrial proxies for climate'. Further, as leaf physiognomic methods as well as NLR approaches have been applied to reconstructing latest Paleocene to EECO precipitation (e.g., Greenwood et al. 2010; Pross et al. 2012; Eldrett et al. 2014; West et al. 2015; Suan et al. 2017; Hyland et al. 2018), the leaf-based and NLR proxies for precipitation should be included, e.g., CLAMP estimates growing season precipitation as well as that of the wettest 3 months and the driest 3 months, but these are not differentiated on p. 31 line 24, nor discussed at any point in section 5.2.

5.2 Leaf morphology-based approaches. This section is very focused on the univariate leaf margin analysis (LMA) and the multivariate CLAMP, and ignores other methods such as digital leaf physiognomy (DiLP; Peppe et al. 2011), and also (as noted above and below) barely mentions the use of leaf physiognomy for reconstructing precipitation (annual and seasonal values) in PETM and EECO studies (e.g., Leaf Area Analysis [LAA] and DiLP). Ignoring DiLP is unfortunate, but perhaps forgivable as DiLP has been applied to very few Paleogene floras to date. However, LAA (Wilf et al. 1998; Peppe et al. 2011) has been applied to a wider set of floras, including LP to EECO floras (e.g., Greenwood et al. 2010; Sunderlin et al. 2011; Smith et al. 2012; West et al. 2015; Lowe et al. 2018) and – as noted – yields data on precipitation that is critical to the DeepMIP project. The discussion of weaknesses and strengths in this section is also out of date as it misses current and recent literature on LMA and CLAMP. Citing

Carpenter et al. (2012) at line 22, p. 30 is idiosyncratic as an example as this paper proposed a method that has been never used in another study, whereas LAA (Wilf et al. 1998) is cited, but this univariate method's application to estimate Paleogene annual precipitation is not mentioned, despite LAA being used in multiple studies of the LP to EECO interval.

5.3 Nearest Living Relative Analysis. I disagree that the 'most widely used method' is the Coexistence Approach (CA), particularly when applied to the LP to EECO interval of concern. By focusing on a method that has been widely criticized – and further is not the most widely used NLR-proxy used in existing studies of the PETM and EECO – the authors create problems rather than solve them. My recommendation (as argued below) is that CA should not be the focus of recommendations.

Grimm and Denk (2012) and Grimm and Potts (2016) posed useful concerns about the analytical approach that underpins the 'coexistence approach' as employed by the PALAEOFLORA group. Principally CA's reliance on single taxa to define upper and lower limits.

Grimm directs deep time climate researchers towards statistical methods such as CR-ACLE (Harbert and Nixon 2015) and an earlier iteration of my bioclimatic approach (Greenwood et al. 2005; see also Ballantyne et al. 2010) where we took a more objective probability based approach to either remove outliers, or to produce an estimate that reflected the highest probability solution for the suite of taxa present in a fossil assemblage. A nice succinct summary of these concerns and the more appropriate probability based approaches is given by Hyland et al. (2018).

In essence the concerns are that:

1) genera that are low species richness today (e.g., Ginkgo, taxodioid Cupressaceae such as Glyptostrobus, Metasequoia and Sequoia, etc) may be restricted today in their climate range for reasons other than climate, and may have occupied wider (or just different) climatic spaces in the Paleogene when they were clearly far more geographically widespread and so potentially more ecologically varied than today – this argues against using single taxa such as Ginkgo or Metasequoia (both of which are commonly present in LP, PETM and EECO macrofloras) to define hard limits;

2) that climatic tolerance may have evolved in some lineages – e.g., palms are often used in Paleogene hyperthermal studies to constrain winter temperatures, however pre-Eocene palms likely didn't include the most cold-tolerant subtribes present today as they were a late Eocene or even Miocene radiation (Reichgelt et al. 2018 and references therein);

3) issues with identification of the NLRs – plant taxa may show the same leaf or pollen morphology in Eocene fossils as today, but come from plants that were phylogenetically sister taxa – i.e. not the same genus – with morphologically different flowers or other plant organs to the NLR, so the climate tolerance of the fossil's NLR may be a non-match – e.g., Platanaceae and Betulaceae leaves and pollen present in many North American floras may show 'Platanus' or 'Carpinus' leaf or pollen morphology, but are from plants with extinct taxon flowers and other organs (Macginitea and Palaeocarpinus); methods like CA are inherently subjective because 'outliers' – e.g., those relictual taxa noted in point 1, or that 'sit outside' the climate range of the majority of taxa – are arbitrarily excluded from the CA, or arbitrarily used to set a limit for the range of possible estimates;

4) in the past there were concerns about where NLR climate range data were coming from – a lack of standardization of data sources and data quality control.

Due to all of the above concerns, a number of authors have adopted Grimm's recommendations or independently came to a similar solution (e.g., Ballantyne et al. 2010). For example, the mathematical approach developed in R by my colleague Tammo Reichgelt – a method with comparable assumptions to CRACLE – where we use probability density functions and Monte Carlo runs to select the highest probability climate estimate based on the whole suite of fossil taxa's NLRs, including taxa at different taxonomic ranks (Family, tribe or genus) to allow for differing degrees of confidence of matching fossils to an extant NLR.

Furthermore, Greenwood et al. (2017), Hyland et al. (2018) and Reichgelt et al. (2018) and others derive the NLR climate range data from the international online portal GBIF.org which warehouses distribution data from most of the world's university, museum and government herbaria and equivalent, with clear policies on data quality and ownership, so verifiable data. The GBIF records are not without bias (geographical gaps or low data density for reasons of local and national politics or economics – rich countries lots of good data, poor countries not so much), but they do constitute the best available, i.e. best practice. Climate data is queried using either WorldClim (Hijmans et al. 2005; Fick and Hijmans 2017) or one of its comparable global climate surfaces (interpolated met station data and digital elevation model coupled with some atmospheric physics to fine tune effects of slope, aspect, continentality etc). I would recommend the authors consider advocating as a standard that NLR methods use GBIF for the source of their extant taxon distributions, with a mathematical climate surface such as WorldClim (Hijmans et al. 2005; Fick and Hijmans 2017) as the analytical procedure to derive the climate range data.

Finally, a suggestion. In Ballantyne et al. (2010), Greenwood et al. (2017) and Lowe et al. (2018) we employ what we call a consensus or an ensemble approach, where multiple terrestrial climate proxies – including both geochemical and paleobotanical methods in some instances – are combined and assessed using probability density analysis. We think this is the solution to the problem of competing proxies; query the data to see where they overlap and are most consistent. I would invite the authors to consider recommending such an approach.

3. Technical corrections.

p. 4, line 5, typo: Paleogene, not Paleogne

p. 4, line 15, omission: cite Greenwood and Wing (1995) as the LAT global N & S

hemisphere latitudinal compilation predecessor of Huber and Caballero (2011).

p. 4, Line 32, typo? I would think 'end-member' rather than without the hyphen.

p. 5, line 24: cite also Eldrett et al. (2014) as well as Suan et al. (2017) as these authors document from terrestrial plant palynomorphs this same point for the PETM at high northern latitudes.

p. 22, lines 9-34: I would like to see mention here the concerns expressed by Eberle et al. (2010) on a possible seasonal bias in Arctic TEX86 reconstructions.

4. References cited.

Ballantyne, A.P., Greenwood, D.R., Sinninghe Damsté, J.S., Csank, A.Z., Eberle, J.J., and Rybczynski, N. 2010. Significantly warmer Arctic surface temperatures during the Pliocene indicated by multiple independent proxies. Geology, 38 (7): 603–606, doi:10.1130/G30815.1

Eberle, J.J., Fricke, H.C., Humphrey, J.D., Hackett, L., Newbrey, M.G. and Hutchison, J.H. 2010. Seasonal variability in Arctic temperatures during early Eocene time. Earth and Planetary Science Letters, 296(3-4): 481-486.

Eldrett, J. S., Greenwood, D. R., Polling, M., Brinkhuis, H., and Sluijs, A. 2014. A seasonality trigger for carbon injection at the Paleocene–Eocene Thermal Maximum. Clim. Past, 10: 759–769, https://doi.org/10.5194/cp-10-759-2014.

Fick, S.E. and Hijmans, R.J., 2017. WorldClim 2: new 1‐km spatial resolution climate surfaces for global land areas. International Journal of Climatology, 37(12): 4302-4315.

GBIF (GlobalBiodiversityInformationFacility). Open Access Biodiversity Data, available at: http://gbif.org Greenwood, D.R. and Wing, S.L., 1995. Eocene continental climates and latitudinal temperature gradients. Geology, 23(11): 1044-1048.

Greenwood, D.R., Basinger, J.F., Smith, R.Y. 2010. How wet was the Arctic Eocene

rainforest? Estimates of precipitation from Paleogene Arctic macrofloras. Geology, 38(1): 15–18, doi: 10.1130/G30218.1

Greenwood, D.R., Keefe, R.L., Reichgelt, T., and Webb, J.A. 2017. Eocene paleobotanical altimetry of Victoria's Eastern Uplands. Australian Journal of Earth Sciences, 64(5): 625ïǍ▪637.

Grimm, G. W. and Denk, T. 2012. Reliability and resolution of the coexistence approach- A revalidation using modern day data. Rev. Palaeobot. Palyno., 172; 33–47.

Grimm, G. W. and Potts, A. J. 2016. Fallacies and fantasies: the theoretical underpinnings of the Coexistence Approach for palaeoclimate reconstruction. Clim. Past, 12: 611–622, https://doi.org/10.5194/cp-12-611-2016

Harbert, R. and Nixon, K. 2015. Climate reconstruction analysis using coexistence likelihood estimation (CRACLE): A method for the estimation of climate using vegetation. Am. J. Bot., 102: 1277– 1289.

Hijmans, R.J., Cameron, S.E., Parra, J.L., Jones, P.G., and Jarvis, A. 2005. Very high resolution interpolated climate surfaces for global land areas, Int. J. Climatol., 25: 1965–1978.

Hyland, E.G., Huntington, K.W., Sheldon, N.D. and Reichgelt, T., 2018. Temperature seasonality in the North American continental interior during the Early Eocene Climatic Optimum. Clim. Past, 14(10): 1391-1404.

Lowe, A.J., Greenwood, D.R., West, C.K., Galloway, J.M., Reichgelt, T., and Sudermann, M. 2018. Plant community ecology and climate on an upland volcanic landscape during the Early Eocene Climatic Optimum: McAbee Fossil Beds, British Columbia, Canada. Palaeogeography, Palaeoclimatology, Palaeoecology 511: 433-448, doi:10.1016/j.palaeo.2018.09.010

Peppe, D.J., Royer, D.L., Cariglino, B., Oliver, S.Y., Newman, S., Leight, E., Enikolopov, G., FernandezâǍŘBurgos, M., Herrera, F., Adams, J.M. and Correa, E., 2011. Sensitivity of leaf size and shape to climate: global patterns and paleoclimatic applications. New Phytologist, 190(3): 724-739.

Pross, J., Contreras, L., Bijl, P.K., Greenwood, D.R., Bohaty, S.M., Schouten, S., Bendle, J.A., Röhl, U., Tauxe, L., Raine, J.I., Huck, C.E., van de Flierdt, T., Jamieson, S.S.R., Stickley, C.E., van de Schootbrugge, B., Escutia, C., Brinkhuis, H., and IODP Expedition 318 Scientists. 2012. Persistent near-tropical warmth on the Antarctic continent during the early Eocene epoch. Nature, 488: 73–77, doi: 10.1038/nature11300

Reichgelt, T., West, C.K. and Greenwood, D.R. 2018. The relation between global palm distribution and climate. Scientific Reports 8(1): 4721, doi: 10.1038/s41598-018-23147-2

Smith, R.Y., Basinger, J.F., and Greenwood, D.R. 2012. Early Eocene plant diversity and dynamics in the Falkland flora, Okanagan Highlands, British Columbia, Canada. Palaeobiodiversity and Palaeoenvironments, 92(3): 309–328, doi: 10.1007/s12549-011-0061-5

Suan, G., Popescu, S.M., Suc, J.P., Schnyder, J., Fauquette, S., Baudin, F., Yoon, D., Piepjohn, K., Sobolev, N.N. and Labrousse, L. 2017. Subtropical climate conditions and mangrove growth in Arctic Siberia during the early Eocene. Geology, 45(6): 539-542.

Sunderlin, D., Loope, G., Parker, N.E. and Williams, C.J., 2011. Paleoclimatic and paleoecological implications of a Paleocene–Eocene fossil leaf assemblage, Chickaloon Formation, Alaska. Palaios, 26(6): 335-345.

Wilf, P., Wing, S.L., Greenwood, D.R. and Greenwood, C.L., 1998. Using fossil leaves as paleoprecipitation indicators: an Eocene example. Geology, 26(3): 203-206.

---

## Referee Comment (RC2) · Marci Robinson (Referee) · 26 Apr 2019

Overview

This DeepMIP contribution outlines methods for compiling proxy land and sea temperature and $CO_2$ data for the latest Paleocene (LP), PETM and EECO across widely distributed sites in order to provide 1) insights into controls on warm climates and 2) boundary conditions and verification data for climate models. An initial database or

"atlas" is provided with the anticipation of expansion, and requirements for proxy methods and data reporting are listed. The authors have done a very nice job thoroughly explaining complex paleothermometry theory and methods for the non-expert. They offer a detailed discussion of the strengths and weaknesses of each proxy, which is expected as the author list contains many of the best experts on the proxies discussed in the paper. As I am an end user of data from many of these proxies, and not a geochemist, I am pleased to see such a complete description of each proxy. It is clear that this contribution will have the added benefit of serving as a reference for climate scientists who are not intimately familiar with the entire array of proxies.

General Comments

Building a global dataset of past climate proxy data is challenging because in the course of data collection, new proxies are developed, existing proxies are improved, and age models are refined. Especially important is the collection of raw data and original sample designations. I am pleased to see in this manuscript the importance placed on the reporting of raw clumped isotope data. This requirement should be extended to all proxy data. Since we know that proxies evolve and develop over time, the availability of raw data will allow for reanalysis and new applications should they appear. On a similar note, the authors mention that brGDGT datasets should be scrutinized for temperature plateaus, which would suggest that the calibration has saturated. This scrutiny should be applied to other proxies as well, including those calibrated to modern data sets that may not be able to estimate temperatures warmer than modern.

I appreciate this multi-proxy approach – collecting a diverse array of paleoenvironmental data with the goal of better understanding the climate system as a whole. This is a necessary step in developing a global temperature reconstruction and will ultimately aid in future analyses of more nuanced, regional, non-temperature reconstructions. As we have seen in other global paleoenvironmental reconstructions, paleothermometry proxies become useful in discerning other aspects of the climate system, like water column structure (e.g., thermocline depth) and ocean circulation (e.g., upwelling strength),

that will be useful as the models develop and will provide additional insight into climate dynamics. It is important to remember that these different temperature proxies record different aspects of "temperature", (i.e., the temperature at a certain water depth or during a certain season) and are likely complementary, yet not always directly comparable.

This DeepMIP atlas is a fantastic compilation of data and represents just the beginning of LP, PETM, and EECO paleoenvironmental global climate reconstruction. What's next? Specifically, how will this database be used in the DeepMIP models? Are the data compiled here enough to set model boundary conditions? I believe a short statement in the conclusions addressing these questions is warranted.

Specific Comments

In Section 6.5, briefly define C3 plants. So much detail has gone into defining the basis of the other proxies, but a simple description of why C3 plants are useful is not included.

Alkenones are not among the SST proxies discussed here, presumably because they are not terribly useful during these time intervals. TEX86 is recommended as a good proxy for when alkenones are not present or are outside their calibration range, and this would be a good place to mention that alkenones are scarce before 40 Ma (as is mentioned later) and that they saturate at temperatures below those expected during these warm intervals. Alkenones are, however, listed as a $CO_2$ proxy, though no alkenone-based $CO_2$ estimates are included in Supplementary Data File 8, possibly because none exist prior to the middle Eocene. Are alkenones included here because they could be a useful $CO_2$ proxy if they were found in LP, PETM and EECO sediments? What is the likelihood of this? Also, Supplementary Data File 8 is not listed in the text or in the supplementary contents in Supplementary Data File 1.

In Supplementary Data File 3, please better define "Setting" on the cover sheet. Are these water depths in meters? Are they modern or paleodepths? If "Setting" is the

same as "Environment" on the following sheets, please use the same terminology. Also, please explain the 2.5, 50 and 97.5 columns. I assume these are percentiles? What do the shaded cells of different colors mean? Please make any changes and explanations consistent across all data tables.

Technical Corrections

Please look at these pages/lines more closely for minor mistakes. I've suggested corrections below. 4/5: Paleogne - Paleogene 9/12: foraminifer - foraminifera 9/15: undertake - undertaken 9/17: biasin - bias in 11/2: are - is 11/22: allows - allow 17/11: foraminifera - foraminiferal 17/34: semi-quantitatively - semi-quantitative 20/10: includes - include 24/13: TEX86 (needs subscripts) 24/15: in for - for (delete "in") 26/24: pShouteroxy - proxy 31/12: 28C - 28°C 33/4: results - result 33/11: 25 - 25°C 35/9,9,14: Ma - Myrs 40/29: are - is 42/31: delete "from a single" 43/15: C3 (not subscript) 47:27: missing a parenthesis 49/30: 20° - 20°C 50/18: sample - sampled

Check the proper order of references. Some are ordered by year after the first author; others are ordered alphabetically by second author. Please also check for consistent punctuation in and around citation parentheses in the text.

Should Farquhar et al. (1982) on page 44, line 18 be cited as Farquhar and Sharkey (1982)? Should Goericke et al. (1994) on page 37, line 16 be cited as Goericke and Fry (1994)? Zeebe (2007) is cited once in the text on page 12, line 33, but two Zeebe (2007) references are listed. Which one is correct?

Check the following references for correct usage of a and b designations: Eagle et al., 2013 a and b Edgar et al., 2013 b Evans et al., 2018 a and b Schouten et al., 2013 a and b Sluijs et al., 2007 a and b

The following citations appear in the manuscript text but not in the References section: Bains et al., 1999: p91, l4 (Figure 2 caption) Crouch et al, in prep: p92, l6 (Figure 3 caption) D'Hondt and Zachos, 1993: p10, l7 Gradstein et al., 2004: p95, l7 (Figure 6

caption) Huff et al., 2003: p30, l25 Kennett and Stott, 1991: p91, l4 (Figure 2 caption) Liu et al., 2009: p23, l25 Lauretano et al., 2016: p90, l4 (Figure 1 caption) Royer et al., 2005: p30, l25 Si and Aubry, 2018: p8, l16 Thomas et al., 2002: p91, l4 (Figure 2 caption) Thompson et al., 2015: p32, l13 Torsvik et al., 2012: p96, l4 (Figure 7 caption) Urey, 1984: p8, l12

The following references appeared in the References section but not in the manuscript text: D'Hondt et al., 1994 Roij et al., 2016 Sluijs et al., 2008 Sluijs et al., 2009 Wilke et al., 2006
* * *

---

## Author Comment (AC1) · 3 Jun 2019

Reply to Reviewer 1 [in square brackets] 4 June 2019

1. General comments.

The paper by Hollis et al. provides a well written summary of the proxy methods used to derive past climate states and greenhouse gas levels from the geological

record – specifically for the latest Paleocene (LP), Paleocene-Eocene thermal maximum (PETM) and early Eocene climatic optimum (EECO). The focus on this geological interval is appropriate - as stated by the authors – as at around 56 Ma the world experienced a geologically short (220 ky) and strong warming, the PETM, that was followed by a series of additional hyperthermals (e.g., ETM-2) and the longer duration EECO from 53 to 50 Ma. The PETM in particular offers a possible analog for current and forecast climate warming due to its short duration and the range of pCO2 reconstructed for the PETM spanning current to projected greenhouse gas levels under different projections.

The authors provide a comprehensive yet succinct review of both paleontological proxies as well as geochemical proxies of temperature of waters and of the air, past CO2 levels, as well as a range of paleoclimate parameters including annual and seasonal precipitation based primarily on terrestrially derived proxies. Further, their review highlights the challenges – strengths and weaknesses – posed by such a wide range of methods and approaches, ranging from concerns over precision versus accuracy, alternative methods of analysis and interpretation, taphonomy, limited geographic coverage, and the issues around compiling such disparate data into a database that can be used by the paleoclimate modelling community (model-model and model-data comparisons). As such, the article largely succeeds, offering a set of guidelines – as the article states – for the selection, compilation and analysis of such proxy data.

The article closes with a proposal for an "atlas" of climate conditions arrayed on an agreed paleogeography for the three selected time intervals to constrain and provide insight into the mechanisms controlling past hyperthermals via 'database ver. 0.1'. The article draws on the expertise of a large set of authors – the DeepMIP team – whose research foci span these topics. The manuscript therefore represents a substantial contribution to modelling science and is appropriate for the journal. I have few concerns, and these mainly reflect topics dealt with in less depth than I would consider necessary, or restricting discussion to particular research teams, as detailed in

the next section. In my specific comments I have focused on my area of expertise; paleobotanical reconstruction of past pCO2 and climate, so "5 Terrestrial proxies for air temperature."

[Many thanks for the positive feedback and constructive review.]

2. Specific comments Section 5 in its title only states 'Terrestrial proxies for air temperature', when this section also considers precipitation (e.g., p. 31 line 24, in section 5.2 or in section 5.3 at line 15 on p. 32), and should be re-titled as 'Terrestrial proxies for climate'.

[We prefer to keep the section 5 title as it is because this paper is focussed on temperature proxies. Although many of the approaches discussed can be used to reconstruct precipitation, we wish to keep this manuscript focussed on temperature and CO2. Precipitation warrants full treatment in a separate paper.]

Further, as leaf physiognomic methods as well as NLR approaches have been applied to reconstructing latest Paleocene to EECO precipitation (e.g., Greenwood et al. 2010; Pross et al. 2012; Eldrett et al. 2014; West et al. 2015; Suan et al. 2017; Hyland et al. 2018), the leaf-based and NLR proxies for precipitation should be included, e.g., CLAMP estimates growing season precipitation as well as that of the wettest 3 months and the driest 3 months, but these are not differentiated on p. 31 line 24, nor discussed at any point in section 5.2.

[We agree that these suggestions are all important and we will action them in a subsequent paper with a greater focus on precipitation proxies.]

5.2 Leaf morphology-based approaches. This section is very focused on the univariate leaf margin analysis (LMA) and the multivariate CLAMP, and ignores other methods such as digital leaf physiognomy (DiLP; Peppe et al. 2011), and also (as noted above and below) barely mentions the use of leaf physiognomy for reconstructing precipitation (annual and seasonal values) in PETM and EECO studies (e.g., Leaf Area Analysis

[LAA] and DiLP). Ignoring DiLP is unfortunate, but perhaps forgivable as DiLP has been applied to very few Paleogene floras to date. However, LAA (Wilf et al. 1998; Peppe et al. 2011) has been applied to a wider set of floras, including LP to EECO floras (e.g., Greenwood et al. 2010; Sunderlin et al. 2011; Smith et al. 2012; West et al. 2015; Lowe et al. 2018) and – as noted – yields data on precipitation that is critical to the DeepMIP project.

[We agree that this section is focussed on LMA and CLAMP. We mention some other leaf morphology-based methods such as DiLP and the Nearest Neighbour Approach in Section 5.2.1, but do not provide details. This is primarily due to the relatively limited application of those other leaf morphology-based temperature proxies to date, and also to the limited scope for detailed discussion in this current manuscript. Precipitation proxies such as LAA were not discussed in any detail due to scope and space. The paper focus is on temperature for all proxies but the reviewer has brought an important point to our attention, that we do not make that sufficiently clear in Section 5, even though it is clearly stated in Section 1 Introduction. We certainly don't want to minimise the importance of precipitation and the hydrological regime in general, we believe this topic warrants another whole paper. This is the reason that LAA and the precipitation aspects of CLAMP were not discussed fully here.]

The discussion of weaknesses and strengths in this section is also out of date as it misses current and recent literature on LMA and CLAMP.

[Happy to be updated here, we certainly don't want to miss any key references. The most recent CLAMP methodology reference seems to be Yang et al. (2015), which we have included, but there are of course more recent articles that apply LMA/CLAMP, are these the ones being referred to as missing? We have now also included some of the additional references Reviewer 1 mentioned in his comments.]

Citing Carpenter et al. (2012) at line 22, p. 30 is idiosyncratic as an example as this paper proposed a method that has been never used in another study, whereas

LAA (Wilf et al. 1998) is cited, but this univariate method's application to estimate Paleogene annual precipitation is not mentioned, despite LAA being used in multiple studies of the LP to EECO interval.

[Carpenter et al. (2012) proxies are for temperature and are included in the Paleogene temperature database with this paper. It therefore seemed appropriate to include the reference in the main text as an example, even though it is certainly a fair point about the limited application of the proxies involved. A key LAA reference is cited, also as an example, but since it is a precipitation proxy, we did not discuss it further in this paper.]

5.3 Nearest Living Relative Analysis. I disagree that the 'most widely used method' is the Coexistence Approach (CA), particularly when applied to the LP to EECO interval of concern.

[We have revised this wording, noting that even if CA were the most widely used method, this does not imply that CA is the best method for the early Paleogene]

By focusing on a method that has been widely criticized – and further is not the most widely used NLR-proxy used in existing studies of the PETM and EECO – the authors create problems rather than solve them. My recommendation (as argued below) is that CA should not be the focus of recommendations. Grimm and Denk (2012) and Grimm and Potts (2016) posed useful concerns about the analytical approach that underpins the 'coexistence approach' as employed by the PALAEOFLORA group. Principally CA's reliance on single taxa to define upper and lower limits.

Grimm directs deep time climate researchers towards statistical methods such as CR-ACLE (Harbert and Nixon 2015) and an earlier iteration of my bioclimatic approach (Greenwood et al. 2005; see also Ballantyne et al. 2010) where we took a more objective nice succinct summary of these concerns and the more appropriate probability based approaches is given by Hyland et al. (2018).

In essence the concerns are that:

[Figure]

1) genera that are low species richness today (e.g., Ginkgo, taxodioid Cupressaceae such as Glyptostrobus, Metasequoia and Sequoia, etc) may be restricted today in their climate range for reasons other than climate, and may have occupied wider (or just different) climatic spaces in the Paleogene when they were clearly far more geographically widespread and so potentially more ecologically varied than today – this argues against using single taxa such as Ginkgo or Metasequoia (both of which are commonly present in LP, PETM and EECO macrofloras) to define hard limits;

2) that climatic tolerance may have evolved in some lineages – e.g., palms are often used in Paleogene hyperthermal studies to constrain winter temperatures, however pre-Eocene palms likely didn't include the most cold-tolerant subtribes present today as they were a late Eocene or even Miocene radiation (Reichgelt et al. 2018 and references therein);

3) issues with identification of the NLRs – plant taxa may show the same leaf or pollen morphology in Eocene fossils as today, but come from plants that were phylogenetically sister taxa – i.e. not the same genus – with morphologically different flowers or other plant organs to the NLR, so the climate tolerance of the fossil's NLR may be a non-match – e.g., Platanaceae and Betulaceae leaves and pollen present in many North American floras may show 'Platanus' or 'Carpinus' leaf or pollen morphology, but are from plants with extinct taxon flowers and other organs (Macginitea and Palaeocarpinus); methods like CA are inherently subjective because 'outliers' – e.g., those relictual taxa noted in point 1, or that 'sit outside' the climate range of the majority of taxa – are arbitrarily excluded from the CA, or arbitrarily used to set a limit for the range of possible estimates;

[We agree with these concerns and note that point 3 is an issue with all NLR approaches, not specific to CA. We have revised the text substantially in accordance with these comments and removed the focus on CA.]

4) in the past there were concerns about where NLR climate range data were coming

from – a lack of standardization of data sources and data quality control.

Due to all of the above concerns, a number of authors have adopted Grimm's recommendations or independently came to a similar solution (e.g., Ballantyne et al. 2010). For example, the mathematical approach developed in R by my colleague Tammo Reichgelt – a method with comparable assumptions to CRACLE – where we use probability density functions and Monte Carlo runs to select the highest probability climate estimate based on the whole suite of fossil taxa's NLRs, including taxa at different taxonomic ranks (Family, tribe or genus) to allow for differing degrees of confidence of matching fossils to an extant NLR.

[Use of probability density analysis solves several issues with NLR, and we now emphasise the advantages of this approach in our revision.]

Furthermore, Greenwood et al. (2017), Hyland et al. (2018) and Reichgelt et al. (2018) and others derive the NLR climate range data from the international online portal GBIF.org which warehouses distribution data from most of the world's university, museum and government herbaria and equivalent, with clear policies on data quality and ownership, so verifiable data. The GBIF records are not without bias (geographical gaps or low data density for reasons of local and national politics or economics – rich countries lots of good data, poor countries not so much), but they do constitute the best available, i.e. best practice. Climate data is queried using either WorldClim (Hijmans et al. 2005; Fick and Hijmans 2017) or one of its comparable global climate surfaces (interpolated met station data and digital elevation model coupled with some atmospheric physics to fine tune effects of slope, aspect, continentality etc).

I would recommend the authors consider advocating as a standard that NLR methods use GBIF for the source of their extant taxon distributions, with a mathematical climate surface such as WorldClim (Hijmans et al. 2005; Fick and Hijmans 2017) as the analytical procedure to derive the climate range data.

[The quality of the NLR database is of vital importance for the accuracy of the climate

estimates, and many of the existing paleodatabases have a strong regional bias and unclear policy in regard to ownership and data quality control. We are grateful that the reviewer raised this issue, and fully agree that GBIF and WorldClim are currently the best options available. We have amended the text accordingly.]

Finally, a suggestion. In Ballantyne et al. (2010), Greenwood et al. (2017) and Lowe et al. (2018) we employ what we call a consensus or an ensemble approach, where multiple terrestrial climate proxies – including both geochemical and paleobotanical methods in some instances – are combined and assessed using probability density analysis. We think this is the solution to the problem of competing proxies; query the data to see where they overlap and are most consistent. I would invite the authors to consider recommending such an approach.

[Yes, multiproxy approaches where possible should be advocated. We touched on this in the first sentence of section 5.2.4 but with insufficient direction. We have added a statement here and in 5.3.4 directing to the examples that Reviewer 1 gives above.]

3. Technical corrections. p. 4, line 5, typo: Paleogene, not Paleogne p. 4, line 15, omission: cite Greenwood and Wing (1995) as the LAT global N & S hemisphere latitudinal compilation predecessor of Huber and Caballero (2011). p. 4, Line 32, typo? I would think 'end-member' rather than without the hyphen. p. 5, line 24: cite also Eldrett et al. (2014) as well as Suan et al. (2017) as these authors document from terrestrial plant palynomorphs this same point for the PETM at high northern latitudes. p. 22, lines 9-34: I would like to see mention here the concerns expressed by Eberle et al. (2010) on a possible seasonal bias in Arctic TEX86 reconstructions.

[All corrections accepted]

4. References cited. [all added] Ballantyne, A.P., Greenwood, D.R., Sinninghe Damsté, J.S., Csank, A.Z., Eberle, J.J., and Rybczynski, N. 2010. Significantly warmer Arctic surface temperatures during the Pliocene indicated by multiple independent proxies. Geology, 38 (7): 603–606, doi:10.1130/G30815.1

Eberle, J.J., Fricke, H.C., Humphrey, J.D., Hackett, L., Newbrey, M.G. and Hutchison, J.H. 2010. Seasonal variability in Arctic temperatures during early Eocene time. Earth and Planetary Science Letters, 296(3-4): 481-486.

Eldrett, J. S., Greenwood, D. R., Polling, M., Brinkhuis, H., and Sluijs, A. 2014. A seasonality trigger for carbon injection at the Paleocene–Eocene Thermal Maximum. Clim. Past, 10: 759–769, https://doi.org/10.5194/cp-10-759-2014.

Fick, S.E. and Hijmans, R.J., 2017. WorldClim 2: new 1âËŸARËĞ km spatial resolution climate surfaces for global land areas. International Journal of Climatology, 37(12): 4302-4315.

GBIF (GlobalBiodiversityInformationFacility). Open Access Biodiversity Data, available at: http://gbif.org

Greenwood, D.R. and Wing, S.L., 1995. Eocene continental climates and latitudinal temperature gradients. Geology, 23(11): 1044-1048.

Greenwood, D.R., Basinger, J.F., Smith, R.Y. 2010. How wet was the Arctic Eocene rainforest? Estimates of precipitation from Paleogene Arctic macrofloras. Geology, 38(1): 15–18, doi: 10.1130/G30218.1

Greenwood, D.R., Keefe, R.L., Reichgelt, T., and Webb, J.A. 2017. Eocene paleobotanical altimetry of Victoria's Eastern Uplands. Australian Journal of Earth Sciences, 64(5): 625-637.

Grimm, G. W. and Denk, T. 2012. Reliability and resolution of the coexistence approach- A revalidation using modern day data. Rev. Palaeobot. Palyno., 172; 33–47.

Grimm, G. W. and Potts, A. J. 2016. Fallacies and fantasies: the theoretical underpinnings of the Coexistence Approach for palaeoclimate reconstruction. Clim. Past, 12: 611–622, https://doi.org/10.5194/cp-12-611-2016

Harbert, R. and Nixon, K. 2015. Climate reconstruction analysis using coexistence likelihood estimation (CRACLE): A method for the estimation of climate using vegetation. Am. J. Bot., 102: 1277– 1289.

Hijmans, R.J., Cameron, S.E., Parra, J.L., Jones, P.G., and Jarvis, A. 2005. Very high resolution interpolated climate surfaces for global land areas, Int. J. Climatol., 25: 1965–1978.

Hyland, E.G., Huntington, K.W., Sheldon, N.D. and Reichgelt, T., 2018. Temperature seasonality in the North American continental interior during the Early Eocene Climatic Optimum. Clim. Past, 14(10): 1391-1404.

Lowe, A.J., Greenwood, D.R., West, C.K., Galloway, J.M., Reichgelt, T., and Sudermann, M. 2018. Plant community ecology and climate on an upland volcanic landscape during the Early Eocene Climatic Optimum: McAbee Fossil Beds, British Columbia, Canada. Palaeogeography, Palaeoclimatology, Palaeoecology 511: 433-448, doi:10.1016/j.palaeo.2018.09.010

Peppe, D.J., Royer, D.L., Cariglino, B., Oliver, S.Y., Newman, S., Leight, E., Enikolopov, G., FernandezâËŸARËĞ Burgos, M., Herrera, F., Adams, J.M. and Correa, E., 2011. Sensi-tivity of leaf size and shape to climate: global patterns and paleoclimatic applications. New Phytologist, 190(3): 724-739.

Pross, J., Contreras, L., Bijl, P.K., Greenwood, D.R., Bohaty, S.M., Schouten, S., Bendle, J.A., Röhl, U., Tauxe, L., Raine, J.I., Huck, C.E., van de Flierdt, T., Jamieson, S.S.R., Stickley, C.E., van de Schootbrugge, B., Escutia, C., Brinkhuis, H., and IODP Expedition 318 Scientists. 2012. Persistent near-tropical warmth on the Antarctic continent during the early Eocene epoch. Nature, 488: 73–77, doi: 10.1038/nature11300

Reichgelt, T., West, C.K. and Greenwood, D.R. 2018. The relation between global palm distribution and climate. Scientific Reports 8(1): 4721, doi: 10.1038/s41598-018-23147-2

Smith, R.Y., Basinger, J.F., and Greenwood, D.R. 2012. Early Eocene plant diversity

and dynamics in the Falkland flora, Okanagan Highlands, British Columbia, Canada. Palaeobiodiversity and Palaeoenvironments, 92(3): 309–328, doi: 10.1007/s12549-011-0061-5

Suan, G., Popescu, S.M., Suc, J.P., Schnyder, J., Fauquette, S., Baudin, F., Yoon, D., Piepjohn, K., Sobolev, N.N. and Labrousse, L. 2017. Subtropical climate conditions and mangrove growth in Arctic Siberia during the early Eocene. Geology, 45(6): 539-542.

Sunderlin, D., Loope, G., Parker, N.E. and Williams, C.J., 2011. Paleoclimatic and paleoecological implications of a Paleocene–Eocene fossil leaf assemblage, Chickaloon Formation, Alaska. Palaios, 26(6): 335-345.

Wilf, P., Wing, S.L., Greenwood, D.R. and Greenwood, C.L., 1998. Using fossil leaves as paleoprecipitation indicators: an Eocene example. Geology, 26(3): 203-206.

Please also note the supplement to this comment:
https://www.geosci-model-dev-discuss.net/gmd-2018-309/gmd-2018-309-AC1-supplement.pdf

---

## Author Comment (AC2) · 4 Jun 2019

Reply to Reviewer 2 [in square brackets] 4 June 2019 ************

Overview This DeepMIP contribution outlines methods for compiling proxy land and sea temperature and $CO_2$ data for the latest Paleocene (LP), PETM and EECO across widely distributed sites in order to provide 1) insights into controls on warm climates and 2) boundary conditions and verification data for climate models. An initial database or

"atlas" is provided with the anticipation of expansion, and requirements for proxy methods and data reporting are listed. The authors have done a very nice job thoroughly explaining complex paleothermometry theory and methods for the non-expert. They offer a detailed discussion of the strengths and weaknesses of each proxy, which is expected as the author list contains many of the best experts on the proxies discussed in the paper. As I am an end user of data from many of these proxies, and not a geochemist, I am pleased to see such a complete description of each proxy. It is clear that this contribution will have the added benefit of serving as a reference for climate scientists who are not intimately familiar with the entire array of proxies. [We greatly appreciate this positive feedback and the very thorough review]

General Comments Building a global dataset of past climate proxy data is challenging because in the course of data collection, new proxies are developed, existing proxies are improved, and age models are refined. Especially important is the collection of raw data and original sample designations. I am pleased to see in this manuscript the importance placed on the reporting of raw clumped isotope data. This requirement should be extended to all proxy data. Since we know that proxies evolve and develop over time, the availability of raw data will allow for reanalysis and new applications should they appear. On a similar note, the authors mention that brGDGT datasets should be scrutinized for temperature plateaus, which would suggest that the calibration has saturated. This scrutiny should be applied to other proxies as well, including those calibrated to modern data sets that may not be able to estimate temperatures warmer than modern.

[The database includes all available raw data for SST proxies, GDGT-based LAT proxies and clumped isotope-based LAT proxies. For other datasets, readers are referred to the source articles. However, we agree that it is helpful to have all the raw data in a single database and we hope to include more raw data in future versions of the database. Temperature plateaus are also mentioned in relation to leaf- and pollen-based LAT estimates. Calibration saturation is not significant issue for other proxies

that benefit from culturing experiments in warmer than present conditions.]

I appreciate this multi-proxy approach – collecting a diverse array of paleoenvironmental data with the goal of better understanding the climate system as a whole. This is a necessary step in developing a global temperature reconstruction and will ultimately aid in future analyses of more nuanced, regional, non-temperature reconstructions. As we have seen in other global paleoenvironmental reconstructions, paleothermometry proxies become useful in discerning other aspects of the climate system, like water column structure (e.g., thermocline depth) and ocean circulation (e.g., upwelling strength), that will be useful as the models develop and will provide additional insight into climate dynamics. It is important to remember that these different temperature proxies record different aspects of "temperature", (i.e., the temperature at a certain water depth or during a certain season) and are likely complementary, yet not always directly comparable.

[Good point. We have added text to indicate that we focus on SST for oxygen isotopes and Mg/Ca ratios, but have tabulated data for thermocline species and referenced sources for benthic data sets. We have also added further comment on using benthic data to infer SST in polar regions]

This DeepMIP atlas is a fantastic compilation of data and represents just the beginning of LP, PETM, and EECO paleoenvironmental global climate reconstruction. What's next? Specifically, how will this database be used in the DeepMIP models? Are the data compiled here enough to set model boundary conditions? I believe a short statement in the conclusions addressing these questions is warranted.

[Good recommendation. We have added this statement to the conclusions. The Deep-MIP database (v. 0.1) is the first step towards a comprehensive compilation of climate proxy data for the latest Paleocene, PETM and EECO. In its current form, it improves upon pervious datasets that have been used to identify areas of agreement as well as mismatches between Paleogene climate data and climate models (e.g. Lunt et

al. 2012; Huber and Caballero, 2011). With further interrogation, using methodologies similar to those of Caballero and Huber (2013), we aim to derive estimates for global mean sea surface, land surface and overall surface temperature for the three time slices, which will help to advance our understanding of climate sensitivity under high CO2 conditions. The next challenge is then to add the full range of available climate data (e.g., salinity, precipitation, humidity, seasonal variability) to the DeepMIP database. With a recent study indicating that high growth/low mitigation scenarios herald a return to early Eocene-like conditions by the end of the century (Burke et al., 2018), the goals of DeepMIP are increasingly important and timely.]

Specific Comments In Section 6.5, briefly define C3 plants. So much detail has gone into defining the basis of the other proxies, but a simple description of why C3 plants are useful is not included.

[We've added the following to the text: Plants are differentiated into three main categories based in the method by which they fix carbon: C3, C4 and CAM. C3 plants have the simplest metabolic pathway in which the isotopic fractionation from atmospheric CO2 to plant tissue has been modelled (Farquhar et al., 1982).]

Alkenones are not among the SST proxies discussed here, presumably because they are not terribly useful during these time intervals. TEX86 is recommended as a good proxy for when alkenones are not present or are outside their calibration range, and this would be a good place to mention that alkenones are scarce before 40 Ma (as is mentioned later) and that they saturate at temperatures below those expected during these warm intervals. Alkenones are, however, listed as a CO2 proxy, though no alkenone-based CO2 estimates are included in Supplementary Data File 8, possibly because none exist prior to the middle Eocene. Are alkenones included here because they could be a useful CO2 proxy if they were found in LP, PETM and EECO sediments? What is the likelihood of this? Also, Supplementary Data File 8 is not listed in the text or in the supplementary contents in Supplementary Data File 1.

[We have added a section on alkenones as SST proxies, and discussed their potential for future early Eocene paleotemperature and paleo-CO2 reconstructions.]

In Supplementary Data File 3, please better define "Setting" on the cover sheet. Are these water depths in meters? Are they modern or paleodepths? If "Setting" is the same as "Environment" on the following sheets, please use the same terminology. [Corrected]

Also, please explain the 2.5, 50 and 97.5 columns. I assume these are percentiles? What do the shaded cells of different colors mean? Please make any changes and explanations consistent across all data tables. [Colours and columns are now defined.]

Technical Corrections Please look at these pages/lines more closely for minor mistakes. I've suggested corrections below. 4/5: Paleogne - Paleogene [done] 9/12: foraminifer - foraminifera [done] 9/15: undertake - undertaken [done] 9/17: biasin - bias in [done] 11/2: are - is [done] 11/22: allows - allow [done] 17/11: foraminifera - foraminiferal [done] 17/34: semi-quantitatively - semi-quantitative [done] 20/10: includes - include [done] 24/13: TEX86 (needs subscripts) [done] 24/15: in for - for (delete "in") [done] 26/24: pShouteroxy - proxy [done] 31/12: 28C - 28âŲ̊ęC [done] 33/4: results - result [done] 33/11: 25 - 25âŲ̊ęC [done] 35/9,9,14: Ma - Myrs [done, but changed to "my", which is the abbreviation used earlier] 40/29: are - is [done] 42/31: delete "from a single" [done] 43/15: C3 (not subscript) [done] 47:27: missing a parenthesis [done] 49/30: 20âŲ̊ę - 20âŲ̊ęC [done] 50/18: sample – sampled [done]

Check the proper order of references. Some are ordered by year after the first author; others are ordered alphabetically by second author. Please also check for consistent punctuation in and around citation parentheses in the text. [All corrected]

Should Farquhar et al. (1982) on page 44, line 18 be cited as Farquhar and Sharkey (1982)? Should Goericke et al. (1994) on page 37, line 16 be cited as Goericke and Fry (1994)? Zeebe (2007) is cited once in the text on page 12, line 33, but two Zeebe (2007) references are listed. Which one is correct? [All corrected and added Farquar

et al. (1982) to refs]

Check the following references for correct usage of a and b designations: Eagle et al., 2013 a and b Edgar et al., 2013 b Evans et al., 2018 a and b Schouten et al., 2013 a and b Sluijs et al., 2007 a and b

[All checked and corrected]

The following citations appear in the manuscript text but not in the References section: • Bains et al., 1999: p91, l4 (Figure 2 caption) [added] • Crouch et al, in prep: p92, l6 (Figure 3 caption) [added] • D'Hondt and Zachos, 1993: p10, l7 [corrected to D'Hondt et al., 1994] • Gradstein et al., 2004: p95, l7 (Figure 6 caption) [added] • Huff et al., 2003: p30, l25 [added] • Kennett and Stott, 1991: p91, l4 (Figure 2 caption) [added] • Liu et al., 2009: p23, l25 [added] • Lauretano et al., 2016: p90, l4 (Figure 1 caption) [corrected to 2015] • Royer et al., 2005: p30, l25 [added] • Si and Aubry, 2018: p8, l16 [added] • Thomas et al., 2002: p91, l4 (Figure 2 caption) [added] • Thompson et al., 2015: p32, l13 [added] • Torsvik et al., 2012: p96, l4 (Figure 7 caption) [added] • Urey, 1984: p8, l12 [corrected to 1951]

The following references appeared in the References section but not in the manuscript text: • D'Hondt et al., 1994 [see above, added to MS] • Roij et al., 2016 [deleted] • Sluijs et al., 2008 [deleted] • Sluijs et al., 2009 [deleted] • Wilke et al., 2006 [deleted]

Please also note the supplement to this comment:
https://www.geosci-model-dev-discuss.net/gmd-2018-309/gmd-2018-309-AC2-supplement.pdf

---

## Referee Comment (RC3) · Navjit Sagoo (Referee) · 9 Jun 2019

Overview This paper provides a comprehensive summary of the methods of data compilation and analysis for three intervals of the DeepMIP project: the latest Paleocene (LP), Paleocene-thermal maximum (PETM) and the early Eocene climatic optimum (EECO). It also describes the planned design, structure and use of the database that the data will eventually be stored in and the envisaged scientific workflow that will

enable reanalysis of data sets, application of alternative / new age models and data reproducibility and finally a preliminary synthesis of the data.

General Comments • This paper will provide a valuable resource not only for the DeepMIP project, but also for providing a detailed overview and strengths and weaknesses of many different proxy methods available for terrestrial and marine data, some of which I was previously unfamiliar with. • The planned design of the database should allow for easier updating of data as new methods / age models become available, and I am very pleased that the authors are asking for raw data to be archived as this will provide a very useful resource. • Overall the text and figures are clear, the content is well written and structured well. The manuscript would benefit from a thorough review by the authors to correct simple typos and inconsistencies.

Specific Comments • C3 land plant carbon isotope proxy for atmospheric CO2 I am unfamiliar with this proxy and therefore sought additional information about this approach. In doing so I came across a recent paper that argues that the model presented in this work (Schubert and Jahren, 2012) fails independent validation tests (see below). The authors discuss validation against ice core data in this section, which given that this is much lower CO2 than for the Paleocene / Eocene, seems insufficient. In addition the positive and negative errors stated in this section (120% and 40% seemed large). Could the authors comment on the findings of the Lomax et al., (2019) paper, which indicate that this proxy is poorly constrained, when growth conditions / water use efficiency is not well understood. Could the authors either update the paper with clear caveats or this section be removed from the paper?

Lomax et al., (2019) carry out validation experiments for this proxy and their results indicate that. . . 'the model fails to accurately predict CO2 with substantial under prediction in CO2 in experiments that were designed to simulate Cenozoic and Mesozoic atmospheric environments. Our findings suggest serious limitations in the proposed proxy as delivered estimates of CO2 are neither precise nor accurate when compared to known growth conditions.'

[Figure]

General typos • Partly based on the lack of clarity with the C3 land plant carbon isotope proxy for atmospheric CO2 described above, would the authors consider adding a summary table that could list all the proxy methods and a few columns that briefly convey, • for example, their maturity (i.e. new in progress, established, well used / understood), a measure of confidence in the method and perhaps whether there is a consistent bias, in order to convey some of this important information to the reader as an overview? • P5, L6 expand CIE here and use abbreviation from here on • P9, L17 bias in • P26, L24 proxy • P33, L11 °C • P33, L24 combining • P42, L31 from a single typically? Unclear sentence • P45, L17 . . .such that it becomes an updated, comprehensive tool . . . (possible change of text) • P46, L10 Southern Ocean

References Lomax, B. H., C. A. Knight, and J. A. Lake. "An experimental evaluation of the use of C3 $\delta$13C plant tissue as a proxy for the paleoatmospheric $\delta$13CO2 signature of air." Geochemistry, Geophysics, Geosystems 13.9 (2012).

---

## Author Comment (AC3) · 15 Jun 2019

We appreciate the positive comments from the reviewer and have addressed the typos and inconsistencies identified by all three reviewers and co-authors. Our responses to other comments are provided below [in square brackets]

Specific Comments: C3 land plant carbon isotope proxy for atmospheric CO2 I' am unfamiliar with this proxy and therefore sought additional information about this approach.

[Figure]

In doing so I came across a recent paper that argues that the model presented in this work (Schubert and Jahren, 2012) fails independent validation tests (see below). The authors discuss validation against ice core data in this section, which given that this is much lower CO2 than for the Paleocene / Eocene, seems insufficient. In addition, the positive and negative errors stated in this section (120% and 40% seemed large). Could the authors comment on the findings of the Lomax et al., (2019) paper, which indicate that this proxy is poorly constrained, when growth conditions / water use efficiency is not well understood. Could the authors either update the paper with clear caveats or this section be removed from the paper? Lomax et al., (2019) carry out validation experiments for this proxy and their results indicate that... 'the model fails to accurately predict CO2 with substantial under prediction in CO2 in experiments that were designed to simulate Cenozoic and Mesozoic atmospheric environments. Our findings suggest serious limitations in the proposed proxy as delivered estimates of CO2 are neither precise nor accurate when compared to known growth conditions.'

[Lomax et al. (2019) demonstrated experimentally that plants grown under water stress showed the same relative change in $\Delta$13C value with increasing CO2 concentration as observed in well-watered experiments (e.g., Schubert and Jahren, 2012), but with lower $\Delta$13C values overall. Their experimental data are consistent with modelled results for the predicted effect of water stress on $\Delta$13C value across multiple levels of CO2 (see Fig. 4a within Schubert and Jahren, 2018). We now note, however, that the approach used by Lomax et al. (2019), and also by Barral et al. (2017) for the Cretaceous, cannot be used to accurately predict CO2. We include these caveats and the following new text in section 6.5.4 Recommended methodology of C3 land plant proxy:

"However, Lomax et al. (2019) demonstrated experimentally that plants grown under water stress showed the same relative change in $\Delta$13C value with increasing CO2 concentration as observed in well-watered experiments (e.g., Schubert and Jahren, 2012), but with lower $\Delta$13C values overall. Their experimental data are consistent with results that modelled the predicted effect of water stress on $\Delta$13C value across

multiple levels of CO2 (see Fig. 4a within Schubert and Jahren, 2018). Nevertheless, both chamber (Lomax et al., 2019) and deep-time (Barral et al., 2017) studies have tried to use Eq. (26) to quantify CO2 and failed to account for the effects of growing conditions and plant species type on $\Delta$13C value. Such attempts highlight the need to use Eq. (27) with robust estimates of $\Delta(\Delta$13C) (after Schubert and Jahren, 2015) in order to accurately quantify CO2." (Equation numbers refer to revised text.)]

General typos: Partly based on the lack of clarity with the C3 land plant carbon isotope proxy for atmospheric CO2 described above, would the authors consider adding a summary table that could list all the proxy methods and a few columns that briefly convey, for example, their maturity (i.e. new in progress, established, well used understood), a measure of confidence in the method and perhaps whether there is a consistent bias, in order to convey some of this important information to the reader as an overview?

[We agree that a summary table of the proxies covered in the paper would be helpful to readers, and so propose to include one in the final version. However, it's difficult to include all the aspects mentioned above in a table. Even the most mature proxy, oxygen isotopes, has a wide range of sources of error and bias, which are best addressed in the text. Instead we add a column to the table to indicate how many of the proxy records included in the data compilation are considered to be reliable.]

P5, L6 expand CIE here and use abbreviation from here on [Corrected] P9, L17 bias in [Corrected] P26, L24 proxy [Corrected] P33, L11 âŮęC [Corrected] P33, L24 combining [Corrected] P42,$'$ L31 from a single typically? Unclear sentence [Corrected] P45, L17$'$ ...such that it becomes an updated, comprehensive tool ... (possible change of text) [Revised] P46, L10 Southern Ocean [Corrected] References Lomax, B. H., C. A. Knight, and J. A. Lake. "An experimental evaluation of the use of C3 $\delta$13C plant tissue as a proxy for the paleoatmospheric $\delta$13CO2 signature of air." Geochemistry, Geophysics, Geosystems 13.9 (2012).

Please also note the supplement to this comment:

https://www.geosci-model-dev-discuss.net/gmd-2018-309/gmd-2018-309-AC3-supplement.pdf

**Supplement:**

**Navjit Sagoo (Referee)** navjit.sagoo@misu.su.se

Overview This paper provides a comprehensive summary of the methods of data compilation and analysis for three intervals of the DeepMIP project: the latest Paleocene (LP), Paleocene-thermal maximum (PETM) and the early Eocene climatic optimum (EECO). It also describes the planned design, structure and use of the database that the data will eventually be stored in and the envisaged scientific workflow that will enable reanalysis of data sets, application of alternative / new age models and data reproducibility and finally a preliminary synthesis of the data.

General Comments: This paper will provide a valuable resource not only for theˊ DeepMIP project, but also for providing a detailed overview and strengths and weaknesses of many different proxy methods available for terrestrial and marine data, some of which I was previously unfamiliar with. The planned design of the databaseˊ should allow for easier updating of data as new methods / age models become available, and I am very pleased that the authors are asking for raw data to be archived as this will provide a very useful resource. Overall the text and figures are clear,ˊ the content is well written and structured well. The manuscript would benefit from a thorough review by the authors to correct simple typos and inconsistencies.

[We appreciate these positive comments and have addressed the typos and inconsistencies identified by all three reviewers and co-authors]

Specific Comments: C3 land plant carbon isotope proxy for atmospheric CO2 Iˊ am unfamiliar with this proxy and therefore sought additional information about this approach. In doing so I came across a recent paper that argues that the model presented in this work (Schubert and Jahren, 2012) fails independent validation tests (see below). The authors discuss validation against ice core data in this section, which given that this is much lower CO2 than for the Paleocene / Eocene, seems insufficient. In addition, the positive and negative errors stated in this section (120% and 40% seemed large). Could the authors comment on the findings of the Lomax et al., (2019) paper, which indicate that this proxy is poorly constrained, when growth conditions / water use efficiency is not well understood. Could the authors either update the paper with clear caveats or this section be removed from the paper?

Lomax et al., (2019) carry out validation experiments for this proxy and their results indicate that… 'the model fails to accurately predict CO2 with substantial under prediction in CO2 in experiments that were designed to simulate Cenozoic and Mesozoic atmospheric environments. Our findings suggest serious limitations in the proposed proxy as delivered estimates of CO2 are neither precise nor accurate when compared to known growth conditions.'

[Lomax et al. (2019) demonstrated experimentally that plants grown under water stress showed the same relative change in $\Delta^{13}C$ value with increasing $CO_2$ concentration as observed in well-watered experiments (e.g., Schubert and Jahren, 2012), but with lower $\Delta^{13}C$ values overall. Their experimental data are consistent with modelled results for the

predicted effect of water stress on $\Delta^{13}C$ value across multiple levels of $CO_2$ (see Fig. 4a within Schubert and Jahren, 2018). We now note, however, that the approach used by Lomax et al. (2019), and also by Barral et al. (2017) for the Cretaceous, cannot be used to accurately predict $CO_2$. We include these caveats and the following new text in section 6.5.4 Recommended methodology of $C_3$ land plant proxy:

"However, Lomax et al. (2019) demonstrated experimentally that plants grown under water stress showed the same relative change in $\Delta^{13}C$ value with increasing $CO_2$ concentration as observed in well-watered experiments (e.g., Schubert and Jahren, 2012), but with lower $\Delta^{13}C$ values overall. Their experimental data are consistent with results that modelled the predicted effect of water stress on $\Delta^{13}C$ value across multiple levels of $CO_2$ (see Fig. 4a within Schubert and Jahren, 2018). Nevertheless, both chamber (Lomax et al., 2019) and deep-time (Barral et al., 2017) studies have tried to use Eq. (26) to quantify $CO_2$ and failed to account for the effects of growing conditions and plant species type on $\Delta^{13}C$ value. Such attempts highlight the need to use Eq. (27) with robust estimates of $\Delta(\Delta^{13}C)$ (after Schubert and Jahren, 2015) in order to accurately quantify $CO_2$." (Equation numbers refer to revised text.)]

General typos: Partly based on the lack of clarity with the C3 land plant carbon isotope proxy for atmospheric CO2 described above, would the authors consider adding a summary table that could list all the proxy methods and a few columns that briefly convey, for example, their maturity (i.e. new in progress, established, well used understood), a measure of confidence in the method and perhaps whether there is a consistent bias, in order to convey some of this important information to the reader as an overview?

[We agree that a summary table of the proxies covered in the paper would be helpful to readers, and so propose to include one in the final version. However, it's difficult to include all the aspects mentioned above in a table. Even the most mature proxy, oxygen isotopes, has a wide range of sources of error and bias, which are best addressed in the text. Instead we add a column to the table to indicate how many of the proxy records included in the data compilation are considered to be reliable.]

P5, L6 expand CIE here and use abbreviation from here on [Corrected]

P9, L17 bias in [Corrected]

P26, L24 proxy [Corrected]

P33, L11 ∘C [Corrected]

P33, L24 combining [Corrected]

P42,´ L31 from a single typically? Unclear sentence [Corrected]

P45, L17´ …such that it becomes an updated, comprehensive tool … (possible change of text) [Revised]

P46, L10 Southern Ocean [Corrected]

References Lomax, B. H., C. A. Knight, and J. A. Lake. "An experimental evaluation of the use of C3 $\delta$13C plant tissue as a proxy for the paleoatmospheric $\delta$13CO2 signature of air." Geochemistry, Geophysics, Geosystems 13.9 (2012).

---

## Author Comment (AC4) · 16 Jun 2019

We appreciate your interest in this important aspect of our study. A preliminary analysis produced a similar GMST estimate for the EECO to that published by Caballero and Huber (2013). However, full characterization of the uncertainty in global mean temperature estimates (surface, SST, LAT) for the latest Paleocene, PETM and EECO is an important research topic in its own right and beyond the scope of this paper. A

follow-up paper is in preparation on this important topic.

Reference: Caballero, R., and Huber, M.: State-dependent climate sensitivity in past warm climates and its implications for future climate projections, Proceedings of the National Academy of Sciences, doi: 10.1073/pnas.1303365110, 2013.